# LEADeR role of miR-205 host gene as long noncoding RNA in prostate basal cell differentiation

Valentina Profumo[1], Barbara Forte [1], Stefano Percio [1], Federica Rotundo [1], Valentina Doldi [1], Elena Ferrari[1], Nicola Fenderico [2], Matteo Dugo [1], Dario Romagnoli[3], Matteo Benelli[3], Riccardo Valdagni[4,5,6], Diletta Dolfini [7], Nadia Zaffaroni [1] & Paolo Gandellini [1]

Though miR-205 function has been largely characterized, the nature of its host gene, MIR205HG, is still completely unknown. Here, we show that only lowly expressed alternatively spliced MIR205HG transcripts act as de facto pri-miRNAs, through a process that involves Drosha to prevent unfavorable splicing and directly mediate miR-205 excision. Notably, MIR205HG-specific processed transcripts revealed to be functional per se as nuclear long noncoding RNA capable of regulating differentiation of human prostate basal cells through control of the interferon pathway. At molecular level, MIR205HG directly binds the promoters of its target genes, which have an Alu element in proximity of the Interferon-Regulatory Factor (IRF) binding site, and represses their transcription likely buffering IRF1 activity, with the ultimate effect of preventing luminal differentiation. As MIR205HG functions autonomously from (albeit complementing) miR-205 in preserving the basal identity of prostate epithelial cells, it warrants reannotation as LEADeR (Long Epithelial Alu-interacting Differentiation-related RNA).

[1] Department of Applied Research and Technological Development, Fondazione IRCCS Istituto Nazionale dei Tumori, Milan 20133, Italy. [2] Oncode Institute and Department of Cell Biology, Centre for Molecular Medicine, University Medical Centre Utrecht, Utrecht 3584 CX, The Netherlands. [3] Centre for Integrative Biology, University of Trento, Trento 38123, Italy. [4] Department of Oncology and Hemato-oncology, University of Milan, Milan 20133, Italy. [5] Prostate Cancer Program, Fondazione IRCCS Istituto Nazionale dei Tumori, Milan 20133, Italy. [6] Radiation Oncology 1, Fondazione IRCCS Istituto Nazionale dei Tumori, Milan 20133, Italy. [7] Department of Biosciences, University of Milan, Milan 20133, Italy. These authors contributed equally: Valentina Profumo, Barbara Forte, Stefano Percio, Federica Rotundo.  Correspondence and requests for materials should be addressed to P.G. (email: paolo.gandellini@istitutotumori.mi.it)

Normally expressed in the basal layers of epithelia[1–3], *miR-205* acts as keeper of the epithelial phenotype. In mice mammary glands, it appears implicated in normal stem cell maintenance[4]. Consistent with this concept, different studies[3,5] observed perinatal lethality in *miR-205* knock-out mice due to severe skin defects deriving from the impairment of stem/ progenitor cell function. In human prostate basal cells, *miR-205* regulates the deposition of the basement membrane, a layer of specialized extracellular matrix that surrounds normal glands to ensure correct tissue polarity and morphogenesis[2].

The expression of *miR-205* was reported as either up or downregulated in human cancers[6], suggesting context-dependent oncogenic or tumor-suppressive functions. In particular, we showed that in prostate adenocarcinoma (PRAD) *miR-205* is almost invariably downmodulated and acts as a tumor suppressor by impinging on various processes, including the repression of epithelial–mesenchymal transition[7], the disruption of tumor–stroma interplay[8] and the impairment of autophagic flux[9]. An in vivo validation of *miR-205* oncosuppressive function was provided by the development of spontaneous mammary tumors in *miR-205*-deficient mice[10].

Human pre-*miR-205* sequence is located in the last intron/exon junction of a gene initially termed *NPCA-5* (alias *LOC642587*), which covers 3.7 kb on chromosome 1q32 and is transcribed into an 899-nt long processed transcript (NM_001104548, hereafter RefSeq) (Supplementary Fig. 1). As its biological function is still unexplored, the gene has been recently ex officio reannotated as *miR-205* Host Gene (*MIR205HG*).

In this work we characterize for the first time the expression pattern and role of *MIR205HG*, showing that (i) *MIR205HG* is mainly expressed in the basal layer of prostate epithelium and lost in PRAD, (ii) the Drosha-mediated processing of specific alternative transcripts of the gene is responsible for *miR-205* production, and (iii) *MIR205HG* functions independently of the hosted miRNA as nuclear intergenic long noncoding RNA (lincRNA) capable of regulating basal-luminal differentiation through repression of the interferon pathway. Mechanistically, the lincRNA directly binds the promoters of target genes, characterized by the presence of an *Alu* element in proximity of an interferon-regulatory factor (IRF) binding site, and buffers IRF1 transcription factor (TF) activity. Because *MIR205HG* processed transcript operates autonomously from *miR-205*, we will refer to it as *LEADeR* (*LEADR*).

## Results

### LEADR levels decrease upon basal–luminal differentiation.

Interrogation of publicly available transcriptomic data revealed that *LEADR/MIR205HG* is normally expressed in epithelia such as skin, prostate and breast, and almost absent in tissues of different embryonic origin (Fig. 1a). Accordingly, histone methylation/acetylation and chromatin state segmentation patterns among ENCODE cell lines indicate active transcription in keratinocytes and mammary epithelial cells compared to other cell types (Supplementary Fig. 1). TCGA data show *LEADR/ MIR205HG* upregulation in tumors with basal phenotype (e.g., cervical and lung squamous cell cancers) and downregulation in breast and prostate adenocarcinomas compared to their normal counterparts, thus mirroring *miR-205* modulations (Fig. 1b; Supplementary Fig. 2a). Reduction of *LEADR/MIR205HG* expression in PRAD was confirmed in one of the largest available microarray datasets (GSE21034), where its levels tend to decrease progressively as the tumor acquires a more undifferentiated or metastatic phenotype (Fig. 1c). In both TCGA and GSE21034 cohorts, *LEADR/MIR205HG* expression alone was able to discriminate tumor vs. normal samples (Fig. 1d), suggesting that

*LEADR/MIR205HG* loss may be an inescapable early event in prostate carcinogenesis. By contrast, no association was found between *LEADR/MIR205HG* expression in the primary tumor and time to biochemical recurrence after surgery (Supplementary Fig. 2b). Among the different cell types composing normal prostate epithelium, *LEADR/MIR205HG* appeared more abundant in basal cells than in luminal, stromal, or endothelial cells (Fig. 1e, Supplementary Fig. 2c). This finding could explain the gene's invariably low expression in PRAD, which is characterized by loss of the basal cell layer, as well as its increased expression in basal/squamous cancers. In addition, in the available prostate cell models, *LEADR/MIR205HG* expression was abundant only in normal cells with basal features, but was reduced in normal cells with luminal phenotype and almost negligible in all of the tested PRAD cell lines (Fig. 1f). Interestingly, when we allowed basal cells to differentiate by increasing calcium and serum concentration, *LEADR/MIR205HG* levels decreased as cells acquired a luminal phenotype (Fig. 1g). Similarly, data from the GSE89050 dataset showed a linear trend in the reduction of *LEADR/ MIR205HG* abundance when comparing frankly basal cells, luminal progenitors and fully differentiated luminal cells sorted from human prostate (Fig. 1h). Altogether these data suggest an epithelial-restricted and basal-enriched expression of *LEADR/ MIR205HG*, which is reduced upon luminal differentiation.

### miR-205 compatible and incompatible LEADR transcripts.

Analysis of TCGA revealed *LEADR/MIR205HG* expression as correlated to that of both the hosted miRNA and p63 (Supplementary Fig. 3a). *LEADR/MIR205HG* expression also responded to p63 modulation in both prostate basal and cervical carcinoma cells (Supplementary Fig. 3b, c), suggesting direct regulation. In this regard, we already demonstrated that p63 binds to sequences at −13Kb and +2Kb from *LEADR/MIR205HG* transcription start site (TSS), which we proposed as regions responsible for *miR-205* regulation[2]. Though the peculiar genomic location of pre-*miR-205* (*miR-205* in the intron and *miR-205\** in the exon, Fig. 2a) could account for the existence of a miRNA-specific promoter, experimental data undermine the hypothesis that *miR-205* biogenesis may be independent of that of *LEADR*. In fact, (i) the region immediately upstream of pre-*miR-205* showed no promoter activity in reporter assays[11], (ii) the closest 5′ end of nascent *miR-205* transcript coincides with *LEADR/MIR205HG* TSS[12], (iii) CRISPR/Cas9 genomic deletion of *LEADR* exons 1–3, including the TSS (Supplementary Fig. 3d), abolished expression of both *LEADR* processed transcript and *miR-205* in RWPE-1 cells (Fig. 2b), and (iv) transfection of the whole *LEADR* genomic sequence under constitutive CMV promoter induced the expression of both *LEADR* and *miR-205* in p63 null DU145 cells (Fig. 2c). All these observations suggest that *LEADR/MIR205HG* and *miR-205* are produced from a unique transcription unit. In our hands, gapmer oligonucleotides designed to target introns of *LEADR/MIR205HG* primary sequence ahead of pre-*miR-205* (Fig. 2a, gapINT1 and gapINT2) were able to significantly abrogate both *LEADR/MIR205HG* and *miR-205* expression (Fig. 2d), confirming that the two RNAs are processed from a common primary transcript.

Canonical *LEADR/MIR205HG* RefSeq configuration, however, seems incompatible with *miR-205* production, because the use of the splice donor site immediately upstream of the final exon would disrupt pre-*miR-205* hairpin (Fig. 2a). To understand whether alternative locus configurations could support miRNA biogenesis, we retrieved data on transcript structures from most recent annotations. Specifically, starting from data generated through targeted RNA capture with third-generation long-read sequencing technology[13], we manually annotated as high-

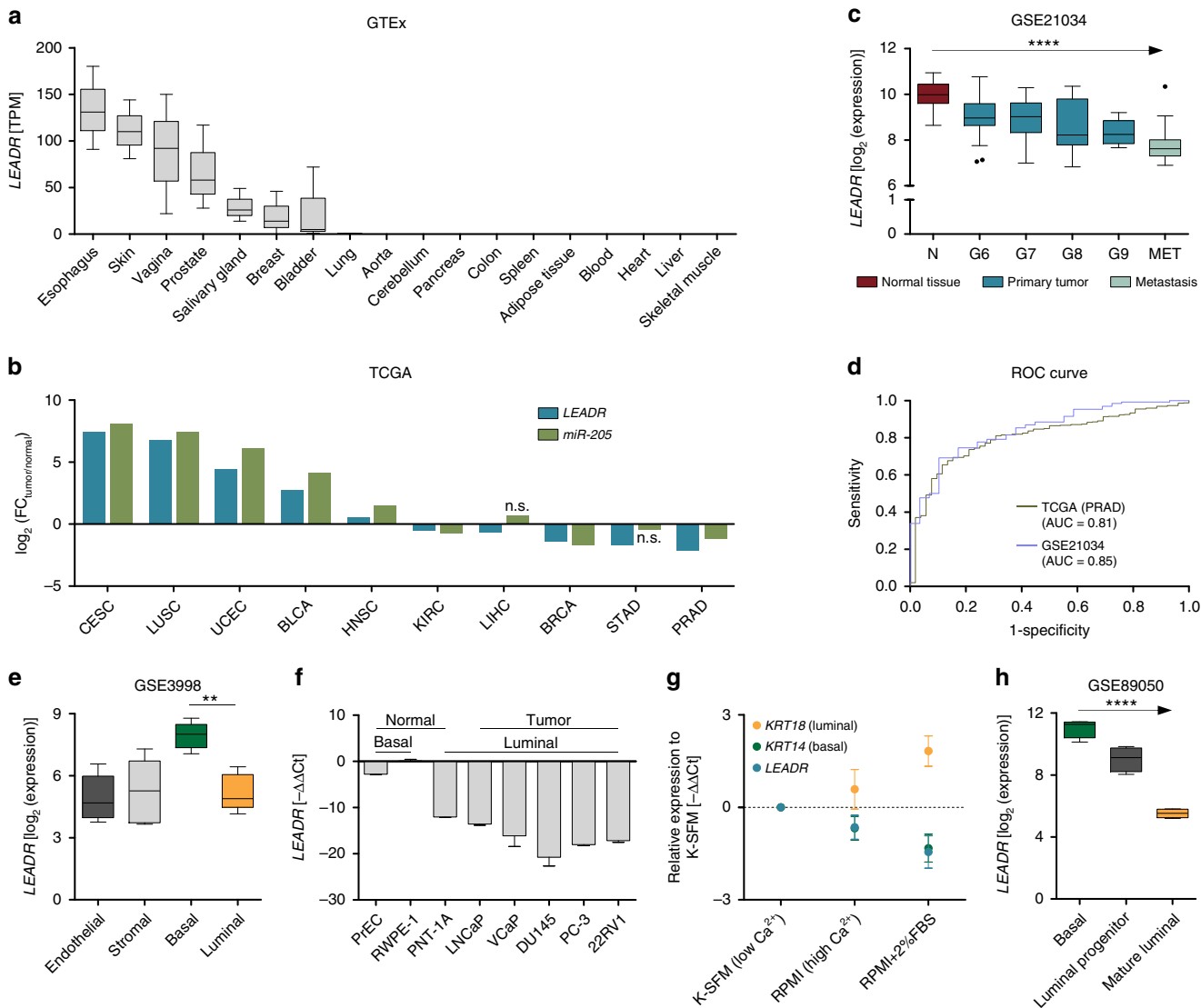

**Fig. 1** *LEADR/MIR205HG* expression is enriched in prostate basal cells and reduced upon luminal differentiation. **a** Tukey's box plot of *LEADR/MIR205HG* expression in human tissues as retrieved from GTEx data portal. **b** Bar plot of *LEADR/MIR205HG* and *miR-205* fold-change (FC) in tumor vs. normal samples from TCGA. Only tumors where *LEADR/MIR205HG* is significantly differentially expressed compared to normal counterpart are shown (Student's *t* test, *p* value threshold 0.05). Note that for all tumors where *LEADR/MIR205HG* is differentially expressed, also *miR-205* is differentially expressed and in the same direction, except for LIHC and STAD, where the difference is not significant (n.s.). All tumor/normal pairs together with acronym meaning are reported in Supplementary Fig. 2a. **c** Tukey's box plot of *LEADR/MIR205HG* expression in normal and tumor prostate samples from GSE21034 dataset. Jonckheere–Terpstra test evidences a significant decreasing trend along tumor progression. Samples are classified as normal (N), primary tumors (ranked based on Gleason pattern score grading system from G6 to G9), and metastatic (MET). **d** ROC curves of the performance of *LEADR/MIR205HG* in classifying tumor and normal tissues from two independent datasets (PRAD-TCGA and GSE21034). Area under the curve (AUC) is reported. **e** Tukey's box plot of *LEADR/MIR205HG* expression in different cell subpopulations from human normal prostate (GSE3998, Student *t* test). **f** qRT-PCR reporting *LEADR/MIR205HG* expression levels in different prostate cell lines (normal prostate total RNA used as reference). Mean + s.d. (*n* = 3 qRT-PCR measurements for each cell line) plotted. **g** qRT-PCR showing progressive downregulation of *LEADR/MIR205HG* along basal–luminal differentiation induced in RWPE-1 cells by Ca$^{2+}$ (RPMI medium) ± serum (FBS). Mean ± s.d. (*n* = 3) plotted. **h** Tukey's box plot of *LEADR/MIR205HG* expression value in frankly basal, luminal progenitor (CD38$^{low}$) and fully differentiated (CD38$^{high}$) luminal cells sorted from human prostate (GSE89050). Jonckheere–Terpstra test evidences a significant decreasing trend along basal–luminal transition. ***p* < 0.01; *****p* < 0.0001. Source data are provided as a Source Data file, together with n of all experiments

confidence transcripts those that had TSS confirmed by CAGE experiments from ENCODE/RIKEN in prostate cells and were supported by the results of a recent genome-wide high-resolution remapping of pri-miRNAs[14] and/or by Gencode v28lift37 Basic annotation (Supplementary Fig. 3e). This led to shortlist 9 different *LEADR* transcripts, characterized by the alternative assembly of 4 modules: exon-1/2 (present in all transcripts with or without retention of the intron); exon-3 (present in all transcripts in short or long version); exon-4 (missing in some transcripts, including the historical RefSeq); 2 alternative terminal exons, the canonical *miR-205* incompatible exon-5.1 and the downstream exon-5.2 (Fig. 2a). Use of the latter would be compatible with *miR-205* excision by positioning pre-*miR-205* completely within an intron. Therefore, alternative splicing/transcription termination may dictate the switch between the canonical miRNA-unproductive configuration, which acts as

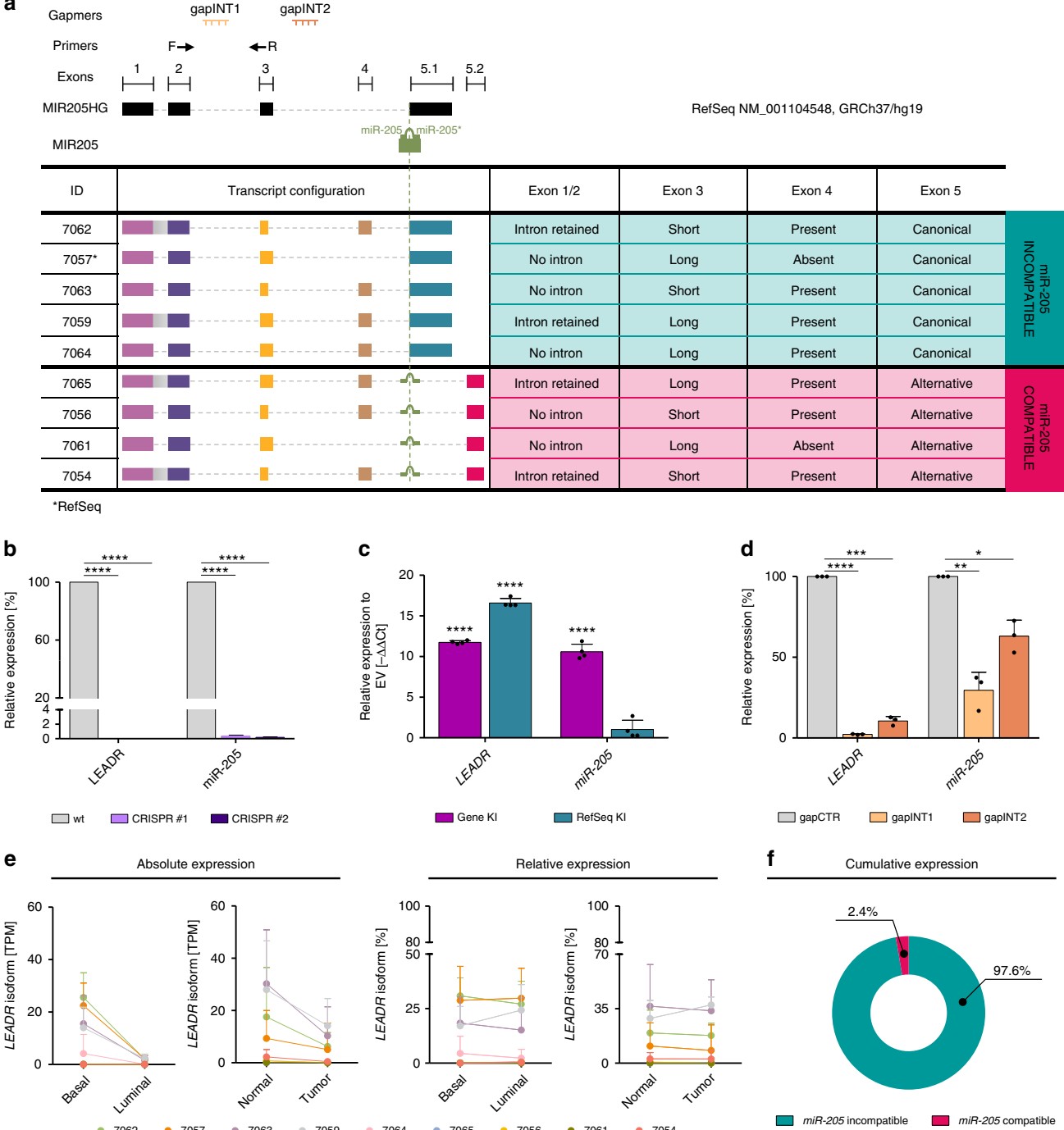

**Fig. 2** *miR-205* compatible and incompatible *LEADR* transcripts. **a** *LEADR/MIR205HG* RefSeq transcript with indication of pre-*miR-205* spanning a splice junction, position of primers (Forward, F, and Reverse, R) generally used for *LEADR* qRT-PCR and target sequences of gapINT1 and gapINT2 in the introns of *LEADR/MIR205HG* primary transcript (*top*). Table reporting bona fide LEADR/MIR205HG alternative transcripts, ranked for expression in prostate basal cells, with indication of exon composition and compatibility with *miR-205* biogenesis (*bottom*). **b** qRT-PCR showing *LEADR* and *miR-205* expression in two independent RWPE-1 cell clones genomically edited for *LEADR* gene using CRISPR/Cas9, as compared to wild-type cells. Mean + s.d. ($n = 3$ qRT-PCR measurements for each clone) plotted. gRNAs used for editing and genotyping of bulk population and single-cell clones are reported in Supplementary Fig. 3d. **c** qRT-PCR showing *LEADR* and *miR-205* expression in DU145 cells knocked-in for either the whole *LEADR* genomic sequence (referred to as "gene") or the RefSeq transcript, as compared to empty vector (EV). Mean + s.d. ($n = 4$) plotted. **d** qRT-PCR showing repression of both *LEADR* and *miR-205* in RWPE-1 cells at day 3 after transfection with two different intronic gapmer oligonucleotides. Mean + s.d. ($n = 3$) plotted. **e** Absolute (*left*) and relative (*right*) expression of *LEADR* transcript isoforms as from RNA-Seq data of basal vs. luminal cells (GSE67070) or of normal vs. tumor tissues (GSE22260). Relative expression calculated as percentage of each isoform respect to the total isoforms. **f** Average cumulative fraction of *miR-205* incompatible and compatible *LEADR* transcripts in prostate cells. *$p < 0.05$; **$p < 0.01$; ***$p < 0.001$; ****$p < 0.0001$ (Student's *t* test). Source data are provided as a Source Data file, together with *n* of all experiments

source of *LEADR*-specific transcripts only, and the alternative one, which acts as *miR-205* primary sequence, though producing additional *LEADR*-specific transcripts as byproducts.

In terms of their absolute expression, we found that all *LEADR* isoforms were more abundant in basal than in luminal cells (Fig. 2e, *left*), in normal than in tumor tissues (Fig. 2e, *left*) and in commercially available normal than in tumor cells (Supplementary Fig. 3f), where expression of all forms approximated to zero. The most abundant transcripts in basal cells were 7062 and 7057 (Fig. 2e), the latter having the same exon composition as the historical NCBI RefSeq (Fig. 2a). No major differences in relative isoform expression were observed among the analyzed samples (Fig. 2e, *right*), nor between the cumulative fraction of *miR-205* incompatible and compatible transcripts, which averagely accounted for 97.6 and 2.4% (Fig. 2f).

**miR-205 biogenesis is Drosha dependent**. To get an insight into *miR-205* production from compatible transcripts, we assessed the role of Drosha. Processing of intronic miRNAs can indeed be independent of the enzyme activity (as is the case of mirtrons, directly excised from host genes by the spliceosome[15]) or dependent on it[16,17]. We found that *miR-205* biogenesis is Drosha-dependent, as Drosha knockdown impaired *miR-205* production, together with that of intergenic *miR-200b/c* and Drosha-dependent intronic *miR-26b*[16], in a way that was proportional to residual enzyme amounts, though not affecting *miR-877* mirtron[18] (Fig. 3a, confirmed by GSE48160 dataset analysis, Supplementary Fig. 3g). Strikingly, Drosha knockdown also reduced the expression of *miR-205* byproducts (lower band in Fig. 3b, measured to assess ex-4/5.2 splicing typical of *miR-205* compatible transcripts), while increasing the expression of a still nonannotated transcript characterized by canonical ex-4/5.1 splicing, retention of partial ex-5.1, and ex-5.2 as terminal exon (upper band, Fig. 3b). These data suggest that Drosha processing precedes the splicing of *miR-205* compatible primary transcripts and that the enzyme occupancy may itself mask the ex-4/5.1 splice site to allow pre-*miR-205* excision. Accordingly, analysis of RNA-Seq data from Drosha and DGCR8 cross-linking immunoprecipitation (CLIP) experiments showed specific peaks covering the ex-4/5.1 splice site (Fig. 3c).

Overall, such results indicate that though an alternative *LEADR/MIR205HG* configuration can actually work as pri-*miR-205*, the majority of *LEADR* processed transcripts are spliced in a way that is incompatible with *miR-205* production, firmly suggesting that the gene is not merely a miRNA host but rather an independent gene entity.

**LEADR is a nuclear long intergenic noncoding RNA**. Though initially defined as a coding gene, *LEADR/MIR205HG* protein product has never been isolated and NP_001098018 has recently been dismissed. Accordingly, analysis of coding potential by Coding Potential Assessment Tool (CPAT)[19] showed that all *LEADR* transcripts are characterized by relatively short open reading frames (ORFs), with none exceeding the cut-off for being considered coding (Supplementary Table 1). Additional inspection revealed *LEADR* as characterized by several features typical of noncoding transcripts, including: (i) low conservation across species (indicative of low-selective pressure on the gene sequence), with human *LEADR/MIR205HG* conserved in primates only (Supplementary Fig. 1); (ii) marked tissue/cell-specificity (Fig. 1)[20]; (iii) no significant matches between the hypothetical aminoacid sequence (read in all the possible translation frames) and known protein domains, as assessed using BLASTP, Pfam, InterPro, SMART, or Blastx tools, and (iv) absence of translation initiation sites, small ORFs or peptides

attributable to *MIR205HG* in PRIDE reprocessing database, according to LNCiPedia repository[21]. In vitro transcription/translation ultimately confirmed that *LEADR* is devoid of protein-coding potential (Fig. 4a) and that it can be referred to as a bona fide lincRNA.

Subcellular fractionation showed that *LEADR* transcript is more abundant in the nucleus than in the cytoplasm of RWPE-1 cells (Fig. 4b), and specifically in the chromatin fraction than in the nucleoplasm (Fig. 4c), indicating that *LEADR* behaves as a nuclear chromatin-associated lincRNA in prostate basal cells.

**LEADR regulates basal–luminal differentiation**. The peculiar expression pattern of *LEADR* prompted us to investigate whether *LEADR* might play a role in epithelial cell differentiation. To this purpose, a loss-of-function approach was pursued in normal immortalized (RWPE-1) and primary prostate basal (PrEC) cells using either a siRNA (siLEADR) or a gapmer antisense oligonucleotide (gapLEADR), each designed to target ex-2/3 junction (shared by all processed transcripts) in an attempt to abrogate the sole expression of *LEADR*. The two molecules markedly reduced *LEADR* levels in both cell types, without affecting the expression of either precursor or mature *miR-205* (Supplementary Fig. 4a). Because gapmers function also in the nucleus[22], gapLEADR was more effective than siLEADR in suppressing nuclear *LEADR* (Supplementary Fig. 4b).

*LEADR* knocked-down RWPE-1 cells shifted from a small, rounded shape to a bigger, elongated columnar phenotype, a transition resembling basal–luminal differentiation (Fig. 5a). Accordingly, cells underwent the typical cytokeratin switch[23], with decrease of basal (KRT5/KRT14) and increase of luminal cytokeratins (KRT8/KRT18) (Fig. 5b, c), and p63 nucleus-to-cytoplasm redistribution (Fig. 5d, Supplementary Fig. 4e)[24]. These findings were recapitulated in PrEC primary basal cells (Fig. 5a, c), where *LEADR* silencing also increased overall androgen receptor (AR) expression at both mRNA (Supplementary Fig. 4f) and protein level (Fig. 5e, f). Moreover, *LEADR* knockdown enhanced AR nuclear translocation under dihydrotestosterone (DHT) stimulation (Fig. 5f), which resulted in increased prostate-specific antigen (PSA) secretion, a feature of terminally differentiated luminal cells (Fig. 5g).

Genome-wide evidence that *LEADR* silencing induces luminal differentiation was obtained by challenging basal- and luminal-specific gene sets derived from gene expression data of frankly luminal and basal prostate cells on the transcriptome of *LEADR*-knocked down RWPE-1 cells. We found a tendency toward positive enrichment of luminal and negative enrichment of basal gene sets, again confirming luminal differentiation (Fig. 5h). Notably, knockdown of *miR-205* only (GSE29782[2]) did not induce any obvious switch to luminal phenotype, whereas simultaneous inhibition of *LEADR* and *miR-205* (obtained through gapINT1) fully recapitulated the transcriptome of overt luminal cells (Fig. 5h). Taken together, these data suggest that *LEADR* exerts functions independently from *miR-205*, being able per se to negatively regulate basal–luminal differentiation, but complements miRNA activity to sustain basal features. Ultimate evidence of *LEADR* repressive role on differentiation came from the assessment of *LEADR*-engineered cell propensity to undergo cytokeratin switch upon culturing in media with increasing differentiative potential (i.e., serum gradient). The analysis showed that, compared with wild type RWPE-1 cells, CRISPRed cells were markedly more prone to luminal transition, whereas ectopically overexpressing cells were more refractory (Fig. 5i).

**LEADR controls genes in the interferon signaling pathway**. Upon *LEADR* silencing, about 64% of the genes resulted up and

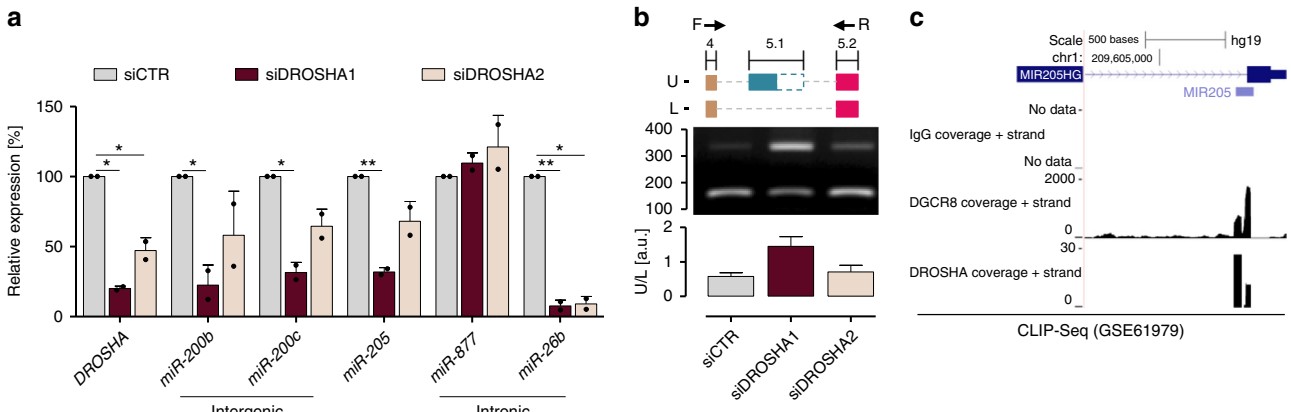

**Fig. 3** *miR-205* biogenesis is Drosha-dependent. **a** qRT-PCR showing expression levels of *DROSHA* mRNA, mature *miR-205* and other intergenic/intronic miRNAs upon Drosha silencing in RWPE-1 cells by two siRNAs. Mean + s.d. ($n = 2$) plotted. **b** End-point RT-PCR showing differential splicing of terminal exons of *LEADR* primary transcript upon Drosha silencing (*top*). Used primers (F on ex-4 and R on ex-5.2) are depicted. Ratio between upper (U) and lower (L) band reported in the bar plot (*bottom*). **c** RNA-Seq peaks of Drosha and DGCR8 occupancy on *LEADR/MIR205HG* primary transcript, as from CLIP experiments (GSE61979). IgG peaks reported as negative control. For comparison, CLIP data for *miR-26b* and *miR-877* are shown in Supplementary Fig. 3h. *$p < 0.05$; **$p < 0.01$ (Student's $t$ test). Source data are provided as a Source Data file, together with $n$ of all experiments

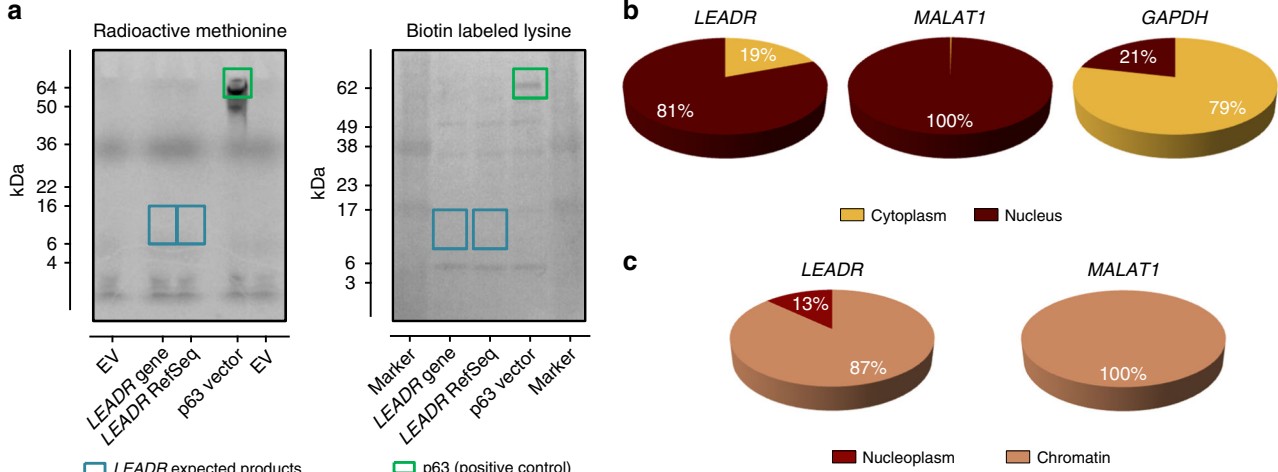

**Fig. 4** *LEADR* is a nuclear chromatin-associated long intergenic noncoding RNA. **a** Polyacrylamide gel electrophoresis of in vitro transcription/translation assay conducted using alternatively [$^{35}$S]-methionine (*left*) or biotinylated lysine (*right*) as detection systems. Vectors containing either the whole genomic sequence ("gene") or RefSeq transcript were used to assess protein production by *LEADR*. No protein products at the predicted molecular weight between 5 and 16 kDa (as indicated by the turquoise rectangle) were observed. Vector expressing p63 was used as positive control (in vitro translated p63 protein indicated by the green rectangle). **b** Pie charts of relative abundance of *LEADR*, *MALAT1* (nuclear control), and *GAPDH* (cytoplasmic control) transcripts in the nuclear and cytoplasmic fraction of RWPE-1 cells, as assessed by qRT-PCR. Mean percentage from three independent experiments is plotted. **c** Pie charts of subnuclear distribution (chromatin or nucleoplasm) of *LEADR* and *MALAT1* transcripts in RWPE-1 cells. Mean percentage from three independent experiments is plotted. Source data are provided as a Source Data file, together with $n$ of all experiments

36% downregulated in RWPE-1 cells, suggesting that lincRNA has a prevalently repressive function (Fig. 6a). These genes were distributed across the genome, with no evidence of regulation *in cis* of genes from chr1q32. As a complementary approach, we ectopically expressed the RefSeq or 7063 (as a prototype of ex-4-containing isoforms) transcript in DU145 cells. Gene set enrichment analysis revealed that the gene sets most affected by *LEADR* modulation in either direction were surprisingly related to inflammation (Fig. 6b). Interferon-related pathways showed the highest normalized enrichment score in each experiment, including overexpression of the whole gene or its silencing by intronic gapINT1 (Fig. 6b). Typically, interferon response is activated as a defense mechanism against double-stranded

RNAs[25]. In this context, it seems instead intimately associated with *LEADR*-regulated program, because it invariably occurs with either single- or double-stranded antisense oligomers, shows an opposite trend of regulation when the lincRNA is ectopically induced (Fig. 6b) and is not appreciably affected by silencing of *miR-205* only (Fig. 6c). The analysis of gene-expression data of true basal and luminal prostate cells (Fig. 6d, *left*) from different datasets revealed that enrichment of interferon genes is a recurrent feature of luminal cells, suggesting that activation of the pathway may invariably accompany differentiation (Fig. 6d, *right*). Accordingly, treatment of basal cells with interferon-β1 induced luminal differentiation, as evidenced by morphological changes (Fig. 6e) and cytokeratin switch (Fig. 6f, g). In PrEC cells

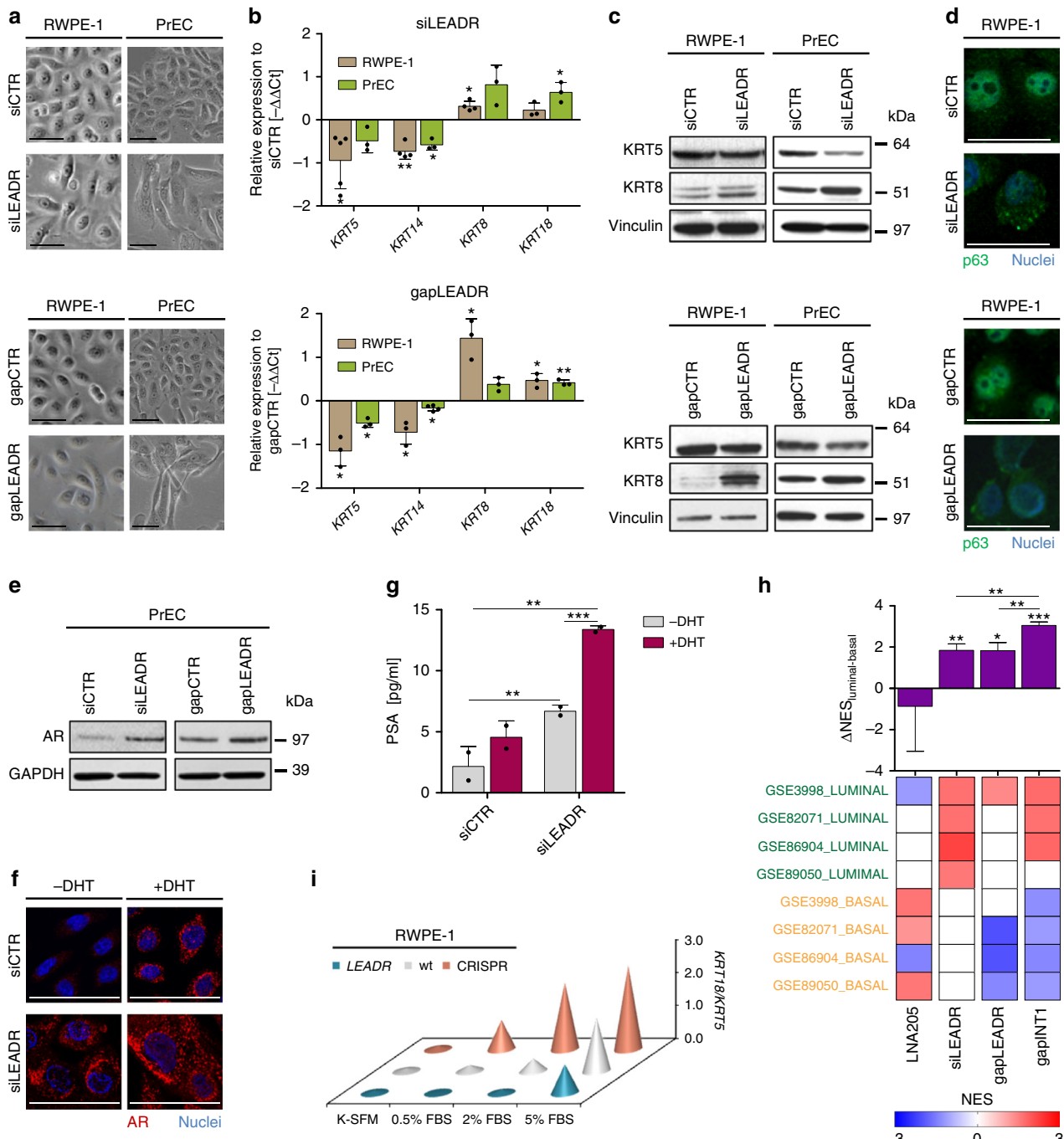

**Fig. 5** *LEADR* regulates basal–luminal differentiation. **a** Bright-field images showing morphological changes occurring in RWPE-1 (*left*) and PrEC (*right*) cells (day 3) upon *LEADR* silencing by siLEADR (*top*) or gapLEADR (*bottom*). Scale bar, 50 µm. Full-size images reported in Supplementary Fig. 4c. **b** qRT-PCR showing changes in basal and luminal cytokeratins in RWPE-1 or PrEC cells upon *LEADR* silencing by siLEADR (*top*) or gapLEADR (*bottom*). Mean + s.d. plotted. **c** Western blots showing changes in basal/luminal cytokeratins in RWPE-1 (*left*) or PrEC (*right*) cells upon *LEADR* silencing by siLEADR (*top*) or gapLEADR (*bottom*). Vinculin used as loading control. **d** Immunofluorescence showing cytoplasmic re-localization of p63 (green) upon *LEADR* silencing in RWPE-1 cells by siLEADR (*top*) or gapLEADR (*bottom*). Nuclei counterstained with DAPI (blue). Scale bar, 50 µm. Full-size images are reported in Supplementary Fig. 4d. **e** Western blots showing changes in Androgen Receptor (AR) expression in PrEC cells upon *LEADR* silencing by siLEADR (*left*) or gapLEADR (*right*). GAPDH used as loading control. **f** Immunofluorescence showing AR (red) expression in PrEC cells upon *LEADR* silencing by siLEADR, in the presence or absence of simultaneous DHT stimulation. Nuclei counterstained with DAPI (blue). Scale bar, 50 µm. Full-size images reported in Supplementary Fig. 4g. **g** ELISA-based quantification of PSA in the conditioned media of PrEC cells silenced for *LEADR* expression by siLEADR (±DHT). Mean + s.d. ($n = 2$) plotted. **h** Heatmap (*bottom*) of the normalized enrichment score (NES) for luminal and basal custom gene sets (obtained from gene-expression data of frankly basal and luminal prostate cells from the indicated datasets) in RWPE-1 cells upon knockdown of *LEADR* and/or *miR-205* (the latter abrogated using an antisense LNA-modified oligomer, LNA205). Bar plot of differential luminal/basal NES is reported (*top*). Mean + s.d. ($n = 4$) plotted. **i** Plot showing propensity of wild type (wt), *LEADR*-overexpressing and CRISPRed RWPE-1 cells to differentiate toward luminal phenotype (measured by qRT-PCR and expressed as *KRT18/KRT5* ratio) upon culturing in media with increasing differentiative potential ($n = 3$). *$p < 0.05$; **$p < 0.01$; ***$p < 0.001$ (Student's *t* test). Source data are provided as a Source Data file, together with *n* of all experiments

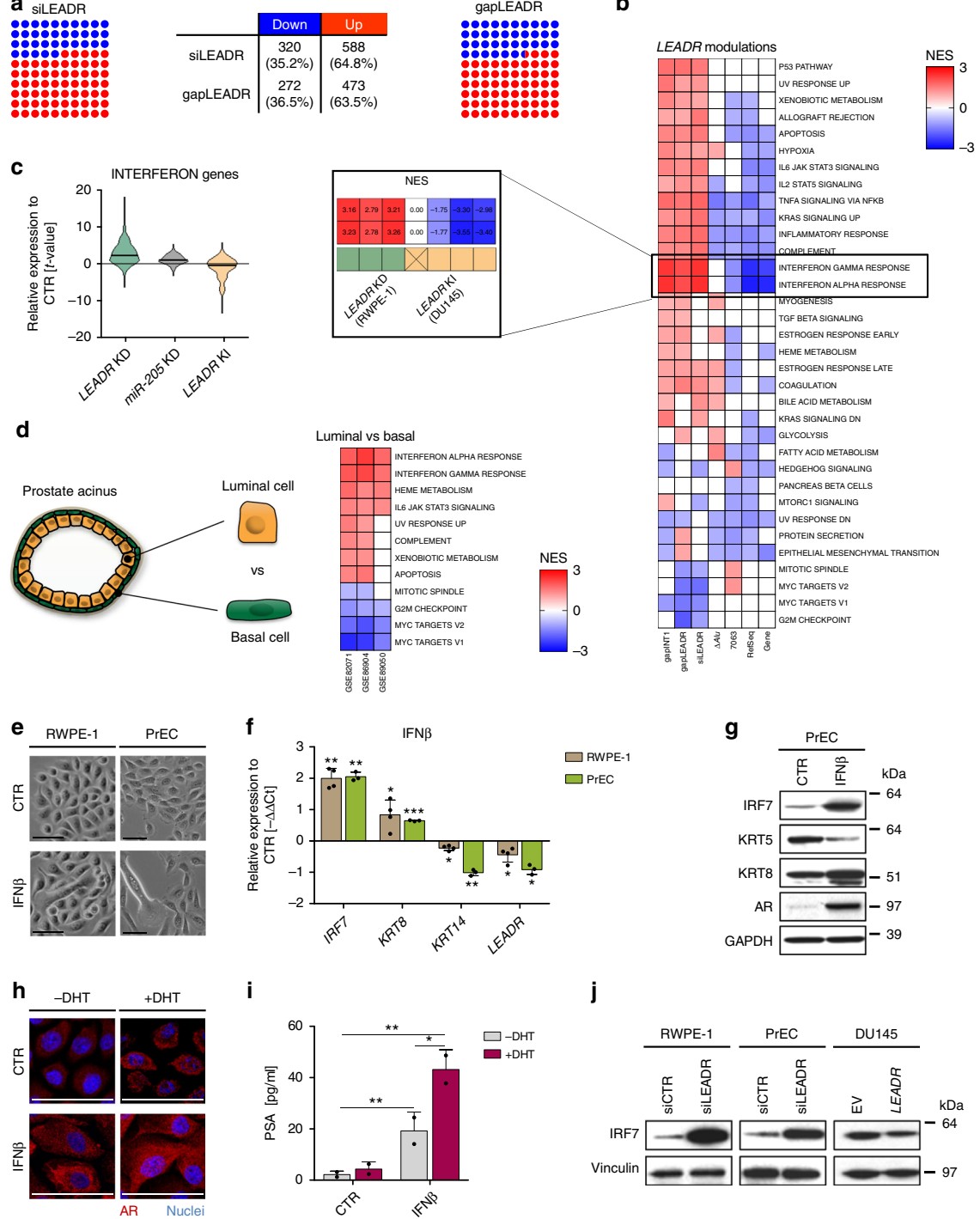

it also increased AR expression (Fig. 6g, h), especially upon concomitant DHT stimulation, with cells adopting a frank luminal phenotype characterized by enhanced AR nuclear translocation (Fig. 6h) and consequent PSA secretion (Fig. 6i).

Notably, interferon stimulation also significantly reduced *LEADR* levels and increased those of *IRF7* (Fig. 6f), one of the genes most affected by *LEADR* manipulation both in basal and tumor cells, according to microarray and western blotting analysis shown in Fig. 6j. *IRF7* was found to be heavily induced, and *LEADR* was repressed, also in PrEC cells led to differentiate by serum stimulation (Supplementary Fig. 5c), again providing additional indirect evidence of interferon involvement in luminal differentiation. A role for inflammatory cytokines in

differentiation has been widely reported in cancer, where it is mainly related to the function of tumor-infiltrating immune cells[26], but only a few reports confirmed the immune-independent role of interferon signaling in normal differentiation[27–30]. In this regard, our data indicate that *LEADR* regulates the expression of interferon signaling genes, which appear to have an immune-independent role in the differentiation of normal prostate epithelial cells.

**LEADR binds to promoters having *Alu* and IRF binding site.** LincRNAs can regulate gene expression by multiple mechanisms, mostly depending on subcellular localization and the nature of molecular interactors (DNA, RNA, and proteins)[31–33]. Because

**Fig. 6** *LEADR* controls genes in the interferon signaling pathway. **a** Dot plots and table indicating the number and percentage of up and downmodulated genes in RWPE-1 cells upon *LEADR* silencing. Each dot represents 1% of genes. **b** Heatmap of NES values of Hallmark v.5.2 of MSigDB gene sets in RWPE-1 cells knocked-down (KD) or in DU145 cells knocked-in (KI) for *LEADR*, using the different approaches described in the paper. Only gene sets with significant and coherent enrichment in at least two of either *LEADR* KD or KI conditions are shown. NES values of Interferon Pathway gene sets in all *LEADR* modulation experiments are reported in the magnification. **c** Violin plot of relative interferon gene expression in cells perturbed for *LEADR* or *miR-205*, compared to respective controls. **d** Heatmap (right) of NES values of Hallmark v.5.2 signatures in publicly available datasets of frankly basal or luminal cells isolated from human prostate acini, as illustrated in the cartoon (left). Only gene sets with significant and coherent enrichment in at least two datasets are shown. **e** Bright-field images showing the morphological changes occurring in RWPE-1 (left) and PrEC (right) cells upon interferon-β1 treatment. Scale bar, 50 μm. Full-size images reported in Supplementary Fig. 5a. **f** qRT-PCR showing changes in IRF7, basal/luminal cytokeratins and *LEADR* in RWPE-1 ($n = 4$) and PrEC ($n = 3$) cells upon interferon-β1 treatment. Mean + s.d. plotted. **g** Western blot showing changes in IRF7, basal/luminal cytokeratins and AR in PrEC cells upon interferon-β1 treatment. GAPDH used as loading control. **h** Immunofluorescence showing AR (red) expression in PrEC cells upon interferon-β1 treatment, in the presence or absence of simultaneous DHT stimulation. Nuclei counterstained with DAPI (blue). Scale bar, 50 μm. Full-size images reported in Supplementary Fig. 5b. **i** ELISA-based quantification of PSA in the conditioned media of PrEC cells upon interferon-β1 treatment (±DHT). Mean + s.d. ($n = 2$) plotted. **j** Western blots showing changes in IRF7 protein abundance in RWPE-1/PrEC or DU145 cells upon *LEADR* KD or KI, respectively. Vinculin used as loading control. $*p < 0.05$; $**p < 0.01$; $***p < 0.001$ (Student's *t* test). Source data are provided as a Source Data file, together with n of all experiments

*LEADR* possesses a chromatin-specific localization, it may regulate gene expression through direct DNA interaction. In search for mechanistic insights, we focused on bona fide *LEADR* targets, selected as the genes found upregulated upon *LEADR* silencing and, correspondingly, downregulated upon overexpression (referred to as "*LEADR*-core up", Fig. 7a, Supplementary Data 1). Quantitative reverse transcription polymerase chain reaction (qRT-PCR) validation in Supplementary Fig. 6a). De novo motif discovery revealed significant enrichment of five motifs (Fig. 7b) in promoters of such genes, all of which homologous to a 300-nt region spanning ex-1/2 boundary of *LEADR* transcript (Fig. 7b, Supplementary Data 2), suggesting that the lincRNA may use such sequence to physically interact with the DNA. This region is defined as a short-interspersed nuclear element (SINE) of the *AluJb* family (Supplementary Fig. 1). The motifs showing the highest enrichment in promoters and/or homology with *LEADR* were motif-1 and -4 (Fig. 7b and Supplementary Data 2), which proved to be essential for *LEADR* function. When deleted in the portion of *Alu* element containing these motifs (*LEADR*-ΔAlu), *LEADR* failed to regulate gene expression (Fig. 6b), especially of "*LEADR*-core up" (Fig. 7c) and interferon genes (Fig. 6b and Supplementary Fig. 6b), in DU145 cells. Unlike in other lincRNAs[34], *Alu* depletion did not alter *LEADR* subcellular localization, because the transcript invariably accumulated in the nucleus of DU145 cells upon ectopic replacement, irrespective of *Alu* presence (Fig. 7d), thus suggesting the existence of a different intrinsic nuclear retention signal.

In view of the repetitive nature and abundance of *Alu* sequences across the genome, it is reasonable to assume that additional features are present in *LEADR*-regulated genes to direct selective targeting. When prediction of TF binding sites was run on promoters of "*LEADR*-core up" genes, we found a significant enrichment in the consensus sequences of several IRF family members (Supplementary Data 3). Accordingly, 76% of them were characterized by the presence of at least one validated IRF peak according to chromatin immunoprecipitation (ChIP)-Seq data (GSE32465 and GSE31477), which is a higher fraction compared to that found in the random genome (57%, hypergeometric test $p = 0.01$). Strikingly, 55% of promoters showed co-occurrence of IRF site and *Alu* sequence in tandem (vs. 38% genome-wide, $p = 0.02$), with no substantial change in the fraction of IRF-only promoters and a marked reduction of *Alu*-only promoters in favor of *Alu*/IRF combination (Fig. 8a). In contrast, promoters of genes downmodulated by *LEADR* silencing ("*LEADR*-gene set dn" or "*LEADR*-core dn", Fig. 7a, Supplementary Data 1) were neither enriched nor depleted of

either *Alu*/IRF or *Alu*-only sites (Supplementary Fig. 6c), suggesting such genes as possible indirect targets.

When the analysis of the *Alu*/IRF elements was extended to the 136 genes commonly upregulated by siLEADR and gapLEADR ("*LEADR*-gene set up", Fig. 7a and Supplementary Data 1) ranked by average fold-induction from microarray data, results showed that presence of *Alu*, and specifically of *Alu* + IRF combination, was the highest in the top-20 modulated genes (90% and 65% of promoters, respectively) and decreased proportionally with the fold-change (Fig. 8b). This finding indicates that *Alu*/IRF co-occurrence is a distinctive trait of *LEADR* targets and that its presence in a given promoter tightly correlates with the probability of that gene being modulated by the lincRNA. Moreover, these data indicate that genes upmodulated upon *LEADR* silencing probably comprise both direct (i.e., characterized by the presence of *Alu* + IRF site) and indirect targets. Based on this evidence, we identified the fraction (51 of 136) of "*LEADR*-gene set up" genes showing co-occurrence of *Alu* and IRF sites as the bona fide "*LEADR*-signature" (Fig. 8c and Supplementary Data 1).

Regarding the topological relationship between the two elements, in most of the cases, the IRF site was more proximal to the TSS than the *Alu* (median distance: 88 vs. 1095 bp), which was located upstream of TSS in the transcription direction in 96% of cases and upstream of the IRF site in 90% of cases (Fig. 8c). The reciprocal distance did not correlate with the intensity of regulation by *LEADR* (Supplementary Fig. 6d), indicating that a certain degree of tolerability exists, provided that *Alu*/IRF spacing falls within the range between 66 and 1973 bp (*IRF7* and *HIST1H1C*, respectively).

We used chromatin isolation by RNA precipitation (ChIRP) to determine whether *LEADR* could physically interact with promoter regions of direct targets in basal cells under physiological/nonoverexpressed conditions. We found that bio-tinylated *LEADR* probes efficiently pulled-down *LEADR* RNA, but not *GAPDH* mRNA nor the nuclear lincRNA *MALAT1*, whereas *lacZ* probes failed to precipitate any human transcript (Fig. 8d). Analysis of *LEADR*-cross-linked DNA revealed that all of the tested *LEADR*-signature genes (i.e., *Alu* + IRF) were indeed bound by the lincRNA in RWPE-1 cells, whereas no significant binding was observed to promoters having different site combinations (Fig. 8e). Interestingly, ChIRP analysis performed on DU145 cells overexpressing the wt or *Alu*-deleted lincRNA showed that deletion of motifs 1–4 of *Alu* element impaired at least in part *LEADR* ability to bind to its targets (Supplementary Fig. 6e, f). These data suggest that *LEADR* is able to directly

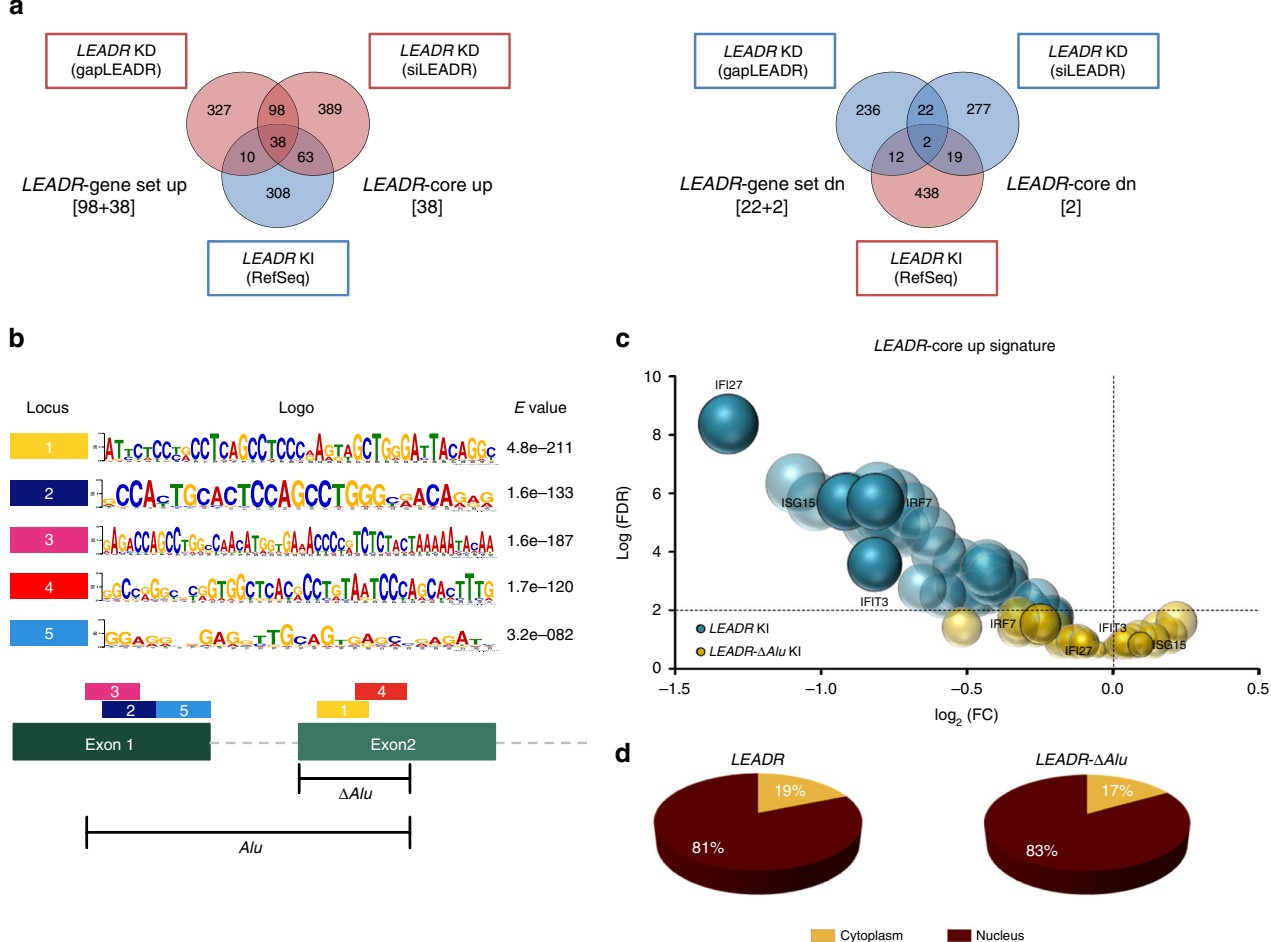

**Fig. 7** *Alu* element in *LEADR* transcript is essential for its function. **a** Venn diagram of "*LEADR*-gene set up" signature—i.e., 136 genes upregulated in RWPE-1 cells upon *LEADR* silencing by two independent approaches (*left*). The subset of 38 genes resulting coherently down-regulated in DU145 cells ectopically overexpressing *LEADR* is indicated as "*LEADR*-core up". Venn diagram of the "*LEADR*-gene set dn" signature, composed of the 24 downregulated genes in RWPE-1 cells upon *LEADR* silencing by two independent approaches (*right*). The subset of two genes resulting coherently upregulated in DU145 cells ectopically overexpressing *LEADR* is indicated as "*LEADR*-core dn". **b** Recurrent motifs (and relative *E* values) in promoters of "*LEADR*-core up" genes, as from de novo motif discovery (*top*). Alignment of motifs with *LEADR* transcript is depicted, as well as full *Alu*/SINE sequence and portion deleted in *LEADR*-Δ*Alu* construct (*bottom*). **c** Bubble plot of "*LEADR*-core up" genes in DU145 cells knocked-in for the wild type (*LEADR* KI) or *Alu*-deleted (*LEADR*-Δ*Alu* KI) *LEADR* RefSeq transcript. **d** Pie charts of relative abundance of wild type and *Alu*-deleted *LEADR* in the nuclear and cytoplasmic fraction of DU145 cells upon ectopic replacement, as assessed by qRT-PCR. Mean percentage from three and two independent experiments, respectively, is plotted. Source data are provided as a Source Data file, together with *n* of all experiments

interact with the promoters of target genes, possibly through *Alu*/*Alu* pairing, and confirm that the IRF site proximity is crucial to secure binding and/or allow regulation.

**LEADR modulates IRF1 occupancy.** We found that *LEADR* physically interacts with IRF1 protein (less so with IRF7) in the nucleus, as assessed by RNA immunoprecipitation performed either in native conditions (Fig. 9a) or upon ultraviolet (UV) cross-linking (Fig. 9b), with the latter method indicating that the interaction is direct. *LEADR*/IRF1 protein interaction was also confirmed by effective pulldown of IRF1 by in vitro transcribed biotinylated *LEADR* RNA (Fig. 9c). ChIP showed that IRF1 occupancy on *LEADR* target gene promoters is negligible in RWPE-1 cells (endogenously expressing high *LEADR* levels) under basal conditions but is increased upon *LEADR* silencing (Fig. 9d). As expected, *LEADR* ectopic overexpression in DU145 cells reduced IRF1 binding, especially to promoters of genes having *Alu*/IRF site combination (Fig. 9e). Altogether these data led to speculate a model where *LEADR* may somehow titrate IRF1

away from its binding site, probably by interacting with the *Alu* element in the DNA and IRF1 protein.

## Discussion

Dysregulation of miRNA function resulting in aberrant protein expression in the targeted pathways is causatively associated with numerous diseases, including cancer[35]. Intensive research on the different routes of miRNA biogenesis has identified canonical and noncanonical pathways for transcription, processing and maturation[36]. Although miRNAs can be located in intergenic regions and transcribed as independent transcription units, the majority of human miRNA loci reside within the introns of the so-called host genes[37], with which they are generally co-transcribed[38]. When miRNAs are located in the introns of non-coding transcripts, the question arises as to whether host genes should be viewed merely as pri-miRNAs or rather as genes with independent, still unknown functions.

Here we report, for the first time, that human *MIR205HG*, which we term *LEADeR* (*LEADR*), is a nuclear lincRNA

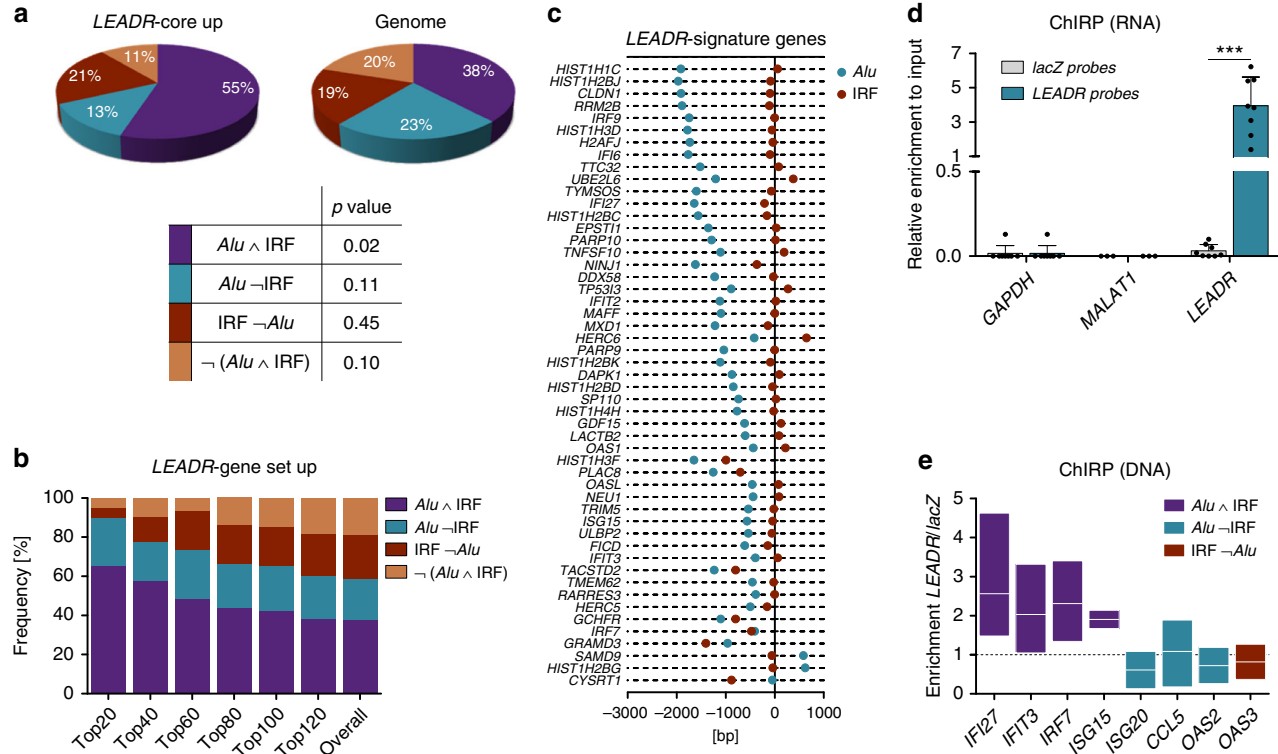

**Fig. 8** *LEADR* physically binds to promoters of genes having *Alu* and IRF binding site motif in tandem. **a** Pie charts of the frequency of *Alu*/IRF sites in "*LEADR*-core up" genes and in genome (*top*). Table reporting hypergeometric test *p* value for each category is also depicted (*bottom*). **b** Bar plot of cumulative frequency of *Alu*/IRF sites in the promoters of "*LEADR*-gene set up" genes, ranked by mean of the FCs in *LEADR* KD experiments. **c** Reciprocal positioning of *Alu*/IRF sites in promoters of "*LEADR*-signature" genes. Genes depicted in the transcription direction, with TSS set to zero. **d** qRT-PCR showing RNA retrieval upon ChIRP performed in RWPE-1 cells using probes complementary to *LEADR* or to *lacZ*. Mean + s.d. plotted. **e** ChIRP-qPCR detection of *LEADR* occupancy on representative target loci (normalized to unrelated genomic region) in RWPE-1 cells. Line at mean plotted (*n* = 3). Source data are provided as a Source Data file, together with *n* of all experiments

functioning independently from the hosted miRNA. Although both RNAs are enriched in the basal layers of epithelia, they nonetheless govern different, non redundant, and essential features of human prostate basal cells: *miR-205* regulates the production of the basement membrane[2], whereas the lincRNA preserves basal identity by regulating differentiation (Fig. 10a).

Supporting this view is the observation that only lowly abundant alternative *LEADR/MIR205HG* transcripts are compatible with *miR-205* production (Fig. 2). Only locus-targeted RNA-Seq[39] in expressing cells would allow to precisely define the structure and abundance of *LEADR* transcripts as well as to detect unstable intermediates of miRNA/lincRNA processing, but the above observation indicates that *LEADR/MIR205HG* exceeds its role as host gene. *miR-205* biogenesis indeed becomes possible only upon selection of a downstream terminal exon, which allows the Drosha-dependent processing of pre-*miR-205* as fully intronic (Fig. 10a). A previous study reported the example of an intron-exon spanning pre-miRNA where either the miRNA or the hosting mRNA can be produced depending on the prevalence of Drosha-mediated or splicing-dependent activity[40]. What is intriguing about *LEADR/MIR205HG* locus is, however, that (i) Drosha activity is not completely uncoupled from splicing, and actually influences splicing of terminal exons in *LEADR* primary transcript, and (ii) the resulting byproducts may apparently retain functionality as *LEADR*, because they contain whole exons from 1 to 4 (Fig. 2a), including the *Alu* sequence. It is known that lincRNAs can promiscuously use alternatively spliced forms, as their function relies prevalently on their modular architecture rather than their sequence[41]. This observation gains support from

the evidence that DU145 cells overexpressing the RefSeq or 7063 transcript display nearly superimposable transcriptomes (Fig. 6b).

The study of lincRNAs is still at a preliminary stage[33,42] because of the poor sequence conservation among even closely related organisms and lack of shared biological features. Our study documents the noncoding nature of *LEADR* and, for the first time, its function as lincRNA able to regulate basal–luminal differentiation through repression of the interferon pathway. Examples of lincRNAs able to modulate interferon genes have been reported, though only in the context of antiviral response[43,44]. Our data suggest an interesting link between innate immune response and the normal epithelial differentiation program, which may be independent of immunity-related functions. This noncanonical role of interferons has been only in part investigated in the skin[27,28] and observed in the mammary gland[29,30].

Mechanistically, we showed that *LEADR* regulates gene expression *in trans* through a direct DNA interaction that presumably involves the *Alu* element in ex-2 of the lincRNA and *Alu* sequences located in the promoters of target genes (Fig. 10a). The hypothesis that exonic *Alu*s may represent essential functional domains of lincRNAs[45] is supported by their higher frequency in exons of lincRNAs than in those of protein-coding genes[46], as well as by the evidence that deletion of *Alu* elements abrogates or even reverses gene-expression changes induced by a number of lincRNAs, such as *APTR*[47], *ANRIL*[48], and *LEADR*.

Notably, proof of direct *Alu*$_{DNA}$/*Alu*$_{RNA}$ interaction has not yet been documented for *Alu*-containing lincRNAs, because this would require sophisticated structural insights. More importantly,

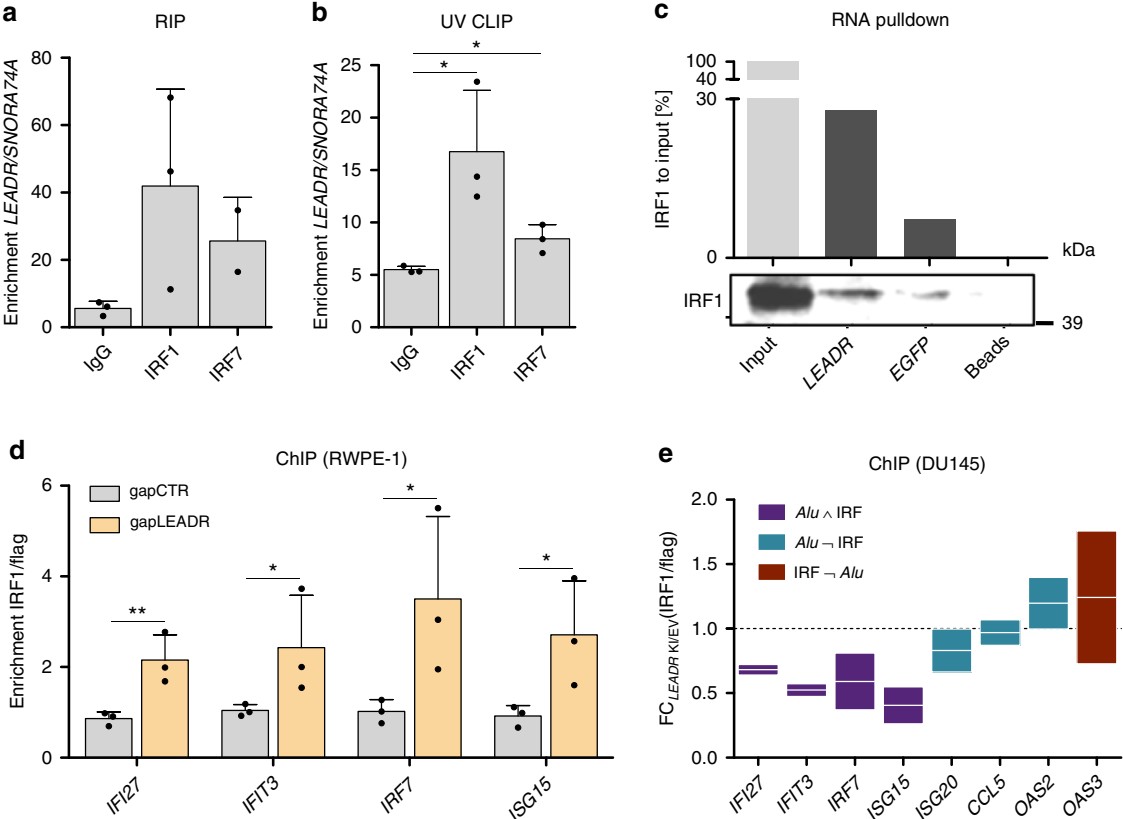

**Fig. 9** *LEADR* interacts with IRF1 protein and modulates its occupancy on promoters. **a** qRT-PCR reporting enrichment of *LEADR* transcript (normalized to *SNORA74A*) in immunoprecipitants from RIP experiments conducted using the indicated antibodies. Mean + s.d. plotted. **b** qRT-PCR reporting enrichment of *LEADR* transcript (normalized to *SNORA74A*) in immunoprecipitants from UV-CLIP experiments conducted using the indicated antibodies. Mean + s.d. ($n = 3$) plotted. **c** Western blot showing enrichment of IRF1 protein (as quantified in the above plot) upon RNA pulldown performed using biotinylated in vitro transcribed *LEADR*. Biotinylated in vitro transcribed *EGFP* and beads only were used as controls. Representative blot out of three independent experiments. **d** ChIP-qPCR detection of IRF1 occupancy on representative target loci upon *LEADR* silencing in RWPE-1 cells, expressed as IRF1/flag ratio normalized to unrelated genomic region. Mean + s.d. ($n = 3$) plotted. **e** ChIP-qPCR detection of IRF1 occupancy on representative target loci upon ectopic *LEADR* overexpression in DU145 cells. Data are expressed as fold change (FC) of IRF1/flag ratio normalized to unrelated genomic region in overexpressing cells compared to EV. Line at mean plotted ($n = 3$). *$p < 0.05$, **$p < 0.01$ (Student's *t* test). Source data are provided as a Source Data file, together with *n* of all experiments

no indication on how lincRNAs bind to selected repetitive elements in the genome has been provided so far. Here, we propose a novel mechanism of target selection by which *LEADR* would preferentially regulate genes having *Alu* in proximity of IRF binding site. The distance between the two elements was shown not to be essential for function, suggesting that *LEADR* (and/or chromatin) may adopt different conformations to allow regulation.

Direct *LEADR*/IRF1 protein interaction as well as *LEADR*-induced changes in IRF1 occupancy led to speculate a scenario where, in basal cells, *LEADR* directly binds the promoters of its targets and disables transcription by simultaneously interacting with IRF1 (e.g., poised state), whereas its loss would release the brake on interferon gene expression and subsequent luminal differentiation (Fig. 10a). Mechanistically, *LEADR* function appears neither associated with recruitment of repressive chromatin modifiers nor with H3K27 trimethylation, as instead reported for *APTR* and *ANRIL*[47,48]. Indeed, the analysis of GSE63094 ChIP-Seq data revealed that in RWPE-1 cells the promoters of "*LEADR*-signature" genes are characterized by active chromatin histone modifications (Supplementary Fig. 6g), thus suggesting that *LEADR* repressive effect may be rather associated with its ability to prevent IRF1 binding (e.g., decoy

mechanism) and/or buffer its activity (e.g., through steric hindrance or recruitment of additional co-factors). Further investigation is required to fully understand the molecular nature of *LEADR* binding to *Alu* elements (e.g., DNA/RNA pairing), the exact mechanism by which the lincRNA interferes with IRF1 transcriptional activity, and the dynamics of reciprocal *LEADR*/IRF1/target gene interaction. Moreover, the observation that several "*LEADR*-signature genes" are histone and histone variants (Fig. 8c) prompts the hypothesis of an additional layer of *LEADR*-mediated control for gene expression, consisting of genome-wide transcriptional reprogramming through histone modulation.

The concept of a regulation model that does not involve epigenetic silencing but is rather based on dynamic changes in the *LEADR*/IRF1/target gene interactome underscores how important it would be for basal cells to exploit the *LEADR*/interferon axis to rapidly respond to external differentiation stimuli in a way remindful of innate antiviral response. In this regard, *LEADR* target genes proved to be biologically relevant in a number of contexts. "*LEADR*-signature" genes were confirmed to be among the most enriched in prostate luminal cells in the human setting, where *LEADR* expression is low, as well as in mice (Fig. 10b), where *LEADR* sequence is not conserved (Supplementary Fig. 1). A broad relevance of the differentiation program regulated by

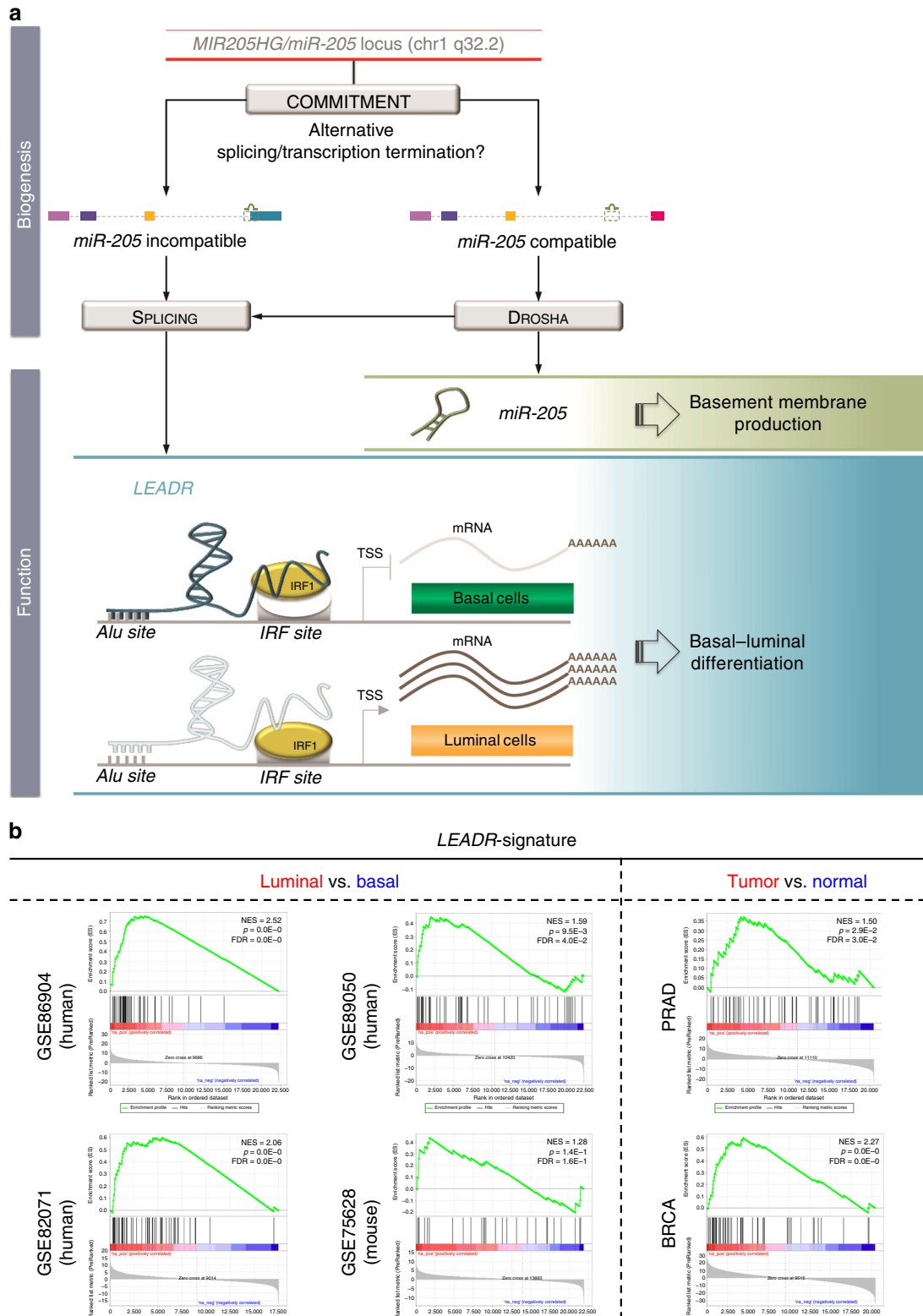

**Fig. 10** Proposed model of *LEADR* function. **a** Schematic representation of biogenesis of *miR-205* and *LEADR*-specific transcripts from *MIR205HG* locus. The hypothetical model of action of *LEADR* as nuclear lincRNA capable of regulating basal–luminal differentiation is depicted. The specific functions of *miR-205* and *LEADR* in basal cells are reported on the right. **b** GSEA of "*LEADR*-signature" genes on GEO datasets of basal/luminal cells (*left*) and in tumor/normal tissues (*right*) from TCGA data (prostate, PRAD, and breast, BRCA, cancers). NES, p, and FDR values are indicated in each plot

*LEADR*, possibly emerged in primates to ensure a more robust mechanism that regulates the control of the pathway, may be consequently envisaged.

"*LEADR*-signature" is also enriched in prostate and breast carcinomas (Fig. 10b), indicating that the aberrant function of *LEADR* or of downstream mediators may play a role in tumorigenesis. Disruption of normal epithelial differentiation pathways has been proposed as a cause of PRAD, based on the evidence that many of the genes commonly altered in prostate neoplasms are also involved in differentiation (e.g., Myc, p38MAPK, Notch, and PI3K/PTEN)[49]. In addition, a role of interferon-stimulated genes in cancer is widely documented[50]. In the context of prostate cancer, it was shown that inflammation can directly promote basal–luminal differentiation and that, when associated with oncogenic aberrations, it can even accelerate PRAD development[51]. As *LEADR* controls basal–luminal transition via repression of the interferon pathway, it is conceivable that defects in its expression/function may result in aberrant differentiation and favor tumorigenesis. Understanding of the *LEADR*-regulated differentiation program would therefore help clarify the mechanisms of tumor initiation in prostate cancer, and could be also translated to other tumor types, such as breast cancer, where basal–luminal transitions have been shown to be relevant hallmarks.

## Methods

**Cell lines and cell-based experiments**. Established human cell lines were purchased from American Type Culture Collection (ATCC, Rockville, MD, USA) and cultured in standard conditions. The normal primary prostate epithelial cells (PrEC) were grown in PrEC basal medium (ATCC; PCS-440-030) supplemented with PrEC growth kit (ATCC; PCS-440-040). The normal immortalized prostate cells (RWPE-1) were maintained in K-SFM (Thermo Fisher Scientific Inc., Waltham, MA, USA) culture medium supplemented with 5 ng/ml epidermal growth factor and 0.05 mg/ml bovine pituitary extract. DU145 prostate cancer cells were cultured in RPMI-1640 (Lonza, Basel, Switzerland) supplemented with 10% FBS (Thermo Fisher Scientific Inc., Waltham, MA, USA). Cell lines were authenticated and periodically monitored by genetic profiling using short tandem repeat analysis (AmpFISTR Identifiler PCR amplification kit, Thermo Fisher Scientific Inc., Waltham, MA, USA). Cells were routinely checked for possible mycoplasma contamination through MycoAlert® Mycoplasma Detection Kit (Lonza, Basel, Switzerland).

To induce luminal differentiation, RWPE-1 cells were grown for 72 h in RPMI-1640 medium (characterized by high-calcium concentration as compared to K-SFM, which is calcium-free to maintain basal features) with or without 2% FBS. Alternatively, cells were grown in their medium supplemented with increasing concentration of FBS. Human recombinant β-interferon (rHuIFN-B, Calbiochem, USA and Canada) was used at the concentration of 1000 IU/ml. Dihydrotestosterone (DHT, Sigma-Aldrich, Saint Louis, MI, USA) was used at the final concentration of 5 nM.

For transfection experiments, cells were seeded at the density of 8000 cells/cm$^2$ in culture vessels. Twenty-four hours later, medium was removed and cells were transfected with the molecule of interest for 4 h, using Optimem medium (Thermo Fisher Scientific Inc., Waltham, MA, USA) and Lipofectamine-2000 reagent (Thermo Fisher Scientific Inc., Waltham, MA, USA) according to the manufacturer's protocol.

Knockdown of *MIR205HG/LEADR* was performed by using alternatively siRNA (used at 30 nM final concentration) or single-stranded LNA/DNA/LNA antisense gapmers (used at 5 nM final concentration). *TP63* expression was modulated using the expression vector and the siRNA reported in ref.[2], whereas *DROSHA* was silenced using two independent siRNAs. All siRNAs were designed by using the online tool siDESIGN (Dharmacon Inc., Chigago, IL, USA) and purchased from Eurofins MWG-Biotech (Ebersberg, Germany). Antisense LNA™ GapmeR were custom designed and purchased from Exiqon (Vedbaek, Denmark). A control siRNA (siCTR) and control gapmer (gapCTR) with no homology to any known human mRNA were used for comparative purposes. The siRNA and gapmer sequences (in sense format) used for this study are listed in Supplementary Table 2.

Overexpression of *MIR205HG/LEADR* was performed by using a pCMV-6AC plasmid vector containing alternatively the whole genomic sequence (from exon 1 to exon 5.2, including all introns), the RefSeq or 7063 transcript, as synthesized by OriGene (Rockville, MD, USA). An empty pCMV-6AC (EV) was included as experimental control. To create stably overexpressing cell lines, cells were transfected as previously described with 2 µg of vector containing Neomycin resistance and, 48 h after transfection, were selected in their culture medium

supplemented with 700 µg/ml Geneticin G418 Sulfate (Thermo Fisher Scientific Inc., Waltham, MA, USA).

Cell morphology was evaluated usually at day 3 after transfection using an Eclipse TE2000-S microscope (Nikon, Japan). Images were acquired by a Digital Camera DXM100F (Nikon, Japan).

**CRISPR/Cas9 genome editing**. To genomically edit *MIR205HG/LEADR*, two specific guide RNAs (gRNAs) flanking the sequence from exon 1 (including the TSS) to exon 3 were designed with http://crispr.mit.edu/ and cloned into the PX459 expression vector obtained from Addgene (#48139—Cambridge, MA, USA), which simultaneously codifies for Cas9 enzyme. The gRNA sequences are listed in Supplementary Table 3.

RWPE-1 cells were transfected simultaneously with 1 µg of each gRNA, then selected with Puromycin (700 µg/ml) for 24 h. Thus, single cell clones were derived and genomic DNA was isolated using PureLink Genomic DNA Mini Kit (Thermo Fisher Scientific Inc, Waltham, MA, USA). Genotyping was performed by PCR using GoTaq Flexi DNA polymerase (Promega, Fitchburg, WI, USA) and the primers listed in Supplementary Table 3. PCR products were cloned into pGEM-T Easy vector system I (Promega, Fitchburg, WI, USA) and subsequently sequenced using T7 sequencing primer.

**RNA isolation and RT**. Total RNA was isolated using QIAzol Lysis Reagent and miRNeasy Mini Kit (QIAGEN, Hilden, Germany) with DNase I digestion (QIAGEN, Hilden, Germany) according to the manufacturer's instructions. RNA integrity was checked through agarose gel electrophoresis. RNA yield and A260/280 ratio were monitored with a NanoDropND-2000c spectrophotometer (Thermo Fisher Scientific Inc., Waltham, MA, USA). For gene-expression profiling studies, RNA integrity numbers were also measured using 2100 Bioanalyzer (Agilent Technologies, Santa Clara, CA, USA). Normal prostate tissue RNA used as calibrator for qRT-PCR analysis was purchased from Thermo Fisher Scientific (Waltham, MA, USA).

cDNA was synthesized using high-capacity cDNA Reverse Transcription Kit with random primers (Thermo Fisher Scientific Inc., Waltham, MA, USA) for total RNA or oligo-dT$_{16}$ primers (Thermo Fisher Scientific Inc., Waltham, MA, USA) for nuclear and cytoplasmic fraction. For miRNA detection, TaqMan MicroRNA Reverse Transcription Kit and sequence specific primers (Thermo Fisher Scientific Inc., Waltham, MA, USA), listed in Supplementary Table 4, were used on total RNA.

**Nucleus–cytoplasm separation for RNA extraction**. For nucleus–cytoplasm separation, all reactions were carried out on ice and centrifugations at 1000$g$ for 3 min at 4 °C. Cells were collected and resuspended in lysis buffer A (Tris-HCl, pH 7.0 10 mM; NaCl 140 mM; MgCl$_2$ 1.5 mM; NP-40 0.50%), left on ice for 5 min and then centrifuged. The supernatants (i.e., cytoplasms), were used for RNA extraction, while the pellets were washed three times with lysis buffer A and then resuspended in lysis buffer B (Tris-HCl, pH 7.0 10 mM; NaCl 140 mM; MgCl$_2$ 1.5 mM; NP-40 0.50%; Tween-40 1%; deoxycholic acid 0.50%). After centrifugation, pellets containing the nuclei were used for RNA extraction as previously described. For subnuclear fractionation, the pellet containing the nuclei was washed with 400 µl lysis buffer A supplemented with with RNase inhibitor and protease inhibitor and then processed as described in ref.[52].

**End-point and quantitative PCR**. Takara Ex Taq Kit (Takara Bio Inc., Japan) was used to perform end-point PCR according to manufacturer's instructions. Primers used are listed in Supplementary Table 5.

Conventional quantification of gene or miRNA expression was assessed by qRT-PCR using No AmpErase TaqMan Universal PCR Master Mix (Thermo Fisher Scientific Inc., Waltham, MA, USA) and specific TaqMan gene-expression assays (Thermo Fisher Scientific Inc., Waltham, MA, USA), as listed in Supplementary Table 4. Amplifications were run on the 7900HT Fast Real-Time PCR System. Data were analyzed by SDS relative quantity (RQ) Manager 1.2.1 software (Thermo Fisher Scientific Inc., Waltham, MA, USA) and reported as RQ or −ΔΔCt with respect to a calibrator sample using the comparative Ct (ΔΔCt) method. *GAPDH/SNORA74A* and *RNU48* were used as normalizers in gene and miRNA expression studies, respectively. If not otherwise specified, *MIR205HG/LEADR* expression was measured using an assay covering ex-2/3 boundary, which detects all possible transcripts.

Enrichment of DNA regions in ChIP and ChIRP experiments was evaluated through SYBR green qPCR run on a CFX Connect Real-time PCR instrument, using SsoAdvanced Universal Sybr Green Master mix (Bio-Rad, Hercules, CA, USA) and primers listed in Supplementary Table 5.

**Immunoblotting analysis**. For total protein extraction, cell were lysed in lysis buffer (Tris-HCl pH 7.4 10 mM; NaCl 10 mM; Triton X-100%; PMSF 1×; Aprotinin 5 µg/ml; Leupeptin 20 µg/ml) for 30 min on ice and then supernatant (i.e., proteins) was boiled at 95 °C for 5 min. Proteins were loaded onto a 4–12% precast Bis–Tris sodium dodecyl sulfate polyacrylamide gel electrophoresis (*SDS-PAGE*) gel (NuPAGE, Thermo Fisher Scientific Inc., Waltham, MA, USA) for separation and transferred onto Hybond nitrocellulose membranes (GE Healthcare Life

Sciences, Buckinghamshire, UK). Filters were blocked for non-specific reactivity by incubation for 1 h at room temperature in 5% skim milk dissolved in 1× PBS-Tween 20 and overnight at 4 °C probed with antibodies listed in Supplementary Data 4. After three washes with 1× PBS-Tween 20, filters were incubated with the secondary horseradish peroxidase-conjugated anti-mouse (NA931V, GE Health-care Life Sciences, Buckinghamshire, UK) or anti-rabbit (NA9340V, GE Healthcare Life Sciences, Buckinghamshire, UK) antibodies for 1 h at room temperature. Immunoreactivity was detected by the enhanced chemiluminescence (ECL) immunodetection system (WP20005, Thermo Fisher Scientific Inc., Waltham, MA, USA).

The Nuclear/Cytoplasm Fractionation Kit (MBL™, Woburn, MA, USA) was used to separate proteins from the fractions, according to manufacturer's protocol. Immunoblotting was then conducted as described above, using the antibodies as listed in Supplementary Data 4. Uncropped scans of all blots are reported in Supplementary Fig. 7.

**Immunofluorescence**. Immunofluorescence was carried out as described in ref. [2], except for permeabilization conducted with methanol:acetone 1:1. Antibodies are listed in Supplementary Data 4. For p63, an Eclipse E600 microscope (Nikon, Japan) and a Digital Camera DXM1200 (Nikon, Japan) were used to acquire images, later processed with Adobe Photoshop Image Reader 7.0. AR imaging was instead performed using a confocal laser scanning microscope Leica TCS SP8 X (Leica Microsystems GmbH, Mannheim, Germany). The fluorochrome was excited by a pulsed super continuum White Light Laser (470–670 nm; 1 nm tuning step size). In particular, AR was excited selecting 598 nm-laser line and detected from 603 to 725 nm, whereas nuclei were visualized using DAPI excited with 405 nm diode laser and detected from 426 to 501 nm. The images were acquired in the scan format 1024 × 1024 using a HC PL APO 63×/1.4 oil immersion objective and a pinhole set to 1 Airy unit. The data were analyzed using the ImageJ software.

**ELISA for PSA**. Conditioned media from cells seeded at the density of 10,000 cells/cm² were collected, clarified by centrifugation and sixfold concentrated with Concentrator Spin 5K MWCO (5185-5991; Agilent Technologies, Santa Clara, CA, USA) at 4000g for 15 min. PSA levels in the concentrated conditioned media were measured using ELISA assay (EHKLK3T; Thermo Fisher Scientific Inc., Waltham, MA, USA), according to the procedures recommended by the manufacturer.

**In vitro transcription/translation**. In vitro translation assay was performed using the TnT Quick Coupled Transcription/Translation System (Promega, Fitchburg, WI, USA), according to the manufacturer's instructions. Reactions were carried out using, alternatively, 1 mM transcend biotin-lysyl-tRNA or 10 mCi/ml [³⁵S]-methionine. The eventual labeled translation products were then separated on SDS-PAGE gels, transferred onto nitrocellulose membrane and visualized by binding of streptavidin–horseradish peroxidase, followed by chemiluminescent detection, for the nonradioactive method, while radioactive products were visualized with autoradiography. The assay was carried out on two MIR205HG/LEADR-specific pCMV-6AC plasmid vectors, one containing the whole genomic sequence and one the RefSeq transcript, and the respective empty vector as negative control. A pcDNA3.1 vector (Thermo Fisher Scientific Inc., Waltham, MA, USA) containing ΔNp63α cDNA[2] served as positive control.

**RNA immunoprecipitation**. RNA immunoprecipitation (RIP) was performed on nuclear extracts obtained from RWPE-1 cells in native conditions, as described in ref. [53]. Lysates were split into three fractions for mock and specific immunoprecipitations, with an aliquot frozen in liquid nitrogen for reference RNA isolation (Input). Totally, 5 μg rabbit anti-IRF1 (sc-497, Santa Cruz Biotechnology Inc., Santa Cruz, CA, USA) or rabbit anti-IRF7 (ab109255, Abcam, Cambridge, UK) antibodies were added to supernatant and incubated overnight at 4 °C with gentle rotation. IgG (Ab2410, Abcam, Cambridge, UK, rabbit) control sample was included. Bound RNAs were precipitated with protein A- and G-agarose (GE Healthcare Life Sciences, Buckinghamshire, UK) and then isolated using QIAzol Lysis Reagent (QIAGEN, Hilden, Germany) reagent. RNA precipitation was favored by adding 1 μg/μl glycogen to isopropanol. qRT-PCR was conducted as described above.

**Cross-linking immunoprecipitation**. UV-CLIP experiments were performed according to Schaukowitch et al.[54]. Briefly, 1 × 10⁷ cells were cross-linked at 150 mJ/cm² using UV-CROSSLINKER (Hoefer Scientific Instruments, San Francisco, CA, USA) for 36 s in ice-cold PBS to preserve protein-RNA interaction. Totally, 10 μg rabbit anti-IRF1 (8478, Cell Signaling Technology Inc., Danvers, MA, USA) or rabbit anti-IRF7 (ab109255, Abcam, Cambridge, UK) antibodies were added to isolated lysed nuclei and incubated overnight at 4 °C with gentle rotation. IgG (Ab2410, Abcam, Cambridge, UK, rabbit) control sample was included. Bound RNAs were precipitated with protein A- and G-agarose (GE Healthcare Life Sciences, Buckinghamshire, UK), washed with increasingly stringent buffers as described in ref. [54] and eluted upon Proteinase K (Thermo Fisher Scientific Inc., Waltham, MA, USA) treatment for 2 h at 50 °C. Eluted RNA was isolated with an equal volume of phenol: chloroform:isoamyl 25:24:1 pH 8 (Thermo Fisher

Scientific Inc., Waltham, MA, USA) and precipitated with ethanol 75% overnight. Isolated RNA was resuspended in nuclease-free water and qRT-PCR was performed as previously described.

RNA-Seq data of DROSHA and DGCR8 CLIP experiments were retrieved from GSE61979. Bedgraph files were then uploaded onto UCSC human assembly hg19 and visualized with UCSC Genome Browser.

**RNA pulldown**. For RNA pulldown experiment, pCMV-6AC-LEADR and pCiNeo-EGFP (Addgene, # 46949, used as control as it codifies for a transcript of length comparable to LEADR) plasmids were digested with XhoI and SpeI restriction enzymes (New England Biolabs, Ipswich, MA, USA) respectively, and purified with Agencourt AMPure XP beads (Beckman Coulter, Brea, CA, USA). Linearized plasmids were in vitro transcribed with biotin–UTP using Biotin RNA Labeling Mix and T7 RNA Polymerase (Hoffmann-La Roche, Basel, Switzerland) and purified with miRNeasy Mini Kit (QIAGEN, Hilden, Germany).

RNA pulldown experiment was carried out basically as described in Doron-Mandel et al.[55] with minor modifications. Totally, 50 μg of Dynabeads MyOne Streptavidin C1 (Thermo Fisher Scientific Inc., Waltham, MA, USA) beads for each pulldown were washed and RNase inactivated according to manufacturer's instruction. Three microgram of each biotinylated RNA was resuspended in RNA Structure Buffer (20 mM Tris-HCl pH 7.5; 0.2 M KCl; 20 mM MgCl₂; 2 mM DTT; 0.8 U/μl RNase inhibitor), heated at 90 °C for 2 min and put on ice for 2 min, then shifted to room temperature for 20 min to allow proper secondary structure. RNA-beads complex and precleared nuclear extract (as described in RIP protocol above) from DU145 cells pre-treated for 6 h with 1000 IU/mL IFN-β1 were incubated as described in ref. [55] and later eluted for Western Blotting as previously described.

**Chromatin immunoprecipitation**. ChIP experiments were conducted essentially as described in ref. [56]. Briefly, 1 × 10⁷ cells were cross-linked using 1% formaldehyde for 10 min, the reaction quenched with 1/20 volume of 2.5 M glycine and centrifuged at 1350g for 5 min; the pellet was washed twice with PBS and resuspended in sonication buffer, sonicated to obtain fragments of approximately 300–600 bp, as verified on agarose gel electrophoresis. Reactions were centrifuged at 20,000g for 10 min, and the supernatants were used for incubations with antibodies overnight at 4 °C. Totally, 5 × 10⁶ equivalents of chromatin were immunoprecipitated with 5 μg of anti-IRF1 (sc-497, Santa Cruz Biotechnology Inc., Santa Cruz, CA, USA), and anti-FLAG (F3165, Sigma-Aldrich, Saint Louis, MI, USA) antibodies. Dnabeads Protein G (Thermo Fisher Scientific Inc., Waltham, MA, USA) were used for recovery of antibody-bound chromatin. Cross-linking was reversed by incubation at 65 °C overnight. Reactions were digested with RNAse A and Proteinase K and DNA purified by phenol–chloroform extraction and ethanol precipitation. DNA was resuspended in TE and used in Sybr Green qPCR, as described above, to amplify the regions of interest. An unrelated genomic region was used as internal negative control. All primers are listed in Supplementary Table 5.

**Chromatin isolation by RNA precipitation**. ChIRP was essentially performed as described in Chu et al.[57]. Ten 20-mer 3'-BiotinTEG modified antisense probes tiling exons of MIR205HG/LEADR transcript were designed by using the online probe designer tool at singlemoleculefish.com, according to the following parameters: number of probes = 1 probe/100 bp of RNA length; target GC% = 45; oligonucleotide length = 20; spacing length = 60-80. Regions of repeats or extensive homology (such as Alu/SINE sequence spanning exons 1 and 2) were omitted and ten probes were generated. A symmetrical set of probes against lacZ RNA was also generated and used as the mock control. 3'-BiotinTEG modified probes were synthesized by Eurofin MWG Operon (Ebersberg, Germany). All probe sequences are listed in Supplementary Table 6.

RNA was obtained from ChIRP-ed samples upon reversal of cross-linking by proteinase K and extraction through TRIzol reagent (Thermo Fisher Scientific Inc., Waltham, MA, USA) according to manufacturer's protocol. qRT-PCR was then carried out using Taqman assays (as in Supplementary Table 4) to check for specific RNA retrieval.

For DNA elution samples were resuspended in DNA elution buffer and incubated at 37 °C for 30 min with end-to-end rotation. Beads were captured by T-1-800 Adna Mag S magnets (AdnaGen, Langenhagen, Germany). Eluted DNA from two steps was combined. Chromatin was reverse cross-linked at 65 °C overnight with 200 mM NaCl and treated with Proteinase K at 45 °C for 2 h. DNA was then extracted with equal volume of phenol:chloroform:isoamyl 25:24:1 pH 8 (Thermo Fisher Scientific Inc., Waltham, MA, USA) and precipitated with ethanol 80% overnight. Isolated DNA was resuspended in elution buffer (QIAGEN, Hilden, Germany) and eluted DNA was subject to Sybr Green qPCR to assess enrichment of genomic regions of interest over unrelated region. All primers are listed in Supplementary Table 5.

**In silico assessment of coding potential**. The coding potential of MIR205HG/LEADR transcripts was initially assessed using the alignment-free CPAT version 1.2.1 (http://lilab.research.bcm.edu/cpat/[19]), which assigns a coding probability on the base of the length of transcript and of the predicted ORF.

The presence of functional regions within MIR205HG/LEADR putative protein sequence was checked by using Pfam (pfam.xfam.org/), BLASTP (blast.ncbi.nlm.

nih.gov/), SMART (smart.embl-heidelberg.de/) and InterPro (www.ebi.ac.uk/interpro/) tools, which analyze the putative protein sequence; whereas, for the analysis of nucleotide sequence, Pfam and BLASTX tool were used.

**Site-directed mutagenesis**. Deletion of *Alu* portion from *MIR205HG/LEADR* RefSeq transcript was obtained through site-directed mutagenesis by PCR, as described in ref. [58]. *MIR205HG/LEADR* vector, previously used to transform Dam-positive in bacteria, was amplified using high fidelity DNA Polymerase Q5 (New England Biolabs, Ipswich, MA, USA), in the presence of the deletion primers listed in Supplementary Table 7. Dpn I restriction enzyme digestion was used to eliminate the wild type methylated vector, whereas remaining products were used to transform competent JM109 cells (Promega, Fitchburg, WI, USA). Colonies were then Sanger-sequenced to check for successful deletion.

**Motif discovery**. In search for mechanistic insights, we selected the genes found upregulated upon *LEADR* silencing and coherently downregulated upon overexpression (referred to as "*LEADR*-core up") as bona fide *LEADR* targets. Such choice was dictated by the evidence that overlap between genes modulated upon silencing by the two different approaches (siRNA and gapmer) was greater (Fisher test $p = 6.88e{-}08$ and $p = 4.36e{-}12$, respectively) among upmodulated (136/588 and 136/473, average fraction of 26%) than downmodulated genes (22/320 and 22/272, average fraction of 7.4%). In addition, the overlap with genes showing opposite trend upon *LEADR* overexpression was greater among upmodulated (38/136 = 28%) than downmodulated (2/22 = 9%) genes. Altogether, these observations suggested that "*LEADR*-core up" genes could be enriched of true direct *LEADR* targets.

Promoter sequences of "*LEADR*-core up" were thus retrieved from RefSeq UCSC Table (hg19) and the first isoform listed for each gene was used for the analyses. MEME Suite[59] and Pscan[60] were used (with standard parameters) for the de novo motif discovery (and their alignment on *LEADR* transcript) and for the inspection of TF binding sites in promoters, respectively.

**Analysis of ChIP-Seq data and site frequency estimation**. ChIP-Seq IRFs binding sites were retrieved from Encode published data (GSE32465 and GSE31477). Histone modifications in RWPE-1 cells were retrieved from GSE63094. Comparison between genomics intervals were performed with UCSC Table Browser. Coordinates of *Alu* sequences across genome were retrieved from Repeat Mask UCSC Table. Frequency of *Alu* and IRF sites in selected gene lists was estimated using the promoters of all genes present on the microarray platform as background.

**Preprocessing of publicly available gene expression data**. Microarray-based gene expression profiles were retrieved as normalized dataset matrix from the Gene Expressed Omnibus (GEO) repository with the following accession numbers: GSE3998, GSE86904, GSE89050, and GSE21034 (samples from patients who received systemic treatment before surgery were excluded from the analyses), GSE5993, GSE29782, GSE75628, and GSE48160. *MIR205HG* data from Genotype-Tissue Expression (GTEx) project [https://www.ebi.ac.uk/gxa/experiments/E-MTAB-5214/Download] were downloaded as TPM values. RNA-Seq data, retrieved from GSE82071 and TCGA firehose (RNAseqv2 level 3 data, accession date 2016-01-28, http://gdac.broadinstitute.org/), were normalized using the trigger mean of *M* value of *edgeR*[61] package and log$_2$-transformed. Only TCGA cancer types with at least 100 tumor and 3 normal samples with available *MIR205HG* and *miR-205* expression were considered for the analysis. All publicly available datasets analyzed through this study are listed in Supplementary Data 5, along with details on technique, type of samples, related publications, and reuse made in this work.

**Gene-expression analysis of *LEADR*-modulated cells**. Transcriptomic data from prostate cells modulated for *MIR205HG/LEADR* expression using different strategies were generated in our laboratory using Illumina HumanHT-12 v4 arrays, as previously described[62]. The silencing of *MIR205HG/LEADR* obtained with the different strategies described in the paper (siLEADR, gapLEADR and gapINT1) was conducted in triplicate (three independent biological replicates, each including control and *LEADR*-specific oligomer) for subsequent gene-expression analysis. Overexpression experiments with wild type *LEADR* (RefSeq, 7063 and whole genomic sequence "gene") were conducted in quadruplicate (each including empty and *LEADR*-specific vector), whereas overexpression of *Alu*-deleted form in triplicate.

Raw data were log$_2$-transformed and normalized using the robust spline method implemented in the *lumi* package[63]. Normalized data were filtered removing probes with neither at least one detection *p* value < 0.01 across samples, nor associated official gene symbol; for probes mapping on the same gene symbol, the one with highest variance was selected.

Gene-expression data and processing pipeline were deposited at Gene-Expression Omnibus, with accession number GSE104003.

**Bioinformatic analyses**. Differential expression was estimated both in terms of fold change (FC) and *t* value, using the limma Bioconductor package[64].

Significance was provided in terms of false-discovery rate (FDR) to take into account the adjustment for multiple hypotheses testing, using a threshold of 0.05. All these analysis were conducted in R environment.

A *t* value preranked gene set enrichment analysis[65] was carried out using Hallmark gene sets v5.2 of the Molecular Signature database (MSigDB) or custom signatures. Customly defined basal or luminal-specific gene sets were obtained selecting either the top-100 significantly down or upregulated genes in luminal vs. basal cells, as from publicly available datasets (gene set lists reported in Supplementary Data 6). The normalized enrichment score was visualized in heatmaps by means of unsupervised hierarchical clustering along the signatures (euclidean distance and ward linkage metrics); FDR *q* value threshold of 0.05 was used to assess a significant enrichment.

*LEADR* transcript structures were retrieved from GSE93848 ([https://public_docs.crg.es/rguigo/CLS/][13] as gtf files and hg38 fasta file from UCSC, and shortlisted through comparison with those retrieved from SRP057660 dataset[14], as described in the Results section. RNA-Seq data from GSE67070, GSE22260, GSE75035, and GSE25183 were retrieved as sra files with sratoolkit tool and transformed in fastq paired files with fastq-dump –split-3 command. RSEM package was used to construct the reference with rsem-prepare-reference –no-polya and then to calculate expression of isoforms with rsem-calculate-expression --paired end.

**Statistical analyses**. Boxplots were generated according to Tukey's visualization, where the box encompasses the 25th and 75th percentile, the horizontal line within the box is the median (50th percentile) of the distribution, and the length of the whiskers is calculated as 1.5 IQR (interquartile range). Barplots were overlaid with dots of single values.

Two-tailed *t* test was generally used to evaluate different behavior between control and treated cells in cell-based experiments. In case the difference was expected to occur only in one direction (Fig. 3a, Fig. 5g, Supplementary Fig. 4f, Fig. 6i, Fig. 9b, and Fig. 9d), one-tailed *t* test was used. Paired test was used when appropriate. To assess differential expression of *MIR205HG/LEADR* in publicly available datasets, Mann–Whitney *U* test was used. Jonckheere–Terpstra test was applied to assess differences in *LEADR* expression across a priori ordered conditions. Hypergeometric test was used to investigate the equality proportion of *Alu*/IRF binding sites in different sets of genes. A threshold of 0.05 was considered statistically significant. Number of replicates for each experiment is listed in the Source Data file. The Kaplan–Meier method was employed to estimate the biochemical recurrence-free survival curves and the log-rank test to assess the significance of the difference between the curves. A *p* value threshold of 0.05 was considered.

**Reporting Summary**. Further information on experimental design is available in the Nature Research Reporting Summary linked to this Article.

## Data availability
Transcriptomic data from prostate cells modulated for *MIR205HG/LEADR* expression generated in our laboratory were deposited at Gene Expression Omnibus, with accession number GSE104003. Accession codes (and web links) for publicly available datasets have been provided in the Methods section and Supplementary Data 5. The Source data underlying Figs. 1a–c, e–h, 2b–f, 3a, b, 4b, c, 5b, c, e, g, i, 6f, g, i, j, 7d, 8d, e, 9a–e and Supplementary Figs. 2a, c, 3b, c, 3f, g, 4a, b, e, f, 5c, 6a, e, f are provided as a Source Data file. A Reporting Summary for this Article is available as a Supplementary Information file. All other relevant data are available within the article and its Supplementary files or from the corresponding authors upon reasonable request.

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

## Acknowledgments

We thank R. El Bezawy, G. Cimino-Reale, and M. Colecchia for helpful technical assistance, and F. Nicassio, M. Folini, and R. Mantovani for constructive discussions. We

thank the Platform of Integrated Biology of Fondazione IRCCS Istituto Nazionale dei Tumori for microarray experiments. We are also grateful to D. Majerna for help in language editing. This work was supported by grants from: Italian Ministry of Health (GR-2013-02355625 to P.G.), CARIPLO Foundation (2015-0866 to P.G.), and I. Monzino Foundation (to N.Z.).

## Author contributions

V.P., B.F., S.P., F.R., V.D., E.F., N.F. and D.D. performed the experiments, analyzed the data, and critically revised the manuscript; S.P., D.D., M.D., D.R. and M.B. performed bioinformatic and statistical analyses and critically revised the manuscript; N.Z. and R.V. contributed advice and critically revised the manuscript; P.G. conceived and supervised the study, analyzed the data, and wrote the manuscript.

## Additional information

**Competing interests:** The authors declare no competing interests.

