## [Peer Review File · Nature Communications]

Reviewers' comments:

Reviewer #1 (Remarks to the Author):

In this manuscript, Profumo et al present evidence for MIR205HG as a nuclear lincRNA that plays a function in maintaining prostate basal cell properties and preventing luminal cell differentiation. Although there are many publications on miR-205, which, like the host gene MIR205HG, is preferentially expressed in the basal cell layer of epithelial organs including the prostate, little is known about MIR205HG. Therefore, the study is very novel. The authors show that interestingly, only one of the MIR205HG transcripts is productive in generating miR-205, which, in fact represents probably only a minor % of the total transcripts. Mechanistically, MIR205HG directly binds the Alu element in the promoter regions of, intriguingly, IFN genes. As the binding is in close proximity to IRF-binding site, MIR205HG represses IFN gene transcription. These mechanistic studies prompted the authors to re-annotate MIR205HG as LEADeR (or LEADR), for long epithelial Alu-interacting differentiation related RNA. The authors' data suggests an exciting model (Fig. 7) of how MIR205HG might regulate gene expression and impact epithelial cell differentiation. The paper presents abundant bioinformatics based data; however, biological interrogations are overall weak. For instance, most biological assays on the effects of LEADeR on basal-luminal differentiation were performed in a single immortalized cell line RWPE-1. The authors should consider purifying out primary human prostatic basal cells and then assaying cellular differentiation using PSA production (e.g., PNAS 94, 10705-10, 1997) or using organoid assays upon manipulating LEADeR levels. Also, their observations that LEADeR directly interacts with IRF protein in the nucleus should be validated in other cell systems. There are also issues in data presentation and figure organizations.

Main critique:

1. Figure 1: LEADeR expression is enriched in prostate basal cells and reduced along luminal differentiation. There are many issues with the figure and figure legend does not have sufficient details to allow a thorough understanding of the data.
 - a. Fig. 1b is presented ahead of Fig. 1a.
 - b. Fig. 1a: What exactly are those 3 types of data presented below the 'RepeatMasker'? These data are not very informative. What does the 'Chromatin State Segmentation' color map mean? It lacks a color code. What are those cells? The histone mark ChIP-Seq heatmaps lack the relative value indicators on Y-axis. Which 7 cell lines were used?
 - c. Fig. 1b: data was acquired from the 'Expression Atlas Data Portal'. Are there biological replicates for these tissues?
 - d. Fig. 1c: The abbreviations for tumor types in TCGA should be spelt out in the legend and the "n" for each tumor type may be indicated in the figure.
 - e. Fig. 1d: The "n" for each cohort of samples should be indicated.
 - f. Fig. 1e: What's the point of this figure? It's not clear from Text or legend.
 - g. Fig. 1f: The "GSE3998" name should be indicated on top of the figure (just like in Fig. 1d) and the "n" for each cohort of samples should also be indicated.
 - h. Fig. 1g: These cells presumably were cultured in different types of media; therefore, the data should be interpreted with caution. What are "n" for each type of cells (biological replicates)?
 - i. Fig. 1h: Are the data from independent biological experiments?
 - j. Fig. 1i: The "GSE89050" name should be indicated on top of the figure and "n" should be indicated.
 - k. Importantly, for this part, authors should perform in situ hybridization experiment for miR-205 and MIR205HG in patient samples to demonstrate both spatial and quantitative differences between basal and luminal cells in normal glands and between normal vs. tumor areas.
2. Figure 2: miR-205 is produced through Drosha-dependent processing of an alternative MIR205HG transcript.
 - a. Supplementary Fig. S2a: Comparisons should be made with RWPE-1 cells without any

manipulations or with cells expressing the control vector(s).

b. Fig. 2c: why was the expression of miR-205, beside LEADR, also restored by overexpressing full-length LEADR, which is considered as miR-205 incompatible? Did Drosha knockdown have an influence on the production of Refseq and full-length LEADR?

c. The authors proposed that ".....alternative splicing/transcription termination may dictate the switch between LEADR.1 and LEADR.2....". What are the relative expression profiles/abundance of the two transcripts in RWPE-1 vs. prostate cancer cells?

3. Figure 3: LEADeR is a nuclear chromatin-associated LincRNA.

a. Fig. 3a: Brief descriptions about some terms in the Table (e.g., Ficket score, hexamer score) should be provided in the legend. Is the maximum 'coding probability' 1?

b. Fig. 3b: the quality of the 'biotin-labeled lysine' gel is not good.

c. The authors proposed that ".....alternative splicing/transcription termination may dictate the switch between LEADR.1 and LEADR.2....". What are the relative expression profiles/abundance of the two transcripts in RWPE-1 vs. prostate cancer cells?

4. Figure 4: LEADeR regulates basal-luminal differentiation.

a. These experiments were done using a single immortalized cell line RWPE-1. What are the changes in AR and PSA, the true differentiation markers? Authors should also consider purifying out primary human prostatic basal cells and then assaying cellular differentiation using PSA production (e.g., PNAS 94, 10705-10, 1997) or using organoid assays upon manipulating LEADeR levels.

b. In Fig. 4h, how many and what genes are included in basal- or luminal- specific gene set? Is there an overlap with LEADR signature?

5. Figure 5: LEADeR controls genes of the interferon signaling.

a. Were these RNA-Seq experiments? How many biological replicates were used? Where was the RNA-Seq QC data?

b. Fig. 5a: The two different cell types should be indicated above. Going with consistent changes in interferon response, there are a lot of inflammatory pathways that got changed, which should be mentioned/discussed in slightly greater details.

c. Fig. 5d-f: Again biological data demonstrating IFN signaling with luminal differentiation was provided in only one cell line and not convincing.

6. Figure 6: LEADeR directly binds promoters of genes having an Alu and IRF binding site motif in tandem.

a. Data in Fig. S3b should appear in the main figure.

c. Fig. 6c was not cited in Text.

Reviewer #2 (Remarks to the Author):

In this manuscript Profumo and colleagues investigate the long noncoding transcript LEADeR, which is generated from the same locus as miR-205, however has a function independent of the microRNA. They show that LEADeR is a nuclear lincRNA that regulates the differentiation of human prostate basal cells. They investigate the mechanism, finding that LEADeR binds to Alu elements, which are present at the promoters of interferon responsive genes. LEADeR inhibits the expression of those genes to prevent luminal differentiation.

This is overall an interesting and well-performed study. However some points need further clarification or reinforcement in order to support the conclusions.

1. The first point refers to the decision between miR-205 and LEADeR generation. The expression levels of the different isoforms in the cell lines and the datasets analyzed are not clear. To get an

idea of are relative expression levels of each isoform in different tissues more information can be extracted from datasets such as GTEx portal.

In addition it would be useful to get the data of the transcript structures from most recent annotations or datasets such as the one generated with PacBio technology available at:

https://public_docs.crg.es/rguigo/CLS/

2. To further support the proposed mechanism of differential splicing it may be useful to analyze DGCR8-CLIP-seq or Drosha-CLIP-seq datasets to find out where Drosha is binding LEADeR. This piece of information will support the proposed mechanism of LEADeR's differential exon usage based on Drosha's binding and masking of a splicing site.

3. The authors select genes regulated by LEADeR based on the gene expression changes observed upon inhibition or overexpression of LEADeR. Can the enforced expression of LEADeR reduce or revert the differentiation of the cells? What is the phenotype of LEADeR- overexpressing cells?

4. A problem of the study is that the authors decide to focus on genes that are up-regulated when LEADeR is inhibited, which is arbitrary since the bias 64/36% is not strong enough evidence to support that LEADeR acts as a repressor.

The authors have restricted the search for motifs in the up-regulated genes, and Alu sequences are found enriched in the promoters of the "LEADR-core up" genes. What sequence elements are found in the rest of the genes? How likely is it to find this type of sequences in random gene promoters?

5. The co-occurrence of Alu elements and IRF sites in the promoter of the genes regulated by LEADeR is very interesting. However the mechanism underlying their regulation by LEADeR is not convincingly shown. The hypothesis that LEADeR binds to IRF1 protein doesn't seem to be supported by the weak RIP data.

6. It has been recently reported that Alu repeats drive lncRNA nuclear retention. In the same line a cell fractionation experiment can be performed to see if LEADeR's localization to chromatin changes when the Alu sequence is not present.

7. Where are the ChIRP probes mapping? Because the homology of sequence between LEADeR and the Alu elements, the enrichment can be due to the direct recognition of the DNA sequences by the probes. This technical considerations are very important when designing and interpreting ChIRP experiments.

8. What is the effect of LEADeR knockdown or overexpression on IRF binding? This should be shown by ChIP on a large set of gene targets, including positive and negative controls.

9. To further explore the role of the sequence elements, different mutants could be made with CRISPR/Cas9 on the endogenous promoter of one of the targets to test the effect on their regulation by LEADeR.

Minor comments

Figure 1

1. Figure 1a. In the RepeatMasker track, show which types of repeating elements are displayed.

2. Figure 1b,c. Complement with mir-205 expression in the same samples. For Figure 1c, which tumor types have a significant difference in LEADeR's expression comparing tumor to normal? Show the p-values.

3. Figure 1f. The data set title is missing (GSE3998) ?

Figure 2

1. Figure 2f. Results from siDROSHA2 and quantification of the bands (plot as %)

Figure 4

1. Figure 4f. Do you see the same change in localization of p63 using gapmers? If so, then add an extra figure panel of p63 immunofluorescence with gapCTR and gapLEADER.

Reviewer #3 (Remarks to the Author):

Both small and long non-coding RNAs play important roles during cell differentiation and tumorigenesis. The authors characterized the function of the lncRNA, LEADeR, which is the host gene of microRNA mir-205, in normal prostate epithelial cells and prostate cancer cell line. They provided evidence suggesting that LEADeR may bind to Alu elements which are located in the promoter regions of certain genes. Meanwhile, LEADeR may also bind to IRF binding site and may inhibit IRF activity. The topic is timing and the finding is interesting. It may provide new insight into prostate cancer biology. However, the mechanism studies are still preliminary. The authors need to experimentally address the follow questions:

Major:

The binding between LEADeR and IRF: The authors performed RNA-IP using IRF antibody suggesting that LEADeR may be associated with IRF, however, the result is need to be further confirmed by RNA-pulldown following by qRT-PCR using labeled RNAs. In addition, the RNA-IP was performed in native condition, it only suggests that LEADeR may be associated with IRF protein. They should demonstrate if this interaction is direct binding (or mediated by other proteins, i.e. indirect binding). Finally, the experiments should also be perform to test if knockdown or overexpression of LEADeR reduces the LEADeR-IRF interaction.

The binding between LEADeR and Alu: Same as the above comments, more validation and functional experiments (e.g. knockdown or overexpression of LEADeR following by ChIRP) should be performed.

The function of mir-205: Although the function of mir-205 has been well-characterized, how mir-205 cooperates with LEADeR during differentiation should be carefully characterized in this study, given that mir-205 and LEADeR are in the same transcription unit and their expression levels are significantly correlated.

Minor:

Figure 1, 2 and 3 are not informative. These figures could be combined to one figure, or some panels in the figures could be used as online figures.

More detailed information about how to analyze array experiments should be provided. What is the cut-off and statistical method? How to choose the cut-off.....

What is the clinical significance of the RNA expression level of LEADeR in prostate and other cancer types? TCGA provides a novel resource to explore.

** See Nature Research's author and referees' website at www.nature.com/authors for information about policies, services and author benefits

Point-by-point response to the referees' comments

Reviewers' comments:

Reviewer #1 (Remarks to the Author):

In this manuscript, Profumo et al present evidence for MIR205HG as a nuclear lincRNA that plays a function in maintaining prostate basal cell properties and preventing luminal cell differentiation. Although there are many publications on miR-205, which, like the host gene MIR205HG, is preferentially expressed in the basal cell layer of epithelial organs including the prostate, little is known about MIR205HG. Therefore, the study is very novel. The authors show that interestingly, only one of the MIR205HG transcripts is productive in generating miR-205, which, in fact represents probably only a minor % of the total transcripts. Mechanistically, MIR205HG directly binds the Alu element in the promoter regions of, intriguingly, IFN genes. As the binding is in close proximity to IRF-binding site, MIR205HG represses IFN gene transcription. These mechanistic studies prompted the authors to re-annotate MIR205HG as LEADeR (or LEADR), for long epithelial Alu-interacting differentiation related RNA. The authors' data suggests an exciting model (Fig. 7) of how MIR205HG might regulate gene expression and impact epithelial cell differentiation. The paper presents abundant bioinformatics based data; however, biological interrogations are overall weak. For instance, most biological assays on the effects of LEADeR on basal-luminal differentiation were performed in a single immortalized cell line RWPE-1. The authors should consider purifying out primary human prostatic basal cells and then assaying cellular differentiation using PSA production (e.g., PNAS 94, 10705-10, 1997) or using organoid assays upon manipulating LEADeR levels. Also, their observations that LEADeR directly interacts with IRF protein in the nucleus should be validated in other cell systems. There are also issues in data presentation and figure organizations.

Main critique:

1. Figure 1: LEADeR expression is enriched in prostate basal cells and reduced along luminal differentiation. There are many issues with the figure and figure legend does not have sufficient details to allow a thorough understanding of the data. a. Fig. 1b is presented ahead of Fig. 1a.

Fig.1 has been overall modified taking into account all referees' requests. We carefully checked citation of figure panels in the text and did our best to add details in figure legends to help understanding. Fig. 1a is cited in the Introduction (page 3, line 18).

b. Fig. 1a: What exactly are those 3 types of data presented below the 'RepeatMasker'? These data are not very informative. What does the 'Chromatin State Segmentation' color map mean? It lacks a color code. What are those cells? The histone mark ChIP-Seq heatmaps lack the relative value indicators on Y-axis. Which 7 cell lines were used?

We now moved RepeatMasker, Chromatin State Segmentation and histone mark tracks to Supplementary Figure S1, where we expanded all tracks to help readability. RepeatMasker is now reporting the different types of repeat elements along *MIR205HG* locus. SINE/Alu element spanning exons 1 and 2 is made evident.

As for chromatin state segmentation, meaning of colors is now made explicit on the left. Specifically: red for Active Promoter, orange for Strong Enhancer, yellow for Weak Enhancer, gray for Repressed, and light gray for low signal (heterochrom/lo). Cell lines are indicated above

each track and acronyms spelt in the legend (HMEC, Human Mammary Epithelial Cells; NHEK, Normal Human Epidermal Keratinocytes; H1-hESC, human Embryonic Stem Cells; K562, chronic myeloid leukemia cell line; GM12878, lymphoblastoid cell line; HSMM, Human Skeletal Muscle Myoblasts; HUVEC, Human Umbilical Vein Endothelial Cells; NHLF, Normal Human Lung Fibroblasts). *MIR205HG/LEADR* chromatin results in open conformation mainly in human mammary epithelial cells (HMEC) and keratinocytes (NHEK).

As far as histone modifications are concerned, track of H3K27 Acetylation is now shown in both packed and extended form to allow a better comprehension of the color code for the different cell lines. Tracks for other histone modifications are shown in the packed form, and show that *MIR205HG/LEADR* is only actively transcribed in NHEK keratinocytes (among ENCODE cell lines).

c. Fig. 1b: data was acquired from the 'Expression Atlas Data Portal'. Are there biological replicates for these tissues?

Illumina body map (i.e. Expression Atlas) data portal does not provide number of replicates, so we interrogated GTEx project where such information is available. The analysis confirmed peculiar expression of *LEADR/MIR205HG* in epithelia as compared to other tissue types. These data are shown in the current Fig. 1b, as mean + standard deviation (sample size for each tissue is reported in Supplementary Table S1).

Additional information on *LEADR/MIR205HG* expression in normal tissues is now shown also in Supplementary Fig. 2a, as from analysis of TCGA data. The plots comparatively report expression of *MIR205HG* and *miR-205* in normal tissues (as requested by referee 2) and corresponding tumors. Number of samples for each tissue type is reported in Supplementary Table S1.

d. Fig. 1c: The abbreviations for tumor types in TCGA should be spelt out in the legend and the "n" for each tumor type may be indicated in the figure.

Figure 1c shows the fold-change of *MIR205HG* and *miR-205* expression between tumor and corresponding normal tissues, as from data individually plotted in Supplementary Fig. 2a (n size reported in Supplementary Table S1). Only tumors where *MIR205HG* is significantly differentially expressed compared to normal counterparts are shown in Fig. 1c (full data in Supplementary Fig. 2a). For space constraints, the abbreviations for tumor types in TCGA are spelt in the legend of Supplementary Fig. 2a.

e. Fig. 1d: The "n" for each cohort of samples should be indicated.

Fig. 1d left has been removed as these data are now shown in Supplementary Fig. 2a for prostate cancer and other tumors. Previous Fig. 1d right (analysis of GSE21034 dataset) is now Fig. 1d. N size for both cohorts is reported in Supplementary Table S1.

f. Fig. 1e: What's the point of this figure? It's not clear from Text or legend.

The figure shows the ROC curve (plot of sensitivity and specificity) for *MIR205HG* capability to classify tumor from normal tissues in TCGA and GSE21034 datasets. The high area under the curve (AUC) values indicate that the sole expression of *MIR205HG* can correctly classify tumor and normal tissues, as outlined in the Results section (page 4, line 23 - page 5, line 1). In the current version of the manuscript, in response to referee 3, we also showed association between

MIR205HG expression in the primary tumor and time to biochemical relapse (page 5, lines 3-4; Supplementary Fig. S2b).

g. Fig. 1f: The “GSE3998” name should be indicated on top of the figure (just like in Fig. 1d) and the “n” for each cohort of samples should also be indicated.

Dataset numbers have been indicated on top of each figure reporting analysis of publicly available datasets. N size for Fig. 1f is reported in Supplementary Table S1.

h. Fig. 1g: These cells presumably were cultured in different types of media; therefore, the data should be interpreted with caution. What are “n” for each type of cells (biological replicates)?

Of course these cell lines are cultured in different media (as from vendor’s suggestions). However, the plot confirms that *LEADR/MIR205HG* expression is low in any of the tested cell type having luminal phenotype (and especially in tumor cells). In contrast, when RWPE-1 or PrEC cells are grown in their medium formulated to maintain the basal phenotype, *MIR205HG* expression is high. Addition of serum to culture media of RWPE-1 or PrEC cells (Figure 1h and 5k respectively) actually decreases *LEADR/MIR205HG* expression in trend with luminal differentiation (as assessed by cytokeratin switch). N for Fig. 1g is reported in the legend and in Supplementary Table S1.

i. Fig. 1h: Are the data from independent biological experiments?

Yes, the data are from 3 independent experiments (now reported in the legend and in Supplementary Table S1).

j. Fig. 1i: The “GSE89050” name should be indicated on top of the figure and “n” should be indicated.

Dataset numbers have been indicated on top of each figure reporting analysis of publicly available dataset. N size is reported in Supplementary Table S1.

k. Importantly, for this part, authors should perform *in situ* hybridization experiment for *miR-205* and *MIR205HG* in patient samples to demonstrate both spatial and quantitative differences between basal and luminal cells in normal glands and between normal vs. tumor areas.

In situ hybridization on archival FFPE tissues, where tissue and cell morphology is preserved, is very challenging, as RNA (especially long RNAs) can be heavily degraded. The other way around, *in situ* hybridization may be more feasible on frozen sections, which however are not ideal for morphological and architectural assessments.

By using *in situ* hybridization on frozen tissues, we already showed specific expression of *miR-205* in epithelial cells of normal prostate glands (and absence in stromal cells) as well as loss in tumor specimens in our previous report Gandellini et al., *Cell Death Differentiation* 2012. The images are reported here below (**Fig. R1**).

Figure R1 (from Gandellini *et al.*, *Cell Death Differentiation* 2012)

To comply with referee's request, we used a biotinylated probe complementary to *MIR205HG* to perform *in situ* hybridization on frozen prostate tissue. Similar to the miRNA, *MIR205HG* is evidently expressed in normal epithelial cells as compared to stromal cells (**Fig. R2, left**). In addition, expression is lost in tumor cells (**Fig. R2, right**).

Figure R2

In situ hybridization performed on frozen tissues however is not suitable to determine specific expression of either *MIR205HG* or *miR-205* in basal vs luminal cells, though a sort of gradient of staining from basal to luminal layer can be appreciated for the lincRNA (**Fig. R2, left**).

In our view, *in situ* hybridization experiments of sufficiently good quality to demonstrate both spatial and quantitative differences between basal and luminal cells in normal glands would require extensive setting up, which is out of the scope of this first report on *MIR205HG*. Analysis of expression data of sorted pure frankly luminal and basal cells obtained from prostate tissues as from different datasets (shown in the paper in Fig. 1f, 1i, S2c) already clearly show that expression of *MIR205HG* is higher in basal compared to luminal cells, and negligible in tumor cells. Additional evidence of this came from the analysis of *LEADR* isoforms, as requested by referee 2. In this regard, Fig. 2e and Supplementary Fig. S3f again show higher expression of *MIR205HG* in isolated basal than luminal cells and loss in tumor cells.

Unfortunately, there is no availability of microRNA expression data on isolated basal and luminal cells to answer about *miR-205* pattern. In this regard, however, it was reported that *miR-*

205 expression is higher (and specific) in basal than luminal cells in mouse prostate (Zhang L *et al.*, *Prostate* 2010). Other reports confirmed the basal-specific expression of *miR-205* in human mammary gland, as from *in situ* hybridization experiments (Sempere LF *et al.*, *Cancer Res* 2007).

To circumvent this problem, we took advantage of GSE86904 dataset, which contains gene expression data of isolated basal and luminal cells obtained on an Affymetrix platform. This array interrogates the expression of a gene using probe sets, which are collections of probes mapping different portions of the entire sequence. We reanalyzed the data at single probe level in order to separate the signal intensity of *LEADR* from that of *miR-205*. Specifically, we selected two probes matching *LEADR* only and two matching *miR-205* only. Results showed that signal of *miR-205*-spanning probes was higher in basal than luminal cells, confirming the same pattern of expression of *LEADR* (Fig. R3).

Figure R3

2. Figure 2: *miR-205* is produced through Drosha-dependent processing of an alternative *MIR205HG* transcript.

a. Supplementary Fig. S2a: Comparisons should be made with RWPE-1 cells without any manipulations or with cells expressing the control vector(s).

Data are already expressed as compared to relative controls, i.e. control siRNA for p63 silencing and empty vector for p63 overexpression. This has now been made more explicit in figure legend (now Fig. S3b).

b. Fig. 2c: why was the expression of *miR-205*, beside *LEADR*, also restored by overexpressing full-length *LEADR*, which is considered as *miR-205* incompatible? Did Drosha knockdown have an influence on the production of Refseq and full-length *LEADR*?

We apologize for initially using the term ‘full-length’ when referring to the whole genomic *LEADR* sequence, which may be misleading. Data in Fig. 2c show that expression of both *miR-205* and *LEADR* (measured using an assay that covers ex-2/ex-3 junction, shared by all transcripts) is restored in DU145 cells only when the whole genomic *LEADR* sequence is transfected (previously indicated as full length, now indicated with ‘gene’, Fig. 2c), and not when only RefSeq transcript (which originates from *miR-205*-incompatible splicing) is

overexpressed. To avoid confusion, in the revised version of the manuscript, we consistently used the following nomenclature:

- **gene** (replacing ‘full length’) to label the vector including the whole *LEADR* genomic sequence;
- **RefSeq**, to label the historically annotated *LEADR* transcript (7057 in the new annotation).
- Specific numbers from 7054 to 7065 for all other newly identified transcript isoforms.

In the revised manuscript, we also incorporated data of CRISPR/Cas9 genomic deletion of the sequence spanning from exon 1 to 3 of *LEADR* (including TSS, see Supplementary Fig. S3d), which resulted in the abrogation of both *LEADR* and *miR-205* expression (page 6, lines 10-12; Fig. 2b), again supporting the hypothesis that the two RNAs share the TSS and are produced starting from a common primary transcript.

As for the effect of Drosha knockdown on *LEADR* transcripts we noticed that:

- It decreased byproduct of *miR-205* excision from compatible primary transcripts (lower band, Fig. 2h), suggesting that in the absence of Drosha the miRNA is not successfully excised;
- It simultaneously favored canonical ex-4/ex-5.1 splicing also in transcripts terminating with ex-5.2, as from sequencing of the upper band on RT-PCR shown in Fig. 2h. This suggests that Drosha also functions by masking a strong splice site to allow *miR-205* excision, a finding now confirmed by RNA-Seq of Drosha and DGCR8 CLIP (Fig. 2i).

The latter effect makes it unfeasible to measure specific expression of *miR-205*-incompatible transcripts, as their unique distinctive trait would be ex-4/ex-5.1 junction, which is created also on *miR-205*-compatible forms upon Drosha knockdown.

However, we measured global *LEADR* expression using assay covering ex-2/ex-3 junction and found that it was greatly enhanced upon Drosha knockdown (**Fig. R4**). Our view is that this is not due to selective increase of incompatible vs compatible forms (overall induction of *LEADR* expression can be also evinced by sum of bands in the gel of Fig. 2h), rather to a feedback transcriptional effect. It is likely that basal cells, in response to Drosha-mediated *miR-205* loss, try to maintain their basal features by increasing transcription of the whole locus. Such aspect would deserve additional investigation and is not included in the revised manuscript.

Figure R4

c. The authors proposed that”alternative splicing/transcription termination may dictate the switch between LEADR.1 and LEADR.2.....”. What are the relative expression profiles/abundance of the two transcripts in RWPE-1 vs. prostate cancer cells? To reply to referee 2, we now manually curated the re-annotation of all possible LEADR transcripts based on transcript structures from most recent annotations, including those of a genome-wide high-resolution remapping of pri-miRNAs by Mendell’s lab (*Chang TC et al., Genome Research, 2015*) and data of targeted RNA capture with third-generation long-read sequencing (*Lagarde J et al., Nature Genetics, 2017*), as from <https://public.docs.crg.es/rguigo/CLS/> (page 7, lines 1-12; Supplementary Fig. S3e). This led to the identification of 9 high confidence LEADR transcripts, characterized by the alternative assembly of 4 modules: exon-1/2 (present in all transcripts with or without retention of intron); exon-3 (present in all transcripts in short or long version); exon-4 (missing in some transcripts, including historical RefSeq); two alternative terminal exons, the canonical miR-205-incompatible ex-5.1 and the downstream ex-5.2 (Table in Fig. 2a, where isoforms are ranked based on abundance in prostate basal cells).

We analyzed absolute abundance and relative percentage of each of such transcript isoforms in prostate basal vs luminal cells (GSE67070); normal vs tumor tissues (GSE22260) and commercially available normal vs tumor cells (data from GSE75035 and GSE25183).

We found that

- 1- As already observed when looking at global LEADR expression levels, all LEADR isoforms are more abundant in basal vs luminal cells (Fig. 2e, left), in normal vs tumor tissues (Fig. 2e, left) and in commercially available normal vs tumor cells (Supplementary Fig. S3f). Specifically, in tumor cell lines, expression of all LEADR isoforms approximates to zero (to answer to the referee’s request to compare expression in RWPE-1 cells vs tumor cell lines). In the text, page 7, lines 18-21.
- 2- Most abundant transcripts in basal cells revealed to be 7062 and 7057 (Fig. 2e), the latter having the same exon composition as the historical RefSeq (Fig. 2a). In the text, page 7, lines 21-22.
- 3- No major differences in relative isoform expression were observed throughout the analyzed samples (Fig. 2e, right), nor between the cumulative fraction of miR-205-incompatible and compatible transcripts, which averagely accounted for 97% and 3% (Fig. 2f). In the text, page 7, line 23 - page 8, lines 1-2.

3. *Figure 3: LEADeR is a nuclear chromatin-associated LincRNA.*
a. *Fig. 3a: Brief descriptions about some terms in the Table (e.g., Fickett score, hexamer score) should be provided in the legend. Is the maximum ‘coding probability’ 1?*

We apologize for not including description of the terms in the table. These are all parameters taken into consideration for calculation of CPAT score. The maximum ‘coding probability’ is 1 and the threshold to define a transcript as potentially coding is 0.364. None of the possible LEADR transcripts exceed the cut-off for being considered as coding.

Specifically:

- Fickett score distinguishes protein-coding RNA and ncRNA according to the combinational effect of nucleotide composition and codon usage bias. Briefly, the Fickett

score is obtained by computing four position values and four composition values (nucleotide content) from the DNA sequence.

- Hexamer score is the log-likelihood ratio used to measure differential hexamer between coding and noncoding sequences. For a given DNA sequence, we calculated the probability of the sequence under the model of coding DNA and under the model of noncoding DNA, and then we took the logarithm of the ratio of these probabilities as the score of coding potential.

Description of these parameters can be found at CPAT website (<http://lilab.research.bcm.edu/cpat/>), which we cite in the Methods (page 47, lines 20-21) for space constraints.

b. Fig. 3b: the quality of the 'biotin-labeled lysine' gel is not good.
We tried to ameliorate the quality of the image.

c. The authors proposed that ".....alternative splicing/transcription termination may dictate the switch between LEADR.1 and LEADR.2.....". What are the relative expression profiles/abundance of the two transcripts in RWPE-1 vs. prostate cancer cells?

This is repetition of point 2c.

4. *Figure 4: LEADeR regulates basal-luminal differentiation.*
a. These experiments were done using a single immortalized cell line RWPE-1. What are the changes in AR and PSA, the true differentiation markers? Authors should also consider purifying out primary human prostatic basal cells and then assaying cellular differentiation using PSA production (e.g., PNAS 94, 10705-10, 1997) or using organoid assays upon manipulating LEADeR levels.

The experiments initially carried out on immortalized RWPE-1 cells were replicated on primary epithelial cells (PrEC). Results showed that *LEADR* silencing by either siRNA or gapmer induced luminal differentiation also in primary basal cells, as evidenced by morphological changes (Fig. 4a, Supplementary Fig. S4c) and cytokeratin switch (assessed by qRT-PCR, Fig. 4b, and western blotting, Fig. 4c). In the text, page 10, lines 12-17.

Regarding the effect of *LEADR* silencing on true differentiation markers AR and PSA, we must anticipate that, in general, culturing of basal cells in low Ca²⁺ serum free media (as are those of RWPE-1 and PrEC cells, see Methods, page 37, lines 5-8) prevents efficient differentiation toward the luminal-secretory phenotype (*Litvinov IV et al., Cancer Res. 2006*). In addition, EGF (present in RWPE-1 culture medium) was shown to inhibit PSA production (*Karthauss WR et al. Cell 2014*). Last, as indicated in the paper suggested by the referee, even frankly luminal cells are inhibited in PSA production when dispersed from the tissue of origin and/or in the absence of supporting stromal cells, making it difficult to assess PSA expression from 2D cultures (*Liu AY et al., Proc Natl Acad Sci U S A. 1997*).

Made these premises, we evaluated AR and PSA expression in PrEC cells silenced for *LEADR*. We found that *LEADR* knockdown by both siRNA and gapmer was sufficient to increase AR expression, in terms of both mRNA (Supplementary Fig. S4f) and protein (Fig. 4e-f). This resulted in increased PSA secretion (Fig. 4g), which however became really evident upon simultaneous DHT treatment (which further enhanced AR expression). See Fig. 4f for AR immunofluorescence and Fig. 4g for ELISA-based measurement of secreted PSA levels in *LEADR*-silenced PrEC cells in the absence or presence of DHT. These data suggest that *LEADR*

is indeed involved in basal-luminal differentiation, though activation of androgen signaling is also important for cells to acquire a terminal secretory phenotype. In the text, page 10, lines 12-17.

Further indication on the involvement of *LEADR* in basal luminal differentiation was collected as follows:

- 1) We derived basal- and luminal-specific gene sets from gene expression data of true luminal and basal cells isolated from normal prostate (4 different datasets). When tested on the transcriptome of *LEADR*-knocked down RWPE-1 cells, we found a tendency for positive enrichment for luminal and negative enrichment for basal gene sets, suggesting that upon *LEADR* silencing RWPE-1 cells strictly resemble frankly luminal cells (page 10, lines 18-23 – page 11, lines 1-5; Fig. 4h). More in general, the pathways that we found to be specifically enriched in true luminal vs basal cells (Fig. 5d) are the same that we found to be mostly modulated upon *LEADR* knock-down or knock-in (Fig. 5b), including inflammation-related gene sets (interferon etc...) but also *MYC* targets and G2M checkpoint genes. Overall, these data confirm that *LEADR* is involved in physiological basal-luminal differentiation.
- 2) We compared the propensity to differentiate of parental RWPE-1 cells with that of the same cells overexpressing constitutive *LEADR* or that of RWPE-1 cells genomically edited for *LEADR* using CRISPR-Cas9. To this purpose, we cultured the two *LEADR*-engineered clones and parental RWPE-1 cells in media with increasing differentiative potential (i.e. serum gradient) and check for differentiation at 3 days using qRT-PCR to measure cytokeratin switch (Fig. 4i, where acquisition of luminal phenotype is expressed as *KRT18/14* ratio). We found that CRISPRed cells were markedly more prone to luminal differentiation, whereas overexpressing ones were more refractory as compared to parental RWPE-1 cells, thus confirming a repressive role of *LEADR* against luminal differentiation (page 11, lines 5-10).
- 3) We also tested the impact of *LEADR* modulation in RWPE-1 cells through 3D acinar morphogenesis in matrigel. When grown as on-top 3D cultures (Tyson DR et al., Prostate 2007), parental cells started to form acinar-like structures, apparently organized in an outer monolayer of cells interacting with the extracellular matrix, and an inner core of more fused cells (**Fig. R5**). Though not forming a proper lumen (note that such cultures were made in K-SFM medium), such 3D structures strictly resemble organoids formed from human basal cells (Karthaus WR et al. Cell 2014) and recapitulate the phenotype of 3D cultures of RWPE-1 as from different studies (Bello-DeOcampo D et al., Prostate 2001; Wang M, et al J Cell Sci. 2017). *LEADR* overexpressing cells instead faced troubles in getting the correct out-in polarity, and tended to initially grow as single cells then as disorganized masses, probably due to an inherent difficulty to differentiate (**Fig. R5**). The other way around, CRISPRed cells tended to grow as fused cell masses from the very beginning, failing to form polarized acinar structures as those originating from parental cells (**Fig. R5**).

We decided not to include these data in the revised manuscript both for space constraints and for their preliminary nature. Optimization of 3D culturing/organoids is not trivial, especially setting up of conditions for immunofluorescence. Without any proper characterization of differentiation/polarity markers and due to the intrinsic interpretative issues, it is hence difficult to draw any solid conclusion about the effect of *LEADR* on

acinar morphogenesis. These results however are again suggestive of a direct involvement of *LEADR* in differentiation, which also impact on 3D organization.

Figure R5

b. In Fig. 4h, how many and what genes are included in basal- or luminal- specific gene set? Is there an overlap with LEADR signature?

This information was included in the previous version of the manuscript, in the Materials and Methods section: “Customly defined basal or luminal gene sets were obtained as the top-100 significantly differentially expressed genes in either condition, as from publicly available data sets”. To help the comprehension, we now modified the text as follows: “Customly defined basal or luminal specific gene sets were obtained selecting either the top-100 significantly down- or up-regulated genes in luminal vs basal cells, as from publicly available data sets” (page 51, lines 16-19). Such gene lists have been now provided as Supplementary Table S12.

As far as the overlap with *LEADR* signature is concerned, we already provided this information in form of gene set enrichment, rather than simple intersection of gene lists (Fig. 7 of the original manuscript, now Fig. 8; page 19, line 23 – page 20, lines 1-7), because we reasoned it could be more appropriate to take into account the intrinsic heterogeneity of data from different publicly available datasets, in terms of gene expression platform, dynamic range and experimental biases. GSEA actually calculates the association of a given geneset to one of the two biological states in a *t*-statistic comparison. Specifically, we assessed enrichment of *LEADR* signature in luminal vs basal cells as from different datasets, including mouse data and human tumors. Results showed that *LEADR*-signature was among the mostly enriched genes in human prostate luminal cells (where expression of *LEADR* is low) (Fig. 8b), hence subtending a role as master regulator of

differentiation. Strikingly, *LEADR* direct target genes were enriched also in prostate luminal cells in the mouse, where *LEADR* is not conserved (Fig. 8b). A broad relevance of the differentiation program regulated by human *LEADR*, which possibly emerged in primates to guarantee a more robust mechanism of control of the pathway, may be consequently envisaged. Enrichment of ‘*LEADR*-signature’ was also found in both prostate and breast carcinomas (Fig. 8c), indicating that aberrant *LEADR* function (or of downstream mediators) may play a role in tumorigenesis.

5. *Figure 5: LEADeR controls genes of the interferon signaling.*
a. Were these RNA-Seq experiments? How many biological replicates were used? Where was the RNA-Seq QC data?

We apologize for the misunderstanding. Gene expression of prostate cells modulated for *MIR205HG/LEADR* was assessed by microarray on Illumina HumanHT-12 v4 arrays. The silencing of *MIR205HG/LEADR* obtained with the different strategies described in the paper (si*LEADR*, gap*LEADR*, gap*INT1*) was conducted in triplicate (3 independent biological replicates, each including control and *LEADR*-specific oligomer) for subsequent gene expression analysis. Overexpression experiments with wild type *LEADR* (RefSeq, 7062 and whole genomic sequence ‘gene’) were conducted in quadruplicate (each including empty and *LEADR*-specific vector), whereas overexpression of *Alu*-deleted form in triplicate. These details are now included in the Methods section (page 50, lines 15-22 - page 51, lines 1-2).

For bioinformatic analyses, all raw data were \log_2 -transformed and normalized using the robust spline method implemented in the *lumi* package. Normalized data were filtered removing probes with neither at least one detection p -value < 0.01 across samples, nor associated official gene symbol; for probes mapping on the same gene symbol, the one with highest variance was selected. This information is included in the Methods section (page 51, lines 3-8). The pipeline as well as both raw and normalized data are also available at Gene Expression Omnibus, with accession number GSE104003 (token for referees provided in the cover letter).

b. Fig. 5a: The two different cell types should be indicated above. Going with consistent changes in interferon response, there are a lot of inflammatory pathways that got changed, which should be mentioned/discussed in slightly greater details.

Cell lines have been now indicated in the magnification of the heatmap (current Fig. 5b).

To answer about the relevance of other inflammatory pathways, we extracted leading edge genes from INTERFERON ALPHA and GAMMA RESPONSE (merged together), INFLAMMATION, COMPLEMENT, IL2 SIGNALING&IL6 SIGNALING (merged together), TNFA SIGNALING and intersected gene lists as depicted in the Venn diagram (**Fig. R6, left**).

Figure R6

We found that overlap between each inflammatory pathway with interferon signaling genes was overall relatively high (average 46%, ranging from 30% of TNF- α to 67% of Complement), most of all if we consider that Hallmark collection has been created to reduce the noise redundancy of functional annotations. For example, the leading edge gene shared by all gene sets is IL6, also referred to interferon- β 2. This would indicate that though GSEA performed using Hallmarks gene sets seems to reveal enrichment of different inflammatory pathways, analysis of leading edge genes suggests specific modulation of genes related to interferon.

Stimulated by the referee, we performed a preliminary experiment to test whether inflammatory stimuli other than interferons may induce basal cells to differentiate. To this purpose we treated RWPE-1 cells with TNF- α (which was the gene set with minor overlap) and found that this induced cytokeratin switch, but also increased interferon-related IRF7 factor (**Fig. R6, right**). This again suggests from one side that pathways enriched upon *LEADR* modulation are actually involved in differentiation, from the other that such inflammatory pathways may all converge onto interferon signaling to drive differentiation. This has not been included in the revised manuscript for space constraints.

c. Fig. 5d-f: Again biological data demonstrating IFN signaling with luminal differentiation was provided in only one cell line and not convincing.

In the revised manuscript, the capability of interferon to induce luminal differentiation has been widely demonstrated on primary epithelial cells (page 12, lines 6-16).

Results mainly confirmed our findings previously observed in RWPE-1 cells, showing that stimulation with interferon- β is able to induce luminal differentiation, as evidenced by:

- changes in cell morphology (Fig. 5e);
- cytokeratin switch, assessed by qRT-PCR (Fig. 5f) and western blotting (Fig. 5g);
- induction of AR expression (assessed by western blotting, Fig. 5g, and immunofluorescence Fig. 5h), which is evident upon treatment with interferon only but is markedly enhanced upon concomitant stimulation with DHT (Fig. 5h);
- increased PSA secretion, which becomes enhanced upon concomitant stimulation with DHT (Fig. 5i).

Additional indirect evidence of interferon involvement in normal differentiation program came from an experiment where PrEC cells were let to differentiate in a medium containing from 2 to 5% serum. In parallel to the acquisition of a more luminal phenotype, as evidenced by clear cytokeratin switch, cells actually underwent a marked induction of the interferon-related gene IRF7 (the same gene we found upregulated upon both *LEADR* silencing and interferon stimulation of RWPE-1 cells) (Fig. 5k). Interestingly, *LEADR* levels were reduced together with luminal differentiation (Fig. 5k).

6. Figure 6: *LEADeR* directly binds promoters of genes having an *Alu* and *IRF* binding site motif in tandem.

a. Data in Fig. S3b should appear in the main figure.

The data have been moved into main figure Fig. 5a and 5b.

c. Fig. 6c was not cited in Text.

This information was included in the previous version of the manuscript, in the results section: "...*LEADeR* deleted for a 100-nt long portion of the *Alu* element containing these motifs (*LEADR-ΔAlu*) was impaired in the capability of regulating gene expression (Supplementary Fig. S3b), especially of '*LEADR*-core up' (Fig. 6c) and interferon genes..."

Now the text has been rephrased (page 13, lines 18-21) and all new figure panels cited accordingly (including Fig. 6c).

Reviewer #2 (Remarks to the Author):

In this manuscript Profumo and colleagues investigate the long noncoding transcript LEADeR, which is generated from the same locus as miR-205, however has a function independent of the microRNA. They show that LEADeR is a nuclear lncRNA that regulates the differentiation of human prostate basal cells. They investigate the mechanism, finding that LEADeR binds to Alu elements, which are present at the promoters of interferon responsive genes. LEADeR inhibits the expression of those genes to prevent luminal differentiation.

This is overall an interesting and well-performed study. However some points need further clarification or reinforcement in order to support the conclusions.

1. *The first point refers to the decision between miR-205 and LEADeR generation. The expression levels of the different isoforms in the cell lines and the datasets analyzed are not clear. To get an idea of are relative expression levels of each isoform in different tissues more information can be extracted from datasets such as GTEx portal. In addition it would be useful to get the data of the transcript structures from most recent annotations or datasets such as the one generated with PacBio technology available at: <https://public.docs.crg.es/rguigo/CLS/>*

We thank the reviewer for the useful suggestions. We now manually curated re-annotation of all possible *LEADR* transcripts based on transcript structures from most recent annotations, including those of a genome-wide high-resolution remapping of pri-miRNAs by Mendell's lab (Chang TC et al., *Genome Research*, 2015) and data of targeted RNA capture with third-

generation long-read sequencing (Lagarde J et al., *Nature Genetics*, 2017), as from https://public_docs.crg.es/rguigo/CLS/ (Supplementary Fig. S3e). The latter have been used as reference to shortlist *bona fide* LEADR transcript configurations as described in the Results section (page 7, lines 1-17). This led to the identification of 9 high confidence LEADR transcripts, characterized by the alternative assembly of 4 modules: exon-1/2 (present in all transcripts with or without retention of intron); exon-3 (present in all transcripts in short or long version); exon-4 (missing in some transcripts, including historical RefSeq); two alternative terminal exons, the canonical *miR-205*-incompatible ex-5.1 and the downstream ex-5.2 (Table in Fig. 2a, where isoforms are ranked based on abundance in prostate basal cells).

We analyzed absolute abundance and relative percentage of each of such transcript isoforms in prostate basal vs luminal cells (GSE67070); normal vs tumor tissues (GSE22260) and commercially available normal vs tumor cells (data from GSE75035 and GSE25183).

We found that

- 1- As already observed when looking at global LEADR expression levels, all LEADR isoforms are more abundant in basal vs luminal cells (Fig. 2e, *left*), in normal vs tumor tissues (Fig. 2e, *left*) and in commercially available normal vs tumor cells (Supplementary Fig. S3f). Specifically, in tumor cell lines, expression of all LEADR isoforms approximates to zero. In the text, page 7, lines 18-21.
- 2- Most abundant transcripts in basal cells revealed to be 7062 and 7057 (Fig. 2e), the latter having the same exon composition as the historical RefSeq (Fig. 2a). In the text, page 7, lines 21-22.
- 3- No major differences in relative isoform expression were observed throughout the analyzed samples (Fig. 2e, *right*), nor between the cumulative fraction of *miR-205*-incompatible and compatible transcripts, which averagely accounted for 97% and 3% (Fig. 2f). In the text, page 7, line 23 - page 8, lines 1-2.

2. To further support the proposed mechanism of differential splicing it may be useful to analyze DGCR8-CLIP-seq or Drosha-CLIP-seq datasets to find out where Drosha is binding LEADeR. This piece of information will support the proposed mechanism of LEADeR's differential exon usage based on Drosha's binding and masking of a splicing site.

To answer this point, we took advantage of GSE61979 dataset including results of DGCR8-CLIP-seq and DROSHA-CLIP-seq performed on human ESCs. Analysis showed specific DGCR8 and DROSHA peaks covering LEADR ex-4/5.1 splice site (Fig. 2i), actually confirming that RNase III complex is able to bind LEADR primary transcript in this region and mask the *miR-205*-incompatible splicing site. For comparison, CLIP data for *miR-26b* and *miR-877*, representative of an intronic miRNA processed by Drosha and a Drosha-independent mirtron, respectively, are shown in Supplementary Fig. S3h. In the text, page 8, lines 15-18.

3. The authors select genes regulated by LEADeR based on the gene expression changes observed upon inhibition or overexpression of LEADeR. Can the enforced expression of LEADeR reduce or revert the differentiation of the cells? What is the phenotype of LEADeR-overexpressing cells?

Overexpression experiments reported in the manuscript were mainly set up to answer questions related to LEADR transcriptional program, rather than to assess reversion of the differentiated phenotype. To this purpose, we used DU145 tumor cells, which do not express measurable levels of either LEADR or *miR-205*, and observed that LEADR replacement induced transcriptional

changes that were consistent (though with opposite trend) with silencing experiments in RWPE-1 cells.

Phenotypically, overexpression of *LEADR* in DU145 cells was not sufficient to induce acquisition of frank basal features (only marginal reduction of *KRT8/18*, as assessed by qRT-PCR as cytokeratin switch, *not shown*), but this is not surprising as these cells are not normal luminal cells, rather tumor cells with fully transformed phenotype. Prostate cancer cells are usually referred to have luminal phenotype based on the absence of basal markers, such as *KRT5/14* or p63. For example, cycle threshold for these genes is over 40, likely suggesting that in these cells such genes are epigenetically repressed so that the sole *LEADR* overexpression, though repressing its direct targets (mainly interferon genes), is not sufficient to ultimately restore expression of basal cytokeratins or p63.

As for *LEADR* enforced overexpression in RWPE-1 cells, we found that it did not markedly increase basal cell phenotype, but we should consider that these cells are characterized by very high endogenous *LEADR* and basal cytokeratin levels. However, *LEADR* overexpression effect became evident when *LEADR*-overexpressing cells were induced to differentiate (page 11, lines 5-10). Specifically, we comparatively challenged the differentiation propensity of parental RWPE-1 cells with that of the same cells overexpressing constitutive *LEADR* or that of RWPE-1 cells genomically edited for *LEADR* using CRISPR-Cas9. To this purpose, we cultured the two *LEADR*-engineered clones and parental RWPE-1 cells in media with increasing differentiative potential (i.e. serum gradient) and checked for differentiation at 3 days using qRT-PCR to measure cytokeratin switch (Fig. 4i, where acquisition of luminal phenotype is expressed as *KRT18/14* ratio). We found that CRISPRed cells were markedly more prone to luminal differentiation, whereas overexpressing ones were more refractory as compared to parental RWPE-1 cells, thus confirming a repressive role of *LEADR* against luminal differentiation.

In the revised manuscript, we took advantage of CRISPRed RWPE-1 cells to show that CRISPR/Cas9 genomic deletion of the sequence spanning from exon 1 to 3 of *LEADR* (including TSS) results in the abrogation of both *LEADR* and *miR-205* expression (page 6, lines 10-12; Fig. 2b), again supporting the hypothesis that the two RNAs share the TSS and are produced starting from a common primary transcript.

4. A problem of the study is that the authors decide to focus on genes that are up-regulated when LEADeR is inhibited, which is arbitrary since the bias 64/36% is not strong enough evidence to support that LEADeR acts as a repressor. The authors have restricted the search for motifs in the up-regulated genes, and Alu sequences are found enriched in the promoters of the "LEADR-core up" genes. What sequence elements are found in the rest of the genes? How likely is it to find this type of sequences in random gene promoters?

We agree that the bias 64/36% may be not strong enough alone to support that *LEADR* acts as a repressor, but we observed that:

- The overlap between genes modulated upon silencing by the two different approaches (siRNA and gapmer) is greater (Fisher test $p=6.88e-08$ and $p=4.36e-12$ respectively, see Venn diagrams in Fig. 6a) among up-modulated (136/588 and 136/473, average fraction of 26%) than down-modulated genes (22/320 and 22/272, average fraction of 7.4%);
- The overlap with genes showing opposite trend upon *LEADR* overexpression is greater among up-modulated (38/136=28%) than down-modulated (2/22=9%) (see Venn diagrams in Fig. 6a).

Altogether these data suggest that *LEADR* direct targets are more likely to be repressed, which prompted us to initially focus on “*LEADR*-core up” genes. In the revised version of the manuscript (methods “motif discovery” section, page 48, lines 17-23 – page 49, lines 1-4), we now detail more on the reasons to focus on up-regulated genes.

However, to fulfill the referee’s request, we performed *de novo* motif discovery also on promoters of down-modulated genes. As we could not focus on “*LEADR*-core dn” genes (only 2), analysis was conducted on *LEADR*-gene set dn (n=22 genes) to make comparison possible with motif discovery run on *LEADR*-core up (n=38). Results show no substantial enrichment of motifs, as evidenced by extremely higher E-values (first 5 motifs with E-values ranging from 3.7e-30 to 4.8e-7, **Fig. R7**) compared to analyses run on *LEADR*-core up genes (first 5 motifs with E-values ranging from 4.8e-211 to 3.2e-82, Fig. 6b). In addition, none of the first three motifs showed sufficient matching with *LEADR* sequence (and the *Alu*), as from FIMO analysis (**Table R1**). This is in trend with analysis of recurrence of *Alu* (and IRF) sequences in the promoters of down-regulated genes, as reported in Supplementary Figure S5c, showing that *Alu* is not enriched.

Figure R7

Motifs on Down-regulated genes							
motif_id	start	stop	strand	score	p-value	q-value	matched_sequence on LEADR
MEME-1	177	205	+	15.098	3.07E-06	0.00523	ttttttttttctgacagggtctactt
MEME-3	595	635	+	5.657	4.15E-06	0.00592	cctcggcagccaccgcccaccaccgcccggccaccaccgta
MEME-2	124	152	-	9.051	2.27E-05	0.0289	AAGGAGAGGGAGTAAAGGTAGCTGGAAAA
Motifs on Up-regulated genes (as in the paper)							
motif_id	start	stop	strand	score	p-value	q-value	matched_sequence on LEADR
MEME-4	341	381	-	56.4388	8.80E-19	1.43E-15	GGCCGGATGCGGTGGCTCACGCCTGTGATCCCAACACTTTG
MEME-1	321	361	+	43.6327	3.68E-16	6.11E-13	ATCCACCTGCCTCGGCCTCCCAAAGTGTGGATCACAGGC
MEME-3	175	224	-	21.3571	6.93E-13	1.18E-09	CACTCCTGCCCGGCGACAAAGTGAGACCCTGTCAGAAAAAAAAAAAAAAG
MEME-2	203	231	-	33.0306	1.64E-12	2.75E-09	GCCACTGCACTCTGCCCGGCGACAAAG
MEME-5	235	263	-	29.7556	7.21E-11	1.19E-07	GGAGGTTGAGGCTGCAGTGAGCCCAAGAT

Table R1

As for the frequency of *Alu* elements in random gene promoters, we had already reported in the paper (previous Fig. 6d, now Fig. 7a) that:

- *Alu* elements are present in 61% (38+23%) of promoters genome-wide, as compared to 68% (55+13%) of “*LEADR*-core up” genes (not strikingly different likely due to the abundance of *Alu* sequences). Among the latter, however, 55/68 (81%) have *Alu*-IRF site combination, as compared to the random genome 38/61 (62%, $p=0.03$). Difference between 55% and 38% of *Alu*-IRF site combination vs all other possible element combinations is also significant ($p=0.02$, table in Fig. 7a), whereas 68% vs 61% is not, suggesting that though *Alu* sequence is essential for *LEADR* function, it may not be sufficient for target gene selection.

- when considering all 136 genes up-regulated in both siRNA and gapmer experiments, fraction of *Alu*-containing promoters is about 60% (“overall” bar in Fig. 7a, right. Again not significantly different from the general frequency in random gene promoters) but is 90% in the top-20 regulated genes and tends to decrease proportionally with target gene induction. This holds true also for *Alu*+IRF combination, again stressing the importance of IRF site for gene selection. These data however also point out that the presence of the *Alu* sequence in a target gene is important for real targeting in a way that was not evident from the fractions calculated in the previous point (consider that “*LEADR*-core up” genes are *bona fide* targets emerging from 3 different modulation experiments but are not ranked based on target repression/induction by *LEADR*).

5. *The co-occurrence of Alu elements and IRF sites in the promoter of the genes regulated by LEADeR is very interesting. However the mechanism underlying their regulation by LEADeR is not convincingly shown. The hypothesis that LEADeR binds to IRF1 protein doesn't seem to be supported by the weak RIP data.*

We performed additional experiments to confirm interaction between *LEADR* and IRF1 protein. Specifically, we carried out CLIP on UV-crosslinked cells and verified that, although washes are more stringent as compared to RIP, *LEADR* is again enriched over *SNORA74A* in IRF1-immunoselected sample as compared to negative antibody (page 16, lines 1-3; Fig. 7f). Such technique also indicates that *LEADR*/IRF1 interaction may be direct. We aimed to provide additional evidence of such interaction using an RNA-centric method (RIP and CLIP are instead protein-centric methods). In this regard, we performed an RNA-pulldown assay where biotinylated *in vitro* transcribed *LEADR* (*EGFP* transcript, which has similar length, was used as control) was mixed with cell lysate and proteins interacting with biotinylated RNA were then recovered through streptavidin-coated magnetic beads. Immunoblotting showed the increased abundance of IRF1 among proteins precipitated together with *LEADR*, as compared to *EGFP* transcript or beads only sample (page 16, lines 4-5; Fig. 7g).

6. *It has been recently reported that Alu repeats drive lincRNA nuclear retention. In the same line a cell fractionation experiment can be performed to see if LEADeR's localization to chromatin changes when the Alu sequence is not present.*

We did not observe any major change in nuclear vs cytoplasmic localization of *LEADR* upon deletion of the *Alu* element, as shown in DU145 cells upon ectopic replacement of the wild type or *Alu*-deleted form of the lincRNA (page 13, lines 21-23 – page 14, lines 1-2; Fig. 6d). This would suggest that *Alu* may not be a nuclear retention signal for *LEADR*. In this regard, we must however point out that we deleted only the portion of *Alu* where the most significant motifs were located, and confirmed such portion as essential for *LEADR* function. We cannot exclude that other portions of the whole *Alu* sequence may be relevant for nuclear localization.

7. *Where are the ChIRP probes mapping? Because the homology of sequence between LEADeR and the Alu elements, the enrichment can be due to the direct recognition of the DNA sequences by the probes. This technical considerations are very important when designing and interpreting ChIRP experiments.*

All ChIRP probes map outside of Alu sequence. We now specified this also in the Methods section (page 46, line 23 – page 47, line 1).

8. *What is the effect of LEADeR knockdown or overexpression on IRF binding? This should be shown by ChIP on a large set of gene targets, including positive and negative controls.*

We had initially shown that IRF1 is not substantially bound to target gene promoters in RWPE-1 cells under basal conditions, i.e. when they express high LEADR levels and LEADR is bound to such promoters (ChIRP data, Fig. 7d). We now show that upon LEADR silencing in RWPE-1 cells, IRF1 occupancy increases at all tested promoters compared to an unrelated genomic region (used to normalize data) (page 16, lines 5-7; Fig. 7h). The other way around, in DU145 cells, LEADR overexpression results in the displacement of IRF1 from target gene promoters (page 16, lines 7-9; Fig. 7i), especially from those having Alu+IRF site combination. Overall these data support a model where as soon as LEADR is bound to Alus, interaction of IRF1 protein with its binding site is somehow inhibited and gene expression repressed (poised state). Upon LEADR displacement, IRF1 has instead access to its binding site and can stimulate transcription of target genes.

9. *To further explore the role of the sequence elements, different mutants could be made with CRISPR/Cas9 on the endogenous promoter of one of the targets to test the effect on their regulation by LEADeR.*

To characterize the physiological role of LEADR in regulating its target genes, we employed CRISPR-Cas9 technology to specifically delete Alu element in the IFIT3 promoter. By transfecting double gRNAs, we could detect the Alu deletion on the bulk population by PCR (as depicted in **Fig. R8**). However, we noticed a relatively low efficiency of genome editing, as compared for example to editing of LEADR sequence performed with the same method (**Fig. R8** and included plot; Supplementary Fig. S3d). This observation can be explained by two different hypotheses: *i*) the selected gRNAs were not efficient in cutting the region of interest or *ii*) Alu deletion or IFIT3 promoter editing could be detrimental for the cells. In support of the latter hypothesis, we derived 15 different single cell clones and none was actually carrying Alu deletion (not even heterozygous). An inducible approach could provide insight into this hypothesis, but surely it represents a time consuming strategy that we could not adopt for reviewing this manuscript. It is also to consider that RWPE-1 cells, though immortalized, required long time for single cell clone selection.

Figure R8

Minor comments

Figure 1

1. *Figure 1a. In the RepeatMasker track, show which types of repeating elements are displayed.*
 The figure has been modified accordingly and provided as Supplementary Fig. S1 to comply with other referees' request to make all tracks explicit and move part of Fig. 1 to online material. In any case, exact position of SINE/Alu element on *LEADR* is also displayed in Fig. 6b.

2. *Figure 1b,c. Complement with miR-205 expression in the same samples. For Figure 1c, which tumor types have a significant difference in LEADeRs expression comparing tumor to normal? Show the p-values.*

Illumina body map data portal does not provide information on *miR-205* expression. So we interrogated TCGA to obtain data of both *LEADR* and *miR-205* in all normal tissues and corresponding tumors (mean + standard deviations), and reported it in Supplementary Fig. 2a (note that only cancer types with at least 100 tumor and 3 normal samples with available *MIR205HG* and *miR-205* expression were considered for the analysis; n size reported in Supplementary Table S1). The analysis confirmed coherent expression of the two RNAs across samples, as also made evident by correlation shown in Supplementary Fig. 3a.

Fig. 1c was redrawn considering only tumors with significant difference ($FDR < 0.05$) in *LEADR* expression between tumor and normal specimens. For these samples, *miR-205* expression has been also added. Note that for all tumors where *LEADR* is differentially expressed as compared to normal counterparts, also *miR-205* is differentially expressed and in the same direction (except for STAD and LIHC where it is not differentially expressed).

3. *Figure 1f. The data set title is missing (GSE3998)?*

Data set title has been added in Fig. 1f and indicated on top of each figure reporting analysis of publicly available dataset whenever appropriate.

Figure 2

1. Figure 2f. Results from siDROSHA2 and quantification of the bands (plot as %)

Results from siDROSHA2 are now shown (current Fig. 2h) and quantification of the bands reported as ratio between upper and lower band in the bar plot (Fig. 2h, bottom).

Figure 4

1. Figure 4f. Do you see the same change in localization of p63 using gapmers? If so, then add an extra figure panel of p63 immunofluorescence with gapCTR and gapLEADER.

Immunofluorescence showing change in localization of p63 using gapmers is provided as Fig. 4d, bottom.

Reviewer #3 (Remarks to the Author):

Both small and long non-coding RNAs play important roles during cell differentiation and tumorigenesis. The authors characterized the function of the lncRNA, LEADeR, which is the host gene of microRNA miR-205, in normal prostate epithelial cells and prostate cancer cell line. They provided evidence suggesting that LEADeR may bind to Alu elements which are located in the promoter regions of certain genes. Meanwhile, LEADeR may also bind to IRF binding site and may inhibit IRF activity. The topic is timing and the finding is interesting. It may provide new insight into prostate cancer biology. However, the mechanism studies are still preliminary. The authors need to experimentally address the follow questions:

Major:

The binding between LEADeR and IRF: The authors performed RNA-IP using IRF antibody suggesting that LEADeR may be associated with IRF, however, the result is need to be further confirmed by RNA-pulldown following by qRT-PCR using labeled RNAs. In addition, the RNA-IP was performed in native condition, it only suggests that LEADeR may be associated with IRF protein. They should demonstrate if this interaction is direct binding (or mediated by other proteins, i.e. indirect binding). Finally, the experiments should also be perform to test if knockdown or overexpression of LEADeR reduces the LEADeR-IRF interaction. The binding between LEADeR and Alu: Same as the above comments, more validation and functional experiments (e.g. knockdown or overexpression of LEADeR following by ChIRP) should be performed.

We agree with the referee regarding the need of validating the data on LEADR/IRF interaction. To this purpose, as suggested, we carried out CLIP on UV-crosslinked cells and verified that, although washes are more stringent as compared to RIP, LEADR is again enriched over SNORA74A in IRF1- immunoselected sample as compared to negative antibody (page 16, lines 1-3; Fig. 7f). This indicates that LEADR/IRF1 interaction is true and may be direct. We aimed to provide additional evidence of such interaction using an RNA-centric method (RIP and CLIP are instead protein-centric methods). In this regard we performed an RNA-pulldown assay where biotinylated *in vitro* transcribed LEADR (EGFP transcript, which has similar length, was used as control) was mixed with cell lysate and proteins interacting with biotinylated RNA were then recovered through streptavidin-coated magnetic beads. Immunoblotting showed the increased

abundance of IRF1 among proteins precipitated together with *LEADR*, as compared to *EGFP* transcript or beads only sample (page 16, lines 4-5; Fig. 7g).

As for the referee's request to test *LEADR/IRF* or *LEADR/Alu* interaction after modulation of *LEADR* expression, our opinion is that modulation of *LEADR* may bias the results of RNA-centric methods, such as ChIRP or RNA pulldown, by increasing/decreasing total *LEADR* levels. For example, upon *LEADR* silencing, global *LEADR* levels are markedly reduced. This means that complementary probes would retrieve less *LEADR* and consequently less *LEADR*-associated DNA in ChIRP experiments mainly due to absence/lower abundance of *LEADR* itself. This would be similar to the lacZ control sample, and would not really test changes in *LEADR* binding to the DNA. Similarly, upon precipitation of IRF transcription factors by RIP or CLIP (protein-centric methods), qRT-PCR measurement of associated *LEADR* would result artifactually affected by globally reduced *LEADR* levels.

An experiment that instead could test if *LEADR* modulation may impact on *LEADR/IRF* axis is chromatin immunoprecipitation to assess IRF binding to the DNA when *LEADR* is overexpressed or silenced. In this regard, we had initially shown that IRF1 is not substantially bound to target gene promoters in RWPE-1 cells in basal conditions (see gapCTR lane in Fig. 7h), i.e. when they express high *LEADR* levels and *LEADR* is bound to such promoters (ChIRP data, Fig. 7d). We now show that upon *LEADR* silencing in RWPE-1 cells, IRF1 occupancy increases at all tested promoters compared to an unrelated genomic region (used to normalize data) (page 16, lines 5-7; Fig. 7h). The other way around, in DU145 cells, *LEADR* overexpression results in the displacement of IRF1 from target gene promoters, especially those having Alu/IRF site combination (page 16, lines 7-9; Fig. 7i). Overall these data support a model where as soon as *LEADR* is bound to *Alus*, interaction of IRF1 protein with its binding site is somehow inhibited and gene expression repressed (poised state). Upon *LEADR* displacement, IRF1 has instead access to its binding site and can stimulate transcription of target genes.

The function of miR-205: Although the function of miR-205 has been well-characterized, how miR-205 cooperates with LEADeR during differentiation should be carefully characterized in this study, given that miR-205 and LEADeR are in the same transcription unit and their expression levels are significantly correlated.

We agree with the referee that characterizing how *LEADR* and *miR-205* specifically cooperate in maintaining the basal phenotype as well as how changes in their expression contribute to differentiation would be very interesting. However, we think this is out of the scope of this paper, where we dissected for the first time the expression pattern, coding/non-coding nature, biological role and mechanism of action of *LEADR* (together with insights into its transcript structures and *miR-205* biogenesis). Future investigations will clarify the exact contribution of the two RNAs in controlling differentiation of basal cells. Nonetheless, a significant piece of information has already been collected by us in this study and in previous work.

As for *miR-205*, we already showed its specific contribution to the maintenance of the basal phenotype through regulation of basement membrane components (*Gandellini et al. Cell Death Differentiation* 2012). In that paper, we also claimed that modulation of *miR-205* alone did not induce any frank cytokeratin switch in basal cells. Here we showed that *LEADR* is instead able to regulate cytokeratin switch (Fig. 4b-c), through regulation of the interferon pathway. Consistently, as appreciable from the violin plot shown in Fig. 5c, knockdown of *miR-205* only

does not significantly affect expression of interferon genes, thus justifying lack of effect on cytokeratin switch.

To respond to the referee's request, we performed an additional experiment where *miR-205* only was abrogated in RWPE-1 cells through an LNA-modified antisense oligomer (LNA205) and actually confirmed the data of *Cell Death Differentiation* paper, with reduction of laminins *LAMA3* and *LAMB3* (i.e. the main components of prostate basement membrane) despite no frank basal to luminal cytokeratin switch (**Fig. R9**). As expected, knockdown of *LEADR* (shown to promote overt cytokeratin switch for the first time in current manuscript) did not markedly change laminin expression (**Fig. R9**). Simultaneous abrogation of *miR-205* and *LEADR* obtained by using intronic oligomer (gapINT1) induced changes in both cytokeratins and laminins, suggesting cooperation and complementation between the two RNAs in maintaining the basal phenotype (**Fig. R9**).

Figure R9

We did not include these data in the revised manuscript for space constraints (mainly word count), as well as because data on *miR-205* knockdown were already reported in our previous paper Gandellini et al. *Cell Death Differentiation* 2012).

However, similar information on the distinct but complementary functions of *LEADR* and *miR-205* was already present in Figure 4h as from gene expression profiling analysis (page 10, lines 18-23 - page 11, lines 1-5).

We derived basal- and luminal-specific gene sets from gene expression data of true luminal and basal cells isolated from normal prostate (4 different datasets). When tested on the transcriptome of *LEADR*-knocked down RWPE-1 cells, we found a tendency for positive enrichment for luminal and negative enrichment for basal gene sets, suggesting that upon *LEADR* silencing RWPE-1 cells start resembling true luminal cells (Fig. 4h). This is in trend with all phenotypic characterizations reported in the paper, showing cytokeratin switch, increased AR expression and enhanced PSA secretion in basal cells silenced for *LEADR*. In Fig. 4h we showed that this is not the case of *miR-205*-knocked down cells, where GSEA does not account for the acquisition of a frank luminal phenotype (modulation of both basal and luminal gene sets despite reduction of laminins).

To give a global measure of luminal differentiation in each experiment, we now calculated $\Delta NES_{\text{luminal-basal}}$ for each publicly available dataset, then averaged $\Delta NES_{\text{luminal-basal}}$ through all

datasets resulting in the following plot (added at the top of Fig. 4h; p -values reported in **Table R2**).

	$\mu \neq 0$	$\mu \neq \mu_{\text{INT1}}$
LNA-205	$p = 0.5607$	
siLEADR	$p = 0.0096$	0.0042
gapLEADR	$p = 0.0147$	0.0075
gapINT1	$p = 0.0010$	

Table R2

Results show no significant trend towards luminal phenotype upon silencing of *miR-205* only (LNA205). In contrast, luminal differentiation is observed in both siLEADR and gapLEADR experiments but becomes extremely evident (and significantly increased as compared to LEADR silencing only) when using gapINT1, which abrogates both *miR-205* and LEADR. This suggests that both RNAs need to be lost to allow frank luminal differentiation, and, the other way around, that both are essential for maintenance of basal phenotype.

Minor:

Figure 1, 2 and 3 are not informative. These figures could be combined to one figure, or some panels in the figures could be used as online figures.

We modified the figures by taking into account all referees' suggestions. Some panels have been moved to online figures.

More detailed information about how to analyze array experiments should be provided. What is the cut-off and statistical method? How to choose the cut-off.....

We provided more detailed information on gene expression experiments and bioinformatic pipeline in the materials and method section (page 50, lines 15-22; page 51, lines 1-14), as follows:

Gene expression analysis of cells modulated for LEADR expression

Transcriptomic data from prostate cells modulated for *MIR205HG/LEADR* expression using different strategies were generated in our laboratory using Illumina HumanHT-12 v4 arrays, as previously described (62). The silencing of *MIR205HG/LEADR* obtained with the different strategies described in the paper (siLEADR, gapLEADR, gapINT1) was conducted in triplicate (3 independent biological replicates, each including control and LEADR-specific oligomer) for subsequent gene expression analysis. Overexpression experiments with wild type LEADR (RefSeq, 7062 and whole genomic sequence 'gene') were conducted in quadruplicate (each including empty and LEADR-specific vector), whereas overexpression of *Alu*-deleted form in triplicate.

Raw data were \log_2 -transformed and normalized using the robust spline method implemented in the *lumi* package (63). Normalized data were filtered removing probes with neither at least one detection p -value < 0.01 across samples, nor associated official gene symbol; for probes mapping on the same gene symbol, the one with highest variance was selected.

Gene expression data and processing pipeline were deposited at Gene Expression Omnibus, with accession number GSE104003.

Bioinformatic analyses

Differential expression was estimated both in terms of fold change (FC) and t -value, using the *limma* Bioconductor package (64). Significance was provided in terms of False Discovery Rate (FDR) to take into account the adjustment for multiple hypotheses testing, using a threshold of 0.05. All these analysis were conducted in R environment.

What is the clinical significance of the RNA expression level of LEADeR in prostate and other cancer types? TCGA provides a novel resource to explore.

We utilized TCGA data to assess association between *LEADR* expression in prostate tumors and patient outcome, in terms of biochemical relapse (BCR)-free or overall survival. We stratified tumors based on *LEADR* median expression but neither log-rank nor Wilcoxon tests showed significance differences in outcome (for either endpoints) between the two groups. Kaplan-Meier curve for BCR-free survival has been included as Supplementary Fig. S2b (and cited in the text, page 5, lines 2-4). We did not include curve for overall survival in the manuscript because the number of events is too low. However the reviewer can see the curve as **Fig. R10**.

These results are not surprising, because loss of *LEADR*, which reflects luminal differentiation, is an early event in prostate carcinogenesis. In fact, analysis of available cohorts shows invariable reduction of *LEADR* in primary prostate tumors as compared to their normal counterparts. In GSE21034 dataset, reduction of *LEADR* is even evident in the lowest grade cancers (G6, Fig. 2d). Consistent with this, expression of *LEADR* alone is sufficient to correctly classify tumor from normal samples (Fig. 2e). Therefore it is likely that once the tumor is established (and *LEADR* lost), other factors may be more relevant to determine patient's prognosis. In the clinical setting, utility of *LEADR* may be hence restricted to diagnosis, which is supported also by its strict correlation with p63 (Supplementary Fig. S3a, *right*). Notably immunohistochemistry for p63 remains the gold standard for prostate cancer diagnosis.

Figure R10

As for the relevance of *LEADR* in other tumor types, we think it would be necessary to know the clinical aspects of each given tumor to correctly set up the analyses. We think this is out of the scope of this paper. However, some hints about the possible relevance in breast cancer are present in the manuscript, such as the down-regulation of *LEADR* (Fig. 1c, Supplementary Fig. 2a) as well as enrichment of *LEADR* signature (Fig. 8c) in breast tumors compared to normal counterparts.

References

Gandellini P, Profumo V, Casamichele A, Fenderico N, Borrelli S, Petrovich G, Santilli G, Callari M, Colecchia M, Pozzi S, De Cesare M, Folini M, Valdagni R, Mantovani R, Zaffaroni N. miR-205 regulates basement membrane deposition in human prostate: implications for cancer development. *Cell Death Differ.* 2012 Nov;19(11):1750-60. doi: 10.1038/cdd.2012.56. Epub 2012 May 4. PubMed PMID: 22555458; PubMed Central PMCID: PMC3469086.

Zhang L, Zhao W, Valdez JM, Creighton CJ, Xin L. Low-density Taqman miRNA array reveals miRNAs differentially expressed in prostatic stem cells and luminal cells. *Prostate.* 2010 Feb 15;70(3):297-304. doi: 10.1002/pros.21064. PubMed PMID: 19827049; PubMed Central PMCID: PMC3031866.

Sempere LF, Christensen M, Silahatoglu A, Bak M, Heath CV, Schwartz G, Wells W, Kauppinen S, Cole CN. Altered MicroRNA expression confined to specific epithelial cell subpopulations in breast cancer. *Cancer Res.* 2007 Dec 15;67(24):11612-20. PubMed PMID: 18089790.

Chang TC, Perteu M, Lee S, Salzberg SL, Mendell JT. Genome-wide annotation of microRNA primary transcript structures reveals novel regulatory mechanisms. *Genome Res.* 2015 Sep;25(9):1401-9. doi: 10.1101/gr.193607.115. PubMed PMID: 26290535; PubMed Central PMCID: PMC4561498.

Lagarde J, Uszczyńska-Ratajczak B, Carbonell S, Pérez-Lluch S, Abad A, Davis C, Gingeras TR, Frankish A, Harrow J, Guigo R, Johnson R. High-throughput annotation of full-length long noncoding RNAs with capture long-read sequencing. *Nat Genet.* 2017 Dec;49(12):1731-1740. doi: 10.1038/ng.3988. Epub 2017 Nov 6. PubMed PMID: 29106417; PubMed Central PMCID: PMC5709232.

Litvinov IV, Vander Griend DJ, Xu Y, Antony L, Dalrymple SL, Isaacs JT. Low-calcium serum-free defined medium selects for growth of normal prostatic epithelial stem cells. *Cancer Res.* 2006 Sep 1;66(17):8598-607. PubMed PMID: 16951173; PubMed Central PMCID: PMC4124600.

Karthus WR, Iaquina PJ, Drost J, Gracanin A, van Boxtel R, Wongvipat J, Dowling CM, Gao D, Begthel H, Sachs N, Vries RGJ, Cuppen E, Chen Y, Sawyers CL, Clevers HC. Identification of multipotent luminal progenitor cells in human prostate organoid cultures. *Cell.* 2014 Sep 25;159(1):163-175. doi: 10.1016/j.cell.2014.08.017. Epub 2014 Sep 4. PubMed PMID: 25201529; PubMed Central PMCID: PMC4772677.

Liu AY, True LD, LaTray L, Nelson PS, Ellis WJ, Vessella RL, Lange PH, Hood L, van den Engh G. Cell-cell interaction in prostate gene regulation and cytodifferentiation. *Proc Natl Acad Sci U S A.* 1997 Sep 30;94(20):10705-10. PubMed PMID: 9380699; PubMed Central PMCID: PMC23453.

Tyson DR, Inokuchi J, Tsunoda T, Lau A, Ornstein DK. Culture requirements of prostatic epithelial cell lines for acinar morphogenesis and lumen formation in vitro: role of extracellular calcium. *Prostate*. 2007 Nov 1;67(15):1601-13. PubMed PMID: 17705248.

Bello-DeOcampo D, Kleinman HK, Deocampo ND, Webber MM. Laminin-1 and alpha6beta1 integrin regulate acinar morphogenesis of normal and malignant human prostate epithelial cells. *Prostate*. 2001 Feb 1;46(2):142-53. PubMed PMID: 11170142.

Wang M, Nagle RB, Knudsen BS, Rogers GC, Cress AE. A basal cell defect promotes budding of prostatic intraepithelial neoplasia. *J Cell Sci*. 2017 Jan 1;130(1):104-110. doi: 10.1242/jcs.188177. Epub 2016 Sep 8. PubMed PMID: 27609833; PubMed Central PMCID: PMC5394777.

Reviewers' comments:

Reviewer #1, Expertise: Cell differentiation (prostate luminal, basal), lncRNA (Remarks to the Author):

This represents a much-improved revision of the manuscript that characterizes a lincRNA that the authors re-annotated as LEADR, derived from the MIR205HG locus. The authors have made conscientious efforts in addressing most of this reviewer's earlier concerns. The paper is mechanistically driven and has a great significance in presenting a novel model of how a single genomic locus may generate an important miRNA (i.e., miR-205) and a lincRNA to coordinately regulate the prostate basal cell properties. Exciting data has also been presented to show how LEADR may regulate basal-luminal differentiation through genomic interactions with Alu/IRF elements and repression of IRF downstream genes. The paper has greatly improved. On the other hand, there are still many technical and formatting related issues/questions that need to be addressed prior to publication.

Main critique:

1. Figure 1:

- 1): Fig. 1a is still too blurry to read. May present it in the Supplementary in an enlarged form.
- 2). The y-axis in some box plots (panels d, f, i) should start from 0.
- 3): The publications (references) associated with the ALL GSE data sets (e.g., the 3 datasets in this figure) should be cited if available. It is recommended that the authors summarize all datasets used in the paper into a Supplementary table.

2. Fig. S1: Some labels on the left should be re-labeled for clarity. The y-axis in Fig. S1c should start from 0.

3. Figure 2:

- 1): Fig 2a: the positions of primers for LEADR gene should be indicated. Also, for each isoform, is it possible to not only show the ID for each isoforms but also include the transcript ID from either Ensemble or NCBI?
- 2): For data in panel d, it was stated that ".....gapmer oligonucleotides designed to target introns of.....a common primary transcript". The data shown does not actually support that conclusion: although it's true that both gapINT1 and gapINT2 abolished the transcription of LEADR, they did not abolish miR-205 expression.
- 3): Pie chart presented in Fig. 2f is inconsistent with the discussion in the text.
- 4): How was the absolute expression of isoforms called throughout the analyzed samples in Figure 2e (left)? What was each isoform normalized to when calculating relative expression in Figure 2e (right)? Would it be possible to indicate the proportion of each isoform in the pie chart (Figure 2f).

4. Figure 4:

- 1): Fig. 4b: Please use two different colors to indicate the two cell types.
- 2): In many experiments presented in this Figure (and associated supplementary data). gapLEADR displayed more apparent effects than siLEADR. However, in Fig. 4h, siLEADR appeared to induce more pronounced luminal gene expression changes than gapLEADR. How do the authors reconcile this discrepancy?
- 3): The radar plot in Fig. 4i is a bit unwieldy for readers and should be presented in a more self-evident format.

5. Figure 5:

- 1): Fig. 5a: What do those blue and red dots mean? This should be indicated in legend. Also, it might be helpful if the authors present a Venn diagram showing the overlaps in both up/down regulated genes in the two KD approaches.

2): Fig. 5f: Please use two different colors to indicate the two cell types.

2): In Fig. 5h (and Fig. 4f), why was AR staining mostly cytoplasmic?

6. Figure 6:

1): Fig. 6a: It's a bit surprising that most altered genes upon LEADR KD by siLEADR vs. gapLEADR did not overlap. Can the authors offer some explanations?

7. Figure 7:

1): In Fig. 7b, 10 of the 51 'LEADR-signature genes' are histone and histone variant genes. Authors should offer some insights on this interesting finding.

2): IRF-7 is strongly induced either by interrupting expression of LEADR gene or addition of IFN-beta (Fig. 5j). Both RIP and UV CLIP assays showed binding of LEADR to IRF7 and IRF1 (Fig. 7e, f). How come IRF1 did not appear in the 'LEADR-signature gene' (Fig. 7b) and IRF7 was not examined in the pulldown assays (Fig. 7g)?

3): Also, in Fig. 7g, data is not very convincing because EGFP pulldown also showed some IRF7 binding.

8. Figure 8: The GSEA plots in panels b and c should be remade/re-labeled for better and clearer presentations with NES, p, and FDR values indicated in each plot. Similar presentations go for other GSEA plots (e.g., Fig. S5b).

Minor points:

9. On page 38, it is stated that "Overexpression of MIR205HG/LEADR was performed by using a pCMV-6AC plasmid vector containing alternatively the whole genomic sequence, the RefSeq or 7063 transcript, as synthesized by OriGene". Similar statement was made on page 50, "Overexpression experiments with wild type LEADR (RefSeq, 7062 and whole genomic sequence 'gene') were conducted in quadruplicate (each including empty and LEADR-specific vector), ...".

The meaning of "whole genomic sequence" is still confusing. How long is it? Does it include the intron fragment that generates miR-205? Authors should indicate in each figure which transcript (7062, 7063 or 7057 etc) was used for overexpression of LEADR and/or miR-205.

10. For WB blots shown, full blot should be presented with clearly marked M.W markers indicated.

11. There are still many language issues such as the wording '....a global repressive attitude; page 11, second paragraph).

Reviewer #2, Expertise: lncRNA mediated regulation of gene expression (Remarks to the Author):

This revised version of the manuscript has improved by incorporating a number of changes.

However, the model where LEADeR binds to both DNA at Alu elements as well as IRF1 protein is not strongly supported by the data.

For instance, the authors claim that the interaction between LEADeR and Alu motifs is mediated by sequence complementarity between RNA and DNA. However at the same time they argue that the RNA is highly structured in this region. If the interaction involves extensive DNA-RNA base pairing, the RNA should be exposed in single stranded conformation. The structural prediction of LEADeR is not informative and does not support the model. In addition, the RNA pulldown data showing interaction between IRF1 and LEADeR (figure 7g) is quite poor. Prior publication the authors should either further reinforce or modify this part, leaving it as a speculative model.

Reviewer #3, Expertise: lncRNA ID/processing and cancer (Remarks to the Author):

My questions have been addressed, and the paper has been significantly improved.

Reviewers' comments:

We thank the reviewers for their efforts and constructive criticisms. We are glad to read that they all found our manuscript improved. Changes made in response to the reviewers are colored in red in the revised manuscript.

Reviewer #1, Expertise: Cell differentiation (prostate luminal, basal), lncRNA (Remarks to the Author):

This represents a much-improved revision of the manuscript that characterizes a lincRNA that the authors re-annotated as LEADR, derived from the MIR205HG locus. The authors have made conscientious efforts in addressing most of this reviewer's earlier concerns. The paper is mechanistically driven and has a great significance in presenting a novel model of how a single genomic locus may generate an important miRNA (i.e., miR-205) and a lincRNA to coordinately regulate the prostate basal cell properties. Exciting data has also been presented to show how LEADR may regulate basal-luminal differentiation through genomic interactions with Alu/IRF elements and repression of IRF downstream genes. The paper has greatly improved. On the other hand, there are still many technical and formatting related issues/questions that need to be addressed prior to publication.

We really appreciate the general comment of this reviewer regarding improvements of our manuscript, its significance, novelty, and mechanistic insight.

Main critique:

1. Figure 1:

1): Fig. 1a is still too blurry to read. May present it in the Supplementary in an enlarged form.

Thanks for the suggestion. We moved Fig. 1a in an enlarged form into Supplementary Fig. 1, of which we attempted to ameliorate readability as requested in point 2. Figure citations in the text have been modified accordingly.

2). The y-axis in some box plots (panels d, f, i) should start from 0.

We report here below the box plots with y-axis starting from 0 to show that no outlier data are present in the range between zero and the shown bars. We modified the plots in main figures accordingly.

3): The publications (references) associated with the ALL GSE data sets (e.g., the 3 datasets in this figure) should be cited if available. It is recommended that the authors summarize all datasets used in the paper into a Supplementary table.

Thanks for the suggestion. We summarized all publicly available datasets analyzed throughout our study in a new Supplementary Table 12. In the table, we reported details on the technique and samples of

each dataset, the related publication (with references), and the specific re-use we made in this study. The table is cited in the methods (page 34, lines 753-755).

2. Fig. S1: Some labels on the left should be re-labeled for clarity. The y-axis in Fig. S1c should start from 0.

Thanks for the suggestion. We created a color legend for chromatin state segmentation and for ENCODE cell lines to improve clarity. We added conservation as from previous Fig. 1a. We also modified the whole image to improve readability of labels (see Supplementary Figure 1).

As for y-axis of Supplementary Fig. 2c (we think the reviewer is referring to this figure), we modified as reported here below.

3. Figure 2:

1): Fig 2a: the positions of primers for LEADR gene should be indicated. Also, for each isoform, is it possible to not only show the ID for each isoforms but also include the transcript ID from either Ensembl or NCBI?

Thanks for the suggestion. We indicated the position for the primers generally used to detect *LEADR*, which span exon-2/exon-3 boundary that is shared by all putative transcripts. This is reported also in Supplementary Table 8 and specified in the methods (page 25, lines 545-547).

As for transcript ID, we used the IDs from capture long sequencing (CLS) project [Lagarde J et al. *Nat Genet.* 2017], which should represent the most comprehensive and potentially most precise collection of *LEADR* transcript configurations, as it enables the identification of full-length transcript models [Uszczynska-Ratajczak B, et al. *Nat Rev Genet.* 2018]. Starting from all possible CLS transcripts, we then shortlisted the 9 most confident ones using the approach described in the paper (page 6, lines 130-131; page 7, 132-135). Briefly, we retained the isoforms that have a TSS confirmed by CAGE experiments from ENCODE/RIKEN in prostate cells and are supported by results of a recent genome-wide high-resolution remapping of pri-miRNAs [Chang T et al. *Genome Res.* 2015] and/or by Genecode v28lift37 Basic annotation (reporting Ensembl transcripts). Supplementary Fig. 3e reports all putative transcript configurations and allows comparison between IDs from CLS project and Genecode/Ensembl transcripts. Notably, some CLS IDs are superimposable on already existing Genecode/Ensembl/NCBI configurations (such as 7057 with historical NCBI RefSeq NM_001104548 and 7064 with Ensembl ENST00000429156), others not. It must be noted however that discrepancies are just related to the fact that some Genecode/Ensembl transcripts are not complete at 5' end or do not account of all possible exon combinations, as instead do CLS transcripts. For all these reasons, we decided to retain original CLS IDs; only 7057 transcript has been referred to as RefSeq in the manuscript for clarity.

2): For data in panel d, it was stated that “.....gapmer oligonucleotides designed to target introns of.....a common primary transcript”. The data shown does not actually support that conclusion:

although it's true that both gapINT1 and gapINT2 abolished the transcription of LEADR, they did not abolish miR-205 expression.

The data reported in the figure actually show that both intronic gapmers are able to induce a statistically significant reduction of mature *miR-205* levels as compared to control gapmer, at day 3 after transfection. What could be surprising is that repression of processed *LEADR* transcripts is greater than that of *miR-205*. However, to interpret these data it could be taken into account that half life of short double-stranded RNAs, such as microRNAs, is much higher than that of long single-stranded RNAs (as is *LEADR*) [Marzi MJ et al. *Genome Res.* 2016]. It is hence presumable that, at that specific time point after transfection, we may still detect residual endogenous *miR-205* that was present in the cell before inhibition of nascent miRNA by cleavage of *LEADR/miR-205* primary transcript. In agreement with this, comparable residual *miR-205* levels were detected after inhibition of miRNA processing by siDrosha1/2 at the same time point (Fig. 2g). This evidence again confirms that once production of mature miRNA is interrupted at some point in the biogenesis pathway (in this case when *LEADR* primary transcript is cleaved by intronic gapmer or miRNA excision is prevented by inhibition of Drosha), residual levels of "old" mature miRNA can be still detected 3 days later. Accordingly, *miR-205* expression levels further diminish at later time points upon transfection with gapINT1, as depicted in the following plot (Fig. R1).

Figure R1

Finally, it is worth mentioning that retrotranscription method used to prepare cDNA for PCR assessment of *LEADR* and *miR-205* is different (as indicated in the methods, page 23, lines 516-521). Basically, *LEADR* is amplified from RNA retrotranscribed with random hexamers, which allows production of cDNA randomly from different RNAs, of which *LEADR* cDNA is just a fraction; in contrast, *miR-205* is amplified from cDNA retrotranscribed using specific primers for mature *miR-205*, which makes PCR more efficient and thus able to detect even lower target RNA amounts.

3): Pie chart presented in Fig. 2f is inconsistent with the discussion in the text.

The pie chart reported in Fig. 2f shows the cumulative percentage of *miR-205* compatible and incompatible transcripts, as averaged through all analyzed samples (i.e. normal and tumor tissues, basal and luminal cells, PrEC and RWPE-1 cells. Commercial tumor cells lines not used because they lack of *LEADR* expression. Samples included in the analysis have been also specified in Supplementary Table 1). Here below we report a chart with original data. We apologize for inconsistency and now correct both in the text (page 7, lines 150-152) and in the figure.

percentage	normal	tumor	basal	luminal	PrEC	RWPE-1	mean	sd
incompatible	96.5	97.0	99.6	98.6	96.1	97.9	97.6	1.3
compatible	3.5	3.0	0.4	1.4	3.9	2.1	2.4	1.3

4): How was the absolute expression of isoforms called throughout the analyzed samples in Figure 2e (left)? What was each isoform normalized to when calculating relative expression in Figure 2e (right)? Would it be possible to indicate the proportion of each isoform in the pie chart (Figure 2f).

We apologize if this was not clear. Absolute isoform expression corresponds to transcripts per million (TPM) in each sample, as described in the methods section: “RNA-Seq data from GSE67070, GSE22260, GSE75035 and GSE25183 were retrieved as sra files with sratoolkit tool and transformed in fastq paired files with fastq-dump –split-3 command. RSEM package was used to construct the reference with rsem-prepare-reference –no-polya and then to calculate expression of isoforms with rsem-calculate-expression –paired-end”. TPM is the output of RSEM tool and was used to compare the abundance of transcripts across samples.

TPM data for each transcript were hence plotted to show that all LEADR isoforms are more abundant in basal than luminal cells (Fig. 2e, left), in normal than tumor tissues (Fig. 2e, left) and in commercially available normal than tumor cell lines (Supplementary Fig. 3f).

In Fig. 2e, right we actually reported the proportion of each isoform for each sample (i.e. each isoform normalized to the total isoforms in that given sample and expressed as percentage, as requested by this reviewer) to show that though abundance of LEADR across samples may vary, there is no substantial change in the proportion of each isoform.

The plot is equivalent to a pie chart, as we make explicit here below for basal and luminal cells (Fig. R2).

Figure R2

We preferred to use this type of chart to make it more comparable with absolute expression data of Fig. 2e, left and better evidence that isoform proportion remains the same regardless of changes in overall LEADR levels across samples. To avoid misunderstanding, we specified in the figure legend (page 46, lines 1024-1025) that fig. 2e, right reports proportion of each isoform on the total isoform amount, expressed as percentage. We also changed y-axis range to 100% to make this self-evident.

For the reviewer only, in the pie charts of Fig. R2 we also indicate the cumulative percentage of miR-205 compatible and incompatible transcripts in the shown samples, to better explain how this information was then used to draw the pie chart of Fig. 2f (see answer to point 3.3).

4. Figure 4:

1): Fig. 4b: Please use two different colors to indicate the two cell types.

The two cell types were already indicated with two different colors (green tones, because we originally used green for basal cells) but we agree with the reviewer's suggestion and chose more contrasting colors to improve clarity.

2): In many experiments presented in this Figure (and associated supplementary data). gapLEADR displayed more apparent effects than siLEADR. However, in Fig. 4h, siLEADR appeared to induce more pronounced luminal gene expression changes than gapLEADR. How do the authors reconcile this discrepancy?

We measured the global luminal transition induced by each *LEADR* manipulation by calculating the average $\Delta NES_{\text{luminal-basal}}$ across all analyzed datasets, as reported in the bar plot in top of this figure. Results showed that gapmer and siRNA induced superimposable transition towards luminal phenotype (only gapINT1 induced a more profound effect), though apparently the siRNA seemed to up-regulate luminal genes whereas gapmer seemed to down-regulate basal ones.

However, this is just a bioinformatic artifact. GSEA actually is run on genes ranked for t-value in each comparison (siLEADR vs siCTR and gapLEADR vs gapCTR). Obviously each experiment is characterized by a different ranking in terms of gene expression (due to several factors, including variability, modulation of off-target genes...see also answer to point 6.1), which results in a different distribution shape of the analyzed gene sets inside the rank and may lead to low statistical significance. In the figure, we rigorously plotted only the NES measures with a significance value of $FDR < 0.05$, thus leading to the apparent discrepancy. From a functional point of view however, results are equivalent, as down-regulation of basal genes or up-regulation of luminal ones have the same biological significance, most of all if these gene sets are customly defined from the comparison between luminal and basal cells (i.e. what is up in luminal cells is down in basal cells and viceversa).

3): The radar plot in Fig. 4i is a bit unwieldy for readers and should be presented in a more self-evident format.

We are sorry for this and modified the radar plot into a graph where luminal transition (summarized by the ratio between luminal KRT18 and basal KRT14) is depicted using a cone. Higher is the cone, higher is the ratio and luminal phenotype. Now it should be more self-evident that, upon serum gradient, CRISPRed clone tends to differentiate more easily than parental cells, whereas *LEADR*-overexpressing cells are more refractory (e.g. in these cells we observe an evident increase of the ratio only upon 5% serum stimulation).

5. Figure 5:

1): Fig. 5a: What do those blue and red dots mean? This should be indicated in legend. Also, it might be helpful if the authors present a Venn diagram showing the overlaps in both up/down regulated genes in the two KD approaches.

Blue and red dots are just a graphical representation of the percentage of up- and down-regulated genes, respectively, upon *LEADR* silencing, with every dot being a 1 % of genes. We now explained this in the legend (page 49, lines 1080-1081).

Regarding the Venn diagram showing the overlaps in both up/down regulated genes in the two KD approaches, this is reported in Fig. 6a.

2): Fig. 5f: Please use two different colors to indicate the two cell types.

The two cell types were already indicated with two different colors (green tones) but we agree with the reviewer's suggestion and chose more contrasting colors to improve clarity.

2): In Fig. 5h (and Fig. 4f), why was AR staining mostly cytoplasmic?

The reviewer is right when saying that AR is overall mostly cytoplasmic in PrEC cells. However this is not surprising for basal cells, which are characterized by very low AR signaling. In addition it is known that AR nuclear translocation is favored by stimulation with its ligands in cells with normal androgen signaling [Nguyen MM et al. *Mol Cell Endocrinol.* 2009], though AR staining may be invariably nuclear in tumor cells characterized by constitutive AR activation.

In our context, AR spots can be detected in the nucleus of primary basal cells either when they acquire luminal features through *LEADR* silencing or IFN stimulation, or when they are stimulated by DHT. Notably, the highest nuclear staining was observed when cells silenced for *LEADR* or stimulated with IFN were simultaneously exposed to DHT, an observation that perfectly matched with PSA data.

Unfortunately, immunofluorescence images included in the previous version of the paper were too small to appreciate AR nuclear spots. To make this more evident, we took novel pictures using confocal microscopy, which allowed us to zoom at higher resolution and at the same time make sure that AR spots were in the nucleus. Enlarged captions are now shown as main figure (Fig. 4f for *LEADR* silencing and Fig. 5h for IFN stimulation), whereas full images are shown as Supplementary Figures (Supplementary Fig. 4g for *LEADR* silencing and Supplementary Fig. 5b for IFN stimulation). Both magnifications are shown here below for the reviewer.

Immunofluorescence data were also confirmed by western blotting on fractionated cytoplasmic and nuclear fractions (**Fig. R3**). This experiment showed that:

- *LEADR* silencing or interferon treatment alone were able to increase nuclear full length (~110 kDa) AR levels, aside increasing overall AR levels as previously shown in the paper. Notably, both treatments increased nuclear to cytoplasmic AR ratio, suggesting that nuclear translocation occurs when cells assume luminal phenotype through these approaches (**Fig. R3A**);
- DHT stimulation did not alter per se nuclear or cytoplasmic abundance of full length AR, though it increased a nuclear-specific band of about 97 kDa (**Fig. R3A**). Such form is slightly smaller than full length AR detected in both whole cell and cytoplasmic lysates, but is distinct from truncated AR forms visible in the range of 70-80 kDa. Hypothetically it could be exon skipping or cleaved variant specifically linked to DHT function. Double band pattern of nuclear AR is however evident in several papers in the literature [Mahmoud AM et al. *PLoS One*. 2013; Wan R et al. *J Endocrinol*. 2013; Sobel RE et al. *In Vitro Cell Dev Biol Anim*. 2006; Belikov S et al. *Biochem Biophys Rep*. 2015];
- *LEADR* silencing or interferon treatment in combination with DHT stimulation induced the highest nuclear translocation. Specifically, this enhanced both full length and 97kDa AR protein amount (**Fig. R3A**).

Notably, the amount of nuclear full length AR (or its nucleus/cytoplasm ratio) highly correlated with PSA production (correlation coefficient of 0.85 and 0.82 respectively), used as surrogate for AR transcriptional activity, whereas lower correlation was found for 97 kDa form ($r=0.59$) and for cytoplasmic AR ($r=0.7$) (**Fig. R3B**). This suggests that though 97 kDa is nuclear-specific and intimately linked with DHT stimulation, it is not suggestive per se of active AR signaling. In this regard, modulation and nuclear translocation of full length AR form induced by *LEADR* silencing or interferon stimulation perfectly mirrored changes in PSA secretion.

A

B

Figure R3

6. Figure 6:

1): Fig. 6a: It's a bit surprising that most altered genes upon LEADR KD by siLEADR vs. gapLEADR did not overlap. Can the authors offer some explanations?

We agree with the referee that this may be a bit surprising. Intuitively, it would be hence expected that the same target genes are altered when silencing a gene with different strategies. In reality, antisense oligomers with different chemistry and mechanism of action produce a number of off-target effects that may also be non-random and specific for the given molecule. This premise made, only genes that overlap between the two different approaches may be regarded as specific *LEADR* targets. As far as the degree of overlap is concerned, it is worth mentioning that a systematic comparison of transcriptome changes occurring after silencing of a given gene using different methods has not been often performed in the literature. A very recent report published in *Nucleic Acids Research* attempted to compare transcriptional changes arising from modulation of selected coding and non-coding genes using siRNA, gapmer and CRISPRi [Stojic L, et al. *Nucleic Acids Res.* 2018]. Surprisingly, the authors found from very poor to even null overlap among genes differentially expressed upon silencing with the different methods. However, as they mentioned in the paper, in almost the cases except one, the global phenomena measured in terms of deregulated pathways (instead of single genes) were concordant among all methods. In a case, depletion of a lincRNA with unknown function using two alternative silencing methods even determined completely different biological readouts, thus potentially leading to different biological conclusions.

In light of these data, our results appear very promising. First of all, the same phenotypic effects (differentiation) were observed using the different silencing methods. Consistent with this evidence, superimposable pathway enrichments, showing regulation of interferon pathway, emerged from GSEA analysis (see heatmap of Fig. 5). To comment on the overlap between differentially expressed genes upon siRNA and gapmer transfection (shown in Venn diagrams of Fig. 6), we calculated the expected number of shared genes between the two sets if differentially expressed genes were randomly sampled from the pool of all genes. Repeating 1000 times the random selection of differentially expressed genes, we obtained a histogram where the expected value of overlap due by chance was estimated to be around 20-25 genes (**Fig. R4**). As a consequence, 136 overlapping genes are much more than those expected by chance. As comparison, we may cite Stojic paper where, upon *Ch-TOG* silencing by siRNA and CRISPRi, only 5 (of 693 and 87, respectively) changed in the same direction, which is instead close to the expected number of shared genes by chance (2.8).

Figure R4

7. Figure 7:

1): In Fig. 7b, 10 of the 51 ‘LEADR-signature genes’ are histone and histone variant genes. Authors should offer some insights on this interesting finding.

We thank the reviewer for this nice observation. Actually, a number of ‘LEADR-signature genes’ are histone and histone variant genes. This may account for another potentially interesting downstream LEADR effect, which could be a genome-wide transcriptional reprogramming through histone regulation. This aspect has not been touched in this manuscript, but we now mention in the discussion that it will warrant future investigation as additional layer of gene regulation by LEADR (page 19, lines 415-418): *“Moreover, the observation that several ‘LEADR-signature genes’ are histone and histone variants (Fig. 7b) prompts the hypothesis that an additional layer of LEADR-mediated control for gene expression may exist, consisting of genome-wide transcriptional reprogramming through histone modulation.”*

2): IRF-7 is strongly induced either by interrupting expression of LEADR gene or addition of IFN-beta (Fig. 5j). Both RIP and UV CLIP assays showed binding of LEADR to IRF7 and IRF1 (Fig. 7e, f). How come IRF1 did not appear in the ‘LEADR-signature gene’ (Fig. 7b) and IRF7 was not examined in the pulldown assays (Fig. 7g)?

The reviewer is right when saying that it is not straightforward why we focused on IRF1 rather than IRF7 and we apologize for this.

IRF7 belongs to the LEADR signature list, it is evidently modulated in both RWPE-1 and DU145 cells upon LEADR manipulation, as from microarray data (mRNA) and western blotting (protein). In contrast, IRF1 never comes out as differentially expressed from gene expression data, nor is modulated at protein level, as evidenced by the western blotting reported here below (**Fig. R5**).

Figure R5

Nonetheless, IRF1 binding sites within promoters of *LEADR* targets were predicted and also validated by publicly available ChIPseq data (page 13, lines 288-292; Supplementary Table 4 and analysis of GSE31477 dataset). Altogether these pieces of evidence prompted us to consider IRF1 as the possible master regulator of *LEADR* effect on transcription (IRF7 is instead to be regarded as one of the downstream targets of *LEADR*/IRF1 axis). Accordingly, RIP and CLIP data showed higher enrichment of *LEADR* RNA in IRF1 rather than IRF7 immunoprecipitants, which stimulated us to confirm IRF1/*LEADR* interaction by RNA pulldown. ChIP data reported in Fig. 7 again confirm a possible involvement of IRF1 in *LEADR*-mediated gene regulation, with IRF1 binding increasing in RWPE-1 cells upon *LEADR* silencing (Fig. 7h) and decreasing in DU145 cells upon *LEADR* overexpression (consistent with target gene expression changes) (Fig. 7i).

Altogether these results led us to speculate a model where *LEADR* buffers IRF1 binding/activity on targets without markedly changing IRF1 protein levels, rather by interacting with it. This aspect of course will be subject of future investigation, as reported in the discussion (page 19, lines 412-415) and also explained in the response to reviewer 2.

3): Also, in Fig. 7g, data is not very convincing because EGFP pulldown also showed some IRF7 binding. We thank the reviewer for the comment. However this is not surprising. Beads themselves may precipitate proteins aspecifically (especially naturally biotinylated proteins) and RNA may be even stickier than beads because it can assume secondary/tertiary structures which favor interaction with proteins. In the experiment we actually allowed RNA to assume its structure. *EGFP* RNA control was included in the experiment with the purpose of correcting for aspecific RNA-protein binding, though in the literature pulldown is often shown comparing the RNA of interest with beads only. We performed densitometry of the blot reported in Fig. 7g (and added quantification, as also reported here below), which showed an about 4-fold enrichment of IRF1 protein pulled down by *LEADR* compared to *EGFP* RNA, which is not negligible, and corresponds to 30% of input.

g

These results are in trend with pulldown data already published in prestigious journals. For example Leucci et al. [*Nature* 2016] claimed that *SAMMSON* lincRNA interacts with p32 protein by reporting a pulldown assay where *HPRT* RNA, used as control, shows some binding to the study protein, and *SAMMSON* pulled down sample is enriched in p32 protein not less than how *LEADR* pulled down sample is enriched in IRF1. In another paper [Zhang Y et al. *Nature Communications* 2018], the authors claimed that *CCRR* RNA interacts with CIP85 protein, by showing that it is able to precipitate about 40% of input CIP85 protein, which again is in trend with our findings.

These premises made, it is worth considering that possible interaction between *LEADR* and IRF1 was also suggested by the results of protein-centric methods (RIP and CLIP). Of course, as also explained to reviewer 2, we are aware that the exact mechanism by which the lincRNA interferes with the transcriptional activity of IRF1 and more in general the dynamics of reciprocal *LEADR*/IRF1/target gene interaction are areas deserving future investigation. For this reason, we modified the text in different parts (page 16, lines 340-342; page 18, lines 387-388; page 18, line 402) to make it more evident that our model, at least in some of its aspects, is speculative and needs future research for complete dissection. For example, we now state in the discussion (page 19, lines 412-415) that: “Further investigation is required to fully understand the molecular nature of *LEADR* binding to Alu elements (e.g. DNA/RNA pairing), the exact mechanism by which the lincRNA interferes with IRF1 transcriptional activity, and the dynamics of reciprocal *LEADR*/IRF1/target gene interaction.”

8. Figure 8: The GSEA plots in panels b and c should be remade/re-labeled for better and clearer presentations with NES, p, and FDR values indicated in each plot. Similar presentations go for other GSEA plots (e.g., Fig. S5b).

Thanks for the suggestion. We re-labeled all GSEA plots, by clearly indicating the NES, p, and FDR values, thus helping appreciate the statistical power of the results.

Minor points:

9. On page 38, it is stated that “Overexpression of MIR205HG/*LEADR* was performed by using a pCMV-6AC plasmid vector containing alternatively the whole genomic sequence, the RefSeq or 7063 transcript, as synthesized by OriGene”. Similar statement was made on page 50, “Overexpression experiments with wild type *LEADR* (RefSeq, 7062 and whole genomic sequence ‘gene’) were conducted in quadruplicate (each including empty and *LEADR*-specific vector), ...”. The meaning of “whole genomic sequence” is still confusing. How long is it? Does it include the intron fragment that generates miR-205? Authors should indicate in each figure which transcript (7062, 7063 or 7057 etc) was used for overexpression of *LEADR* and/or miR-205.

We are sorry for not being clear. When we use the term “whole genomic sequence”, we intend a vector where the full genomic sequence of *LEADR* from exon1 to exon 5.2, including all introns (among which the one that generates *miR-205*), has been cloned, thus making it possible for the transfected cells to simultaneously produce all *LEADR* transcripts and *miR-205*. In contrast, vectors for specific *LEADR* transcripts, such as RefSeq (i.e. 7057) or 7063, only carry exons combined to produce that specific isoform. Accordingly, transfection of DU145 cells with RefSeq vector restored *LEADR* expression only, whereas transfection with *LEADR* full genomic sequence also allowed production of *miR-205* (Fig. 2c; note that for uniformity with panel b and d, we changed plotting of panel c. Now measured genes are on x-axis and type of transfection, gene or RefSeq, as colored series). This has been clarified also in the text (page 22, lines 482-483). We also made sure to indicate in each figure which vector was used for *LEADR*/*miR-205* overexpression. For simplicity, we preferred the nomenclature RefSeq instead of 7057.

10. For WB blots shown, full blot should be presented with clearly marked M.W markers indicated. Thanks for the suggestion. We included M.W. markers in all full blots (see Supplementary Figures). We also included predicted M.W. of the proteins of interest in both full and cropped blots.

11. There are still many language issues such as the wording ‘...a global repressive attitude; page 11, second paragraph).

We apologize for this. We now made our manuscript edited for language by a professional.

Reviewer #2, Expertise: lncRNA mediated regulation of gene expression (Remarks to the Author):

This revised version of the manuscript has improved by incorporating a number of changes.

However, the model where LEADeR binds to both DNA at Alu elements as well as IRF1 protein is not strongly supported by the data.

For instance, the authors claim that the interaction between LEADeR and Alu motifs is mediated by sequence complementarity between RNA and DNA. However at the same time they argue that the RNA is highly structured in this region. If the interaction involves extensive DNA-RNA base pairing, the RNA should be exposed in single stranded conformation. The structural prediction of LEADeR is not informative and does not support the model. In addition, the RNA pulldown data showing interaction between IRF1 and LEADeR (figure 7g) is quite poor. Prior publication the authors should either further reinforce or modify this part, leaving it as a speculative model.

We are glad the reviewer recognized our efforts in implementing and improving our manuscript.

We are sorry we could not comment on RNA structure prediction during the first round of revision. We were stimulated to do prediction of *LEADR* secondary structure by reading of the paper on *ANRIL* by Holdt LM [*Plos Genetics* 2013]. In that published work, the authors show that the lncRNA *ANRIL*, for which they demonstrate a function based on the presence of an *Alu*-element (though not demonstrating direct *Alu/Alu* interaction), folds in a way that the *Alu* is located in a stem-loop structure. They conclude that stem-loop structure of *Alu* element in *ANRIL* RNA suggests RNA-chromatin interaction as a potential effector mechanism [*Holdt LM Plos Genetics* 2013]. We reasoned that this could be also the case of *LEADR* and commented accordingly in the previous version of our manuscript. We missed to point out that motifs 1 and 4, for which we demonstrated major enrichment in target gene promoters and showed to be essential for *LEADR* function, are actually positioned in the least structured domains of the whole *Alu* sequence, thus not excluding their possible exposure in single strand conformation. However, we are aware that such prediction of secondary structure is very preliminary and made using the Vienna package with default parameters (minimum free energy). This likely does not take into account of the physiological context where *LEADR* can fold and interact with the DNA. Being not expert of structural biochemistry, we agree with the referee that these data are not sufficiently informative in this form to support or not the model, so we decided to remove them from the manuscript. We now also made explicit that though *LEADR* physically interacts with the DNA, as demonstrated by ChIRP (Fig. 7d), DNA/RNA complementarity is only one of the possible ways in which *LEADR Alu* interacts with *Alu*-containing promoters of target genes (page 15, line 329; page 18, lines 387-388; page 19, line 412-415). In this regard we had already stated in the original version of this manuscript that “*Notably, proof of direct Alu_{DNA}/Alu_{RNA} interaction has not yet been documented for *Alu*-containing lincRNAs, because this would require sophisticated structural insights*” (page 18, lines 394-395). We also changed the abstract accordingly (“... *MIR205HG* directly binds *Alu* elements located in the promoter of target genes” was changed into “... *MIR205HG* directly binds the promoters of its target genes, which have an *Alu* element in proximity of the Interferon-Related Factor (*IRF*) binding site”) (page 2, lines 31-33).

Regarding the reviewer’s comment on how the model is overall supported by the data, we would like to point out that in this work we provided several insights into previously unknown aspects of *LEADR*

expression pattern, transcript configurations, biological function and mechanism of action. Regarding the mechanistic model for gene expression regulation, we showed that:

- 1) *LEADR* physically binds to target gene promoters by ChIRP (Fig. 7c-d);
- 2) Motifs in *Alu* element present in *LEADR* RNA are essential for the lincRNA to regulate gene expression (Figure 5b, 6c, Supplementary Figures 6d-e);
- 3) In the current version, we also show that *Alu*-deleted *LEADR* does not bind to target genes, as assessed by ChIRP (Supplementary Figures 6d-e). This strengthens the hypothesis that *LEADR* action relies on *Alu* element for DNA binding, though not proving that this occurs through direct DNA/RNA pairing;
- 4) *LEADR* modulation in either direction (silencing/overexpression) influences IRF1 occupancy on target genes (Fig. 7h-i), which is consistent with repression or activation of such genes in gene expression experiments (Supplementary Fig. 6a).

We speculated that interference with IRF1 functions may rely on direct physical interaction between *LEADR* and IRF1 protein, as from results independently obtained by RIP, CLIP and RNA pulldown. Regarding the quality of pulldown experiment, our data are in trend with others recently published in prestigious journals [Leucci *et al. Nature* 2016; Zhang Y *et al. Nature Communications* 2018], as explained in response 7.3 to reviewer 1.

Made this premise, we agree with the referee that the exact mechanism by which the lincRNA interferes with the transcriptional activity of IRF1 and more in general the dynamics of reciprocal *LEADR*/IRF1/target gene interaction are areas deserving future investigation. For this reason, we modified the text in different parts (colored in red: page 16, lines 340-342; page 18, lines 387-388; page 18, line 402; page 52, lines 1149-1152) to make it more evident that our model, at least in some of its aspects, is speculative and needs future research for complete dissection. For example we now state in the discussion (page 19, lines 412-415) that: “*Further investigation is required to fully understand the molecular nature of LEADR binding to Alu elements (e.g. DNA/RNA pairing), the exact mechanism by which the lincRNA interferes with IRF1 transcriptional activity, and the dynamics of reciprocal LEADR/IRF1/target gene interaction.*”

Reviewer #3, Expertise: LncRNA ID/processing and cancer (Remarks to the Author):

My questions have been addressed, and the paper has been significantly improved.

We thank the reviewer for the extremely positive comment and we are glad of fulfilling all of his/her requests.

References

Lagarde J, Uszczynska-Ratajczak B, Carbonell S, Pérez-Lluch S, Abad A, Davis C, Gingeras TR, Frankish A, Harrow J, Guigo R, Johnson R. High-throughput annotation of full-length long noncoding RNAs with capture long-read sequencing. *Nat Genet.* 2017 Dec;49(12):1731-1740. doi: 10.1038/ng.3988. Epub 2017 Nov 6. PubMed PMID: 29106417; PubMed Central PMCID: PMC5709232.

Uszczynska-Ratajczak B, Lagarde J, Frankish A, Guigó R, Johnson R. Towards a complete map of the human long non-coding RNA transcriptome. *Nat Rev Genet.* 2018 Sep;19(9):535-548. doi: 10.1038/s41576-018-0017-y. Review. PubMed PMID: 29795125.

Chang TC, Pertea M, Lee S, Salzberg SL, Mendell JT. Genome-wide annotation of microRNA primary transcript structures reveals novel regulatory mechanisms. *Genome Res.* 2015 Sep;25(9):1401-9. doi: 10.1101/gr.193607.115. PubMed PMID: 26290535; PubMed Central PMCID: PMC4561498.

- Marzi MJ, Ghini F, Cerruti B, de Pretis S, Bonetti P, Giacomelli C, Gorski MM, Kress T, Pelizzola M, Muller H, Amati B, Nicassio F. Degradation dynamics of microRNAs revealed by a novel pulse-chase approach. *Genome Res.* 2016 Apr;26(4):554-65. doi: 10.1101/gr.198788.115. Epub 2016 Jan 28. PubMed PMID: 26821571; PubMed Central PMCID: PMC4817778.
- Nguyen MM, Dincer Z, Wade JR, Alur M, Michalak M, Defranco DB, Wang Z. Cytoplasmic localization of the androgen receptor is independent of calreticulin. *Mol Cell Endocrinol.* 2009 Apr 10;302(1):65-72. doi: 10.1016/j.mce.2008.12.010. Epub 2008 Dec 25. PubMed PMID: 19150386; PubMed Central PMCID: PMC2806808.
- Mahmoud AM, Zhu T, Parray A, Siddique HR, Yang W, Saleem M, Bosland MC. Differential effects of genistein on prostate cancer cells depend on mutational status of the androgen receptor. *PLoS One.* 2013 Oct 22;8(10):e78479. doi: 10.1371/journal.pone.0078479. eCollection 2013. PubMed PMID: 24167630; PubMed Central PMCID: PMC3805529.
- Wan R, Zhu C, Guo R, Jin L, Liu Y, Li L, Zhang H, Li S. Dihydrotestosterone alters urocortin levels in human umbilical vein endothelial cells. *J Endocrinol.* 2013 Aug 29;218(3):321-30. doi: 10.1530/JOE-13-0138. Print 2013 Sep. PubMed PMID: 23801677.
- Sobel RE, Wang Y, Sadar MD. Molecular analysis and characterization of PrEC, commercially available prostate epithelial cells. *In Vitro Cell Dev Biol Anim.* 2006 Jan-Feb;42(1-2):33-9. PubMed PMID: 16618209.
- Belikov S, Bott LC, Fischbeck KH, Wrangé Ö. The polyglutamine-expanded androgen receptor has increased DNA binding and reduced transcriptional activity. *Biochem Biophys Rep.* 2015 Jul 26;3:134-139. doi: 10.1016/j.bbrep.2015.07.014. eCollection 2015 Sep. PubMed PMID: 29124176; PubMed Central PMCID: PMC5668691.
- Stojic L, Lun ATL, Mangei J, Mascalchi P, Quarantotti V, Barr AR, Bakal C, Marioni JC, Gergely F, Odom DT. Specificity of RNAi, LNA and CRISPRi as loss-of-function methods in transcriptional analysis. *Nucleic Acids Res.* 2018 Jul 6;46(12):5950-5966. doi: 10.1093/nar/gky437. PubMed PMID: 29860520.
- Leucci E, Vendramin R, Spinazzi M, Laurette P, Fiers M, Wouters J, Radaelli E, Eyckerman S, Leonelli C, Vanderheyden K, Rogiers A, Hermans E, Baatsen P, Aerts S, Amant F, Van Aelst S, van den Oord J, de Strooper B, Davidson I, Lafontaine DL, Gevaert K, Vandesompele J, Mestdagh P, Marine JC. Melanoma addiction to the long non-coding RNA SAMMSON. *Nature.* 2016 Mar 24;531(7595):518-22. doi:10.1038/nature17161. PubMed PMID: 27008969.
- Zhang Y, Sun L, Xuan L, Pan Z, Hu X, Liu H, Bai Y, Jiao L, Li Z, Cui L, Wang X, Wang S, Yu T, Feng B, Guo Y, Liu Z, Meng W, Ren H, Zhu J, Zhao X, Yang C, Zhang Y, Xu C, Wang Z, Lu Y, Shan H, Yang B. Long non-coding RNA CCRR controls cardiac conduction via regulating intercellular coupling. *Nat Commun.* 2018 Oct 9;9(1):4176. doi: 10.1038/s41467-018-06637-9. PubMed PMID: 30301979; PubMed Central PMCID: PMC6177441.
- Holdt LM, Hoffmann S, Sass K, Langenberger D, Scholz M, Krohn K, Finstermeier K, Stahringer A, Wilfert W, Beutner F, Gielen S, Schuler G, Gäbel G, Bergert H, Bechmann I, Stadler PF, Thiery J, Teupser D. Alu elements in ANRIL non-coding RNA at chromosome 9p21 modulate atherogenic cell functions through trans-regulation of gene networks. *PLoS Genet.* 2013;9(7):e1003588. doi: 10.1371/journal.pgen.1003588. Epub 2013 Jul 4. PubMed PMID: 23861667; PubMed Central PMCID: PMC3701717.

REVIEWERS' COMMENTS:

Reviewer #1 (Remarks to the Author):

The authors have satisfactorily addressed my questions. There are only two minor points that need to be amended.

1. About description of Fig. 2f: The text (page 7, lines 151-152) writes "..... of miR-205-incompatible and compatible transcripts, which averagely accounted for 97.6% and 2.4%...." but the pie chart in Fig. 2f shows the opposite.
2. Page 12, line 255: '.... let....' should be '.... led.....'.

Reviewer #2 (Remarks to the Author):

The authors have addressed all my comments so I recommend the publication of the article.

Reviewer #3 (Remarks to the Author):

NA

REVIEWERS' COMMENTS:

We thank again all the reviewers for their efforts and constructive criticisms, which contributed to substantially ameliorate our work.

Reviewer #1 (Remarks to the Author):

The authors have satisfactorily addressed my questions. There are only two minor points that need to be amended.

We are happy the reviewer appreciated our efforts.

1. About description of Fig. 2f: The text (page 7, lines 151-152) writes “..... of miR-205-incompatible and compatible transcripts, which averagely accounted for 97.6% and 2.4%...” but the pie chart in Fig. 2f shows the opposite.

We apologize for the inconsistency. We corrected accordingly.

2. Page 12, line 255: ‘.... let....’ should be ‘.... led.....’.

Thanks for the observation. We modified accordingly.

Reviewer #2 (Remarks to the Author):

The authors have addressed all my comments so I recommend the publication of the article.

We thank the reviewer for the positive comment.

Reviewer #3 (Remarks to the Author):

NA